# Local Differential Privacy for Regret Minimization in Reinforcement Learning

**Evrard Garcelon**
Facebook AI Research & CREST, ENSAE
Paris, France
evrard@fb.com

**Vianney Perchet**
CREST, ENSAE Paris & Criteo AI Lab
Palaiseau, France,
vianney@ensae.fr

**Ciara Pike-Burke**
Imperial College London
London, United Kingdom
c.pikeburke@gmail.com

**Matteo Pirotta**
Facebook AI Research
Paris, France
matteo.pirotta@gmail.com

## Abstract

Reinforcement learning algorithms are widely used in domains where it is desirable to provide a personalized service. In these domains it is common that user data contains sensitive information that needs to be protected from third parties. Motivated by this, we study privacy in the context of finite-horizon Markov Decision Processes (MDPs) by requiring information to be obfuscated on the user side. We formulate this notion of privacy for RL by leveraging the local differential privacy (LDP) framework. We establish a lower bound for regret minimization in finite-horizon MDPs with LDP guarantees which shows that guaranteeing privacy has a multiplicative effect on the regret. This result shows that while LDP is an appealing notion of privacy, it makes the learning problem significantly more complex. Finally, we present an optimistic algorithm that simultaneously satisfies $\varepsilon$-LDP requirements, and achieves $\sqrt{K}/\varepsilon$ regret in any finite-horizon MDP after $K$ episodes, matching the lower bound dependency on the number of episodes $K$.

## 1 Introduction

The practical successes of Reinforcement Learning (RL) algorithms have led to them becoming ubiquitous in many settings such as digital marketing, healthcare and finance, where it is desirable to provide a personalized service [e.g., 1, 2]. However, users are becoming increasingly wary of the amount of personal information that these services require. This is particularly pertinent in many of the aforementioned domains where the data obtained by the RL algorithm are highly sensitive. For example, in healthcare, the state encodes personal information such as gender, age, vital signs, etc. In advertising, it is normal for states to include browser history, geolocalized information, etc. Unfortunately, [3] has shown that, unless sufficient precautions are taken, the RL agent leaks information about the environment (i.e., states containing sensitive information). That is to say, observing the policy computed by the RL algorithm is sufficient to infer information about the data (e.g., states and rewards) used to compute the policy (scenario ①). This puts users' privacy at jeopardy. Users therefore want to keep their sensitive information private, not only to an observer but also to the service provider itself (i.e., the RL agent). In response, many services are adapting to provide stronger protection of user privacy and personal data, for example by guaranteeing privacy directly on the user side (scenario ②). This often means that user data (i.e., trajectories of states, actions, rewards) are privatized before being observed by the RL agent. In this paper, we study the effect that this has on the learning problem in RL.

35th Conference on Neural Information Processing Systems (NeurIPS 2021).

Differential privacy (DP) [4] is a standard mechanism for preserving data privacy, both on the algorithm and the user side. The $(\varepsilon, \delta)$-DP definition guarantees that it is statistically hard to infer information about the data used to train a model by observing its predictions, thus addressing scenario ①. In online learning, $(\varepsilon, \delta)$-DP has been studied in the multi-armed bandit framework [e.g., 5, 6]. However, [7] showed that DP is incompatible with regret minimization in the contextual bandit problems. This led to considering weaker or different notions of privacy [e.g., 7, 8]. Recently, [9] transferred some of these techniques to RL, presenting the first private algorithm for regret minimization in finite-horizon problems. In [9], they considered a relaxed definition of DP called *joint differential privacy* (JDP) and showed that, under JDP constraints, the regret only increases by an additive term which is logarithmic in the number of episodes. Similarly to DP, in the JDP setting the privacy burden lies with the learning algorithm which directly observes user states and trajectories containing sensitive data. In particular, this means that the data itself is not private and could potentially be used –for example by the owner of the application– to train other algorithms with no privacy guarantees. An alternative and stronger definition of privacy is *Local Differential Privacy* (LDP) [10]. This requires that the user's data is protected at collection time before the learning agent has access to it. This covers scenario ② and implies that the learner is DP. Intuitively, in RL, LDP ensures that each sample (states and rewards associated to an user) is already private when observed by the learning agent, while JDP requires computation on the entire set of samples to be DP. Recently, [11] showed that, in contrast to DP, LDP is compatible with regret minimization in contextual bandits.[1] LDP is thus a stronger definition of privacy, simpler to understand and more user friendly. These characteristics make LDP more suited for real-world applications. However, as we show in this paper, guaranteeing LDP in RL makes the learning problem more challenging.

**Contributions.** In this paper, we study LDP for regret minimization in finite horizon reinforcement learning problems with $S$ states, $A$ actions, and a horizon of $H$.[2] Our contributions are as follows. **1)** We provide a regret lower bound for $(\varepsilon, \delta)$-LDP of $\Omega\big(H\sqrt{SAK}/\min\{e^\varepsilon - 1, 1\}\big)$, showing LDP is inherently harder than JDP, where the lower-bound is only $\Omega\big(H\sqrt{SAK} + SAH\log(KH)/\varepsilon\big)$ [9]. **2)** We propose the first LDP algorithm for regret minimization in RL. We use a general privacy-preserving mechanism to perturb information associated to each trajectory and derive LDP-OBI, an optimistic model-based $(\varepsilon, \delta)$-LDP algorithm with regret guarantees. **3)** We present multiple privacy-preserving mechanisms that are compatible with LDP-OBI and show that their regret is $\widetilde{O}(\sqrt{K}/\varepsilon)$ up to some mechanism dependent terms depending on $S, A, H$. **4)** We perform numerical simulations to evaluate the impact of LDP on the learning process. For comparison, we build a Thompson sampling algorithm [e.g., 12] for which we provide LDP guarantees but no regret bound.

**Related Work.** The notion of differential privacy was introduced in [4] and is now a standard in machine learning [e.g., 13, 14, 15]. Several notions of DP have been studied in the literature, including the standard DP and LDP notions. While LDP is a stronger definition of privacy compared to DP, recent works have highlighted that it possible to achieve a trade-off between the two settings in terms of privacy and utility. The shuffling model of privacy [16, 17, 18, 19, 20] allows to build $(\varepsilon, \delta)$-DP algorithm with an additional $(\varepsilon', \delta')$-LDP guarantee (for $\varepsilon' \approx \varepsilon + \ln(n)$, any $\delta' > 0$ where $n$ is the number of samples), hence it is possible to trade-off between DP, LDP, and utility in this setting. However, the scope of this paper is ensuring $(\varepsilon, \delta)$-LDP guarantees for a fixed $\varepsilon$. In this case, shuffling will not provide an improvement in utility (error) (see Thm 5.2 in Sec. 5.1 of [17] and App. I).

The bandit literature has investigated different privacy notions, including DP, JDP and LDP [5, 6, 21, 7, 22, 23, 11, 24]. In contextual bandits, [7] derived an impossibility result for learning under DP by showing a regret lower-bound $\Omega(T)$ for any $(\epsilon, \delta)$-DP algorithm. Since the contextual bandit problem is a finite-horizon RL problem with horizon $H = 1$, this implies that DP is incompatible with regret minimization in RL as well. Regret minimization in RL with privacy guarantees has only been considered in [9], where the authors extended the JDP approach from bandit to finite-horizon

---

[1]This shows that there are peculiarities in the DP definitions that are unique to sequential decision-making problems such as RL. The discrepancy between DP and LDP in RL is due to the fact that, when guaranteeing DP, actions taken by the learner cannot depend on the current state (this would break the privacy guarantee). On the other hand, in the LDP setting, the user executes a policy prescribed by the learner on its end (i.e., directly on non-private states) and send a privatized result (sequence of states and rewards observed by executing the policy) to the learner. Hence the user can execute actions based on its current state leading to a sublinear regret.

[2]We do not explicitly focus on preventing malicious attacks or securing the communication between the RL algorithm and the users. This is outside the scope of the paper.

RL problems. They proposed a variation of UBEV [25] using a randomized response mechanism to guarantee $\varepsilon$-JDP with an additive cost to the regret bound. While *local differential privacy* [10] has attracted increasing interest in the bandit literature [e.g., 21, 23, 11, 24], it remains unexplored in the RL literature, and we provide the first contribution in that direction. Finally, outside regret minimization, DP has been studied in off-policy evaluation [26], in control with DP guarantees on only the reward function [27], and in distributional RL [28].

## 2 Preliminaries

We consider a finite-horizon time-homogeneous Markov Decision Process (MDP) [29, Chp. 4] $M = (\mathcal{S}, \mathcal{A}, p, r, H)$ with state space $\mathcal{S}$, action space $\mathcal{A}$, and horizon $H \in \mathbb{N}^+$. Every state-action pair is characterized by a reward distribution with mean $r(s, a)$ supported in $[0, 1]$ and a transition distribution $p(\cdot|s, a)$ over next state.[3] We denote by $S = |\mathcal{S}|$ and $A = |\mathcal{A}|$ the number of states and actions. A non-stationary Markovian deterministic (MD) policy is defined as a collection $\pi = (\pi_1, \ldots, \pi_H)$ of MD policies $\pi_h : \mathcal{S} \to \mathcal{A}$. For any $h \in [H] := \{1, \ldots, H\}$ and state $s \in \mathcal{S}$, the value functions of a policy $\pi$ are defined as $Q_h^\pi(s, a) = r(s, a) + \mathbb{E}_\pi \left[ \sum_{i=h+1}^{H} r(s_i, a_i) \right]$ and $V_h^\pi(s) = Q^\pi(s, \pi_h(s))$. There exists an optimal Markovian and deterministic policy $\pi^\star$ [29, Sec. 4.4] such that $V_h^\star(s) = V_h^{\pi^\star}(s) = \max_\pi V_h^\pi(s)$. The Bellman equations at stage $h \in [H]$ are defined as $Q_h^\star(s, a) = r_h(s, a) + \max_{a'} \mathbb{E}_{s' \sim p_h(s, a')} \left[ V_{h+1}^\star(s') \right]$. The value iteration algorithm (a.k.a. backward induction) computes $Q^\star$ by applying the Bellman equations starting from stage $H$ down to 1, with $V_{H+1}^\star(s) = 0$ for any $s$. The optimal policy is simply the greedy policy: $\pi_h^\star(s) = \arg\max_a Q_h^\star(s, a)$. By boundness of the reward, all value functions $V_h^\pi(s)$ are bounded in $[0, H - h + 1]$ for any $h$ and $s$.

**The general interaction protocol.** The learning agent (e.g., a personalization service) interacts with an unknown MDP with multiple users in a sequence of episodes $k \in [K]$ of fixed length $H$. At each episode $k$, an user $u_k$ arrives and their personal information (e.g., location, gender, health status, etc.) is encoded by the state $s_{1,k}$. The learner selects a policy $\pi_k$ that is sent to the user $u_k$ for local execution on "clear" states. The outcome of the execution, i.e., a trajectory, $X_k = (s_{kh}, a_{kh}, r_{kh}, s_{k,h+1})_{h \in [H]}$ is sent to the learner to update the policy. Note that we have not yet explicitly taken into consideration privacy in here. We evaluate the performance of a learning algorithm $\mathfrak{A}$ which plays policies $\pi_1, \ldots, \pi_K$ by its cumulative regret after $K$ episodes

$$\Delta(K) = \sum_{k=1}^{K} (V_1^\star(s_{1,k}) - V_1^{\pi_k}(s_{1,k})). \tag{1}$$

### 2.1 Local Differential Privacy in RL

In many application settings, when modelling a decision problem as a finite horizon MDP, it is natural to view each episode $k \in [K]$ as a trajectory associated to a specific user. In this paper, we assume that the sensitive information is contained in the states and rewards of the trajectory. Those quantities need to be kept private. This is reasonable in many settings such as healthcare, advertising, and finance, where states encode personal information, such as location, health, income etc. For example, an investment service may aim to provide each user with investment suggestions tailored to their income, deposit amount, age, risk level, properties owned, etc. This information is encoded in the user state and evolves over time as a consequence of investment decisions. The service provides guidances in the form of a policy (e.g., where, when and how much to invest) and the user follows the strategy for a certain amount of time. After that and based on the newly acquired information the provider may decide to change the policy. However, the user may want to keep their personal and sensitive information private to the company, while still receiving a personalised and meaningful service. This poses a fundamental challenge since in many cases, this information about actions taken in each state is essential for learning and creating a personalized experience for the user. The goal of a private RL algorithm is thus to ensure that the sensitive information remains private, while preserving the learnability of the problem.

Privacy in RL has been tackled in [9] through the lens of *joint differential privacy* (JDP). Intuitively, JDP requires that when a user changes, the actions observed by the other $K - 1$ users will not

---

[3]We can simply modify the algorithm to handle step dependent transitions and rewards. The regret is then multiplied by a factor $H\sqrt{H}$.

change much [9]. The privacy burden thus lies with the RL algorithm. The algorithm has access to all the information about the users (i.e., trajectories) containing sensitive data. It then has to provide guarantees about the privacy of the data and carefully select the policies to execute in order to guarantee JDP. This approach to privacy requires the user to trust the RL algorithm to privately handle the data and not to expose or share sensitive information, and does not cover the examples mentioned above.

In contrast to prior work, in this paper, we consider *local differential privacy* (LDP) in RL. This removes the requirement that the RL algorithm observes the true sensitive data, achieving stronger privacy guarantees. LDP requires that an algorithm has access to user information (trajectories in RL) only through samples that have been privatized before being passed to the learning agent. This is different to JDP or DP where the trajectories are directly fed to the RL agent. In LDP, information is secured locally by the user using a private randomizer $\mathcal{M}$, before being sent to the RL agent. The appeal of this local model is that *privatization can be done locally on the user-side*. Since nobody other than the user has ever access to any piece of non private data, this local setting is far more private. There are several variations of LDP available in the literature. In this paper, we focus on the non-interactive setting. We argue that this is more appropriate for RL. Indeed, due to the RL interaction framework, the data generated by user $k$ is a function of the data of all users $l < k$, therefore the data are not i.i.d. and the standard definition of sequential interactivity for LDP (Eq. 1 in [10]) is not applicable. It is therefore more natural to study the non-interactive setting (Eq. 2 in [10]) in RL. We formally define this below.

Following the definition in [9], a user $u$ is characterized by a starting state distribution $\rho_{0,u}$ (i.e., for user $u$, $s_1 \sim \rho_{0,u}$) and a tree of depth $H$, describing all the possible sequence of states and rewards corresponding to all possible sequences of actions. Alg. 1 describes the LDP private interaction protocol between $K$ unique users $\{u_1, \ldots, u_K\} \subset \mathcal{U}^K$, with $\mathcal{U}$ the set of all users, and an RL algorithm $\mathfrak{A}$. For any $k \in [K]$, let $s_{1,k} \sim \rho_{0,u_k}$ be the initial state for user $u_k$ and denote by $X_{u_k} = \{(s_{k,h}, a_{k,h}, r_{k,h}) \mid h \in [H]\} \in \mathcal{X}_{u_k}$ the trajectory corresponding to user $u_k$ executing a policy $\pi_k$. We write $\mathcal{M}(X_{u_k})$ to denote the privatized data generated by the randomizer for user $u_k$. The goal of mechanism $\mathcal{M}$ is to privatize sensitive informations while encoding sufficient information for learning. With these notions in mind, LDP in RL can be defined as follows:

**Definition 1.** *For any $\varepsilon \geq 0$ and $\delta \geq 0$, a privacy preserving mechanism $\mathcal{M}$ is said to be $(\varepsilon, \delta)$-Locally Differential Private (LDP) if and only if for all users $u, u' \in \mathcal{U}$, trajectories $(X_u, X_{u'}) \in \mathcal{X}_u \times \mathcal{X}_{u'}$ and all $O \subset \{\mathcal{M}(\mathcal{X}_u) \mid u \in \mathcal{U}\}$:*

$$\mathbb{P}\left(\mathcal{M}(X_u) \in O\right) \leq e^\varepsilon \, \mathbb{P}\left(\mathcal{M}(X_{u'}) \in O\right) + \delta \tag{2}$$

*where $\mathcal{X}_u$ is the space of trajectories associated to user $u$.*

Def. 1 ensures that if the RL algorithm observes the output of the privacy mechanism $\mathcal{M}$ for two different input trajectories, then it is statistically difficult to guess which output is from which input trajectory. As a consequence, the users' privacy is preserved.

## 3 Lower Bound

We provide a lower bound on the regret that any LDP RL algorithm must incur. For this, as is standard when proving lower bounds on the regret in RL [e.g., 30, 31], we construct a hard instance of the problem. The proof (see App. B) relies on the fact that LDP acts as Lipschitz function, with respect to the KL-divergence, in the space of probability distribution.

**Theorem 2** (Lower-Bound)**.** *For any algorithm $\mathfrak{A}$ associated to a $\varepsilon$-LDP mechanism, any number of states $S \geq 3$, actions $A \geq 2$ and $H \geq 2 \log_A(S - 2) + 2$, there exists an MDP $M$ with $S$ states and $A$ actions such that: $\mathbb{E}_M(\Delta(K)) \geq \Omega\left(\frac{H\sqrt{SAK}}{\min\{\exp(\varepsilon)-1,1\}}\right)$.*

The lower bound of Thm. 2 shows that the price to pay for LDP in the RL setting is a factor $1/(\exp(\varepsilon) - 1)$ compared to the non-private lower bound of $H\sqrt{SAK}$. The regret lower bound scales multiplicatively with the privacy parameter $\varepsilon$. The recent work of [9] shows that for JDP, the regret in finite-horizon MDPs is lower-bounded by $\Omega\left(H\sqrt{SAK} + \frac{1}{\varepsilon}\right)$. Thm. 2 shows that the local differential privacy setting is inherently harder than the joint differential privacy one for small $\epsilon$, as our lower-bound scales with $\sqrt{K}/\varepsilon$ when $\varepsilon \cong 0$. Both bounds scale with $\sqrt{K}$ when $\varepsilon \to +\infty$.

| **Algorithm 1** Locally Private Episodic RL | **Algorithm 2** LDP-OBI ($\mathcal{M}$) |
|---|---|
| **Input:** Agent: $\mathfrak{A}$, Local Randomizer: $\mathcal{M}$, Users: $u_1, \ldots, u_K$ 
 **for** $k = 1$ **to** $K$ **do** 
 $\quad$ Agent $\mathfrak{A}$ computes $\pi_k$ using $\{\mathcal{M}(X_{u_l})\}_{l \in [K-1]}$ 
 $\quad$ User $u_k$ receives $\pi_k$ from agent $\mathfrak{A}$ and observes $s_{1,k} \sim \rho_{0,u_k}$ 
 $\quad$ User $u_k$ executes policy $\pi_k$ on "non-private" states and observes a trajectory $X_{u_k} = \{(s_{h,k}, a_{h,k}, r_{h,k})\}_{h \in [H]}$ 
 $\quad$ User $u_k$ sends back private data $\mathcal{M}(X_{u_k})$ to $\mathfrak{A}$ 
 **end for** | **Input:** $\delta \in (0,1)$, $\alpha > 1$, randomizer $\mathcal{M}$ with parameters $(\epsilon_0, \delta_0)$ 
 **for** $k = 1$ **to** $K$ **do** 
 $\quad$ Compute $\widetilde{p}_k$ and $\widetilde{r}_k$ as in Eq. (4) using $\{\mathcal{M}(X_{u_l})\}_{l \in [K-1]}$, $\beta_k^r$ and $\beta_k^p$ as in Prop. 4 using $\{c_{k,i}(\varepsilon_0, \delta_0, \frac{3\delta}{2k^2\pi^2})\}_i$, and $b_{h,k}$ 
 $\quad$ Compute $\pi_k$ as in Eq. (5) and send it to user $u_k$ 
 $\quad$ User $u_k$ executes policy $\pi_k$, collects trajectory $X_k$ and sends back privatized value $\mathcal{M}(X_k)$ 
 **end for** |

# 4 Exploration with Local Differential Privacy

A standard approach to the design of the private randomizer $\mathcal{M}$ is to inject noise into the data to be preserved [14]. A key challenge in RL is that we cannot simply inject noise to each component of the trajectory since this will break the *temporal consistency* of the trajectory and possibly prevent learning. In fact, a trajectory is not an arbitrary sequence of states, actions, and rewards but obeys the Markov reward process induced by a policy. Fortunately, Def. 1 shows that the output of the randomizer need not necessarily be a trajectory but could be any private information built from it. In the next section, we show how to leverage this key feature to output succinct information that preserves the information encoded in a trajectory while satisfying the privacy constraints. We show that the output of such a randomizer can be used by an RL algorithm to build estimates of the unknown rewards and transitions. While these estimates are biased, we show that they carry enough information to derive optimistic policies for exploration. We leverage these tools to design LDP-OBI, an optimistic model-based algorithm for exploration with LDP guarantees.

## 4.1 Privacy-Preserving Mechanism

Consider the locally-private episodic RL protocol described in Alg. 1. At the end of each episode $k \in [K]$, user $u_k$ uses a private randomizer $\mathcal{M}$ to generate a private statistic $\mathcal{M}(X_{u_k})$ to pass to the RL algorithm $\mathfrak{A}$. This statistic should encode sufficient information for the RL algorithm to improve the policy while maintaining the user's privacy. In *model-based* settings, a sufficient statistic is a local estimate of the rewards and transitions. Since this cannot be reliably obtained from a single trajectory, we resort to counters of visits and rewards that can be aggregated by the RL algorithm.

For a given trajectory $X = \{(s_h, a_h, r_h)\}_{h \in [H]}$, let $R_X(s,a) = \sum_{h=1}^{H} r_h \mathbb{1}_{\{s_h = s, a_h = a\}}$, $N_X^r(s,a) = \sum_{h=1}^{H} \mathbb{1}_{\{s_h = s, a_h = a\}}$ and $N_X^p(s,a,s') = \sum_{h=1}^{H-1} \mathbb{1}_{\{s_h = s, a_h = a, s_{h+1} = s'\}}$ be the true non-private statistics, which the agent will never observe. We design the mechanism $\mathcal{M}$ so that for a given trajectory $X$, $\mathcal{M}$ returns private versions $\mathcal{M}(X) = (\widetilde{R}_X, \widetilde{N}_X^r, \widetilde{N}_X^p)$ of these statistics. Here, $\widetilde{R}_X(s,a)$ is a noisy version of the cumulative reward $R_X(s,a)$, and $\widetilde{N}_X^r$ and $\widetilde{N}_X^p$ are perturbed counters of visits to state-action and state-action-next state tuples, respectively. At the beginning of episode $k$, the algorithm has access to the aggregated private statistics:

$$\widetilde{R}_k(s,a) = \sum_{l<k} \widetilde{R}_{X_{u_l}}(s,a), \ \ \widetilde{N}_k^r(s,a) = \sum_{l<k} \widetilde{N}_{X_{u_l}}^r(s,a), \ \ \widetilde{N}_k^p(s,a,s') = \sum_{l<k} \widetilde{N}_{X_{u_l}}^p(s,a,s') \quad (3)$$

We denote the non-private counterparts of these aggregated statistics as $R_k(s,a) = \sum_{l<k} R_{X_{u_l}}(s,a)$, $N_k^r(s,a) = \sum_{l<k} N_{X_{u_l}}^r(s,a)$ and $N_k^p(s,a,s') = \sum_{l<k} N_{X_{u_l}}^p(s,a,s')$, these are also *unknown* to the RL agent. Using these private statistics, we can define conditions that a private randomizer must satisfy in order for our RL agent, LDP-OBI, to be able to learn the reward and dynamics of the MDP.

**Assumption 3.** *The private randomizer $\mathcal{M}$ satisfies $(\varepsilon_0, \delta_0)$-LDP, Def. 1, with $\varepsilon_0, \delta_0 \geq 0$. Moreover, for any $\delta > 0$ and $k \geq 0$, there exist four finite strictly positive function, $c_{k,1}(\varepsilon_0, \delta_0, \delta), c_{k,2}(\varepsilon_0, \delta_0, \delta), c_{k,3}(\varepsilon_0, \delta_0, \delta), c_{k,4}(\varepsilon_0, \delta_0, \delta) \in \mathbb{R}_+^{\star}$ such that with probabilty at least*

$1 - \delta$ *for all* $(s, a, s') \in \mathcal{S} \times \mathcal{A} \times \mathcal{S}$:

$$\left| \widetilde{R}_k(s, a) - R_k(s, a) \right| \le c_{k,1}(\varepsilon_0, \delta_0, \delta), \qquad \left| \widetilde{N}_k^r(s, a) - N_k^r(s, a) \right| \le c_{k,2}(\varepsilon_0, \delta_0, \delta)$$

$$\left| \sum_{s'} N_k^p(s, a, s') - \widetilde{N}_k^p(s, a, s') \right| \le c_{k,3}(\varepsilon_0, \delta_0, \delta), \quad \left| N_k^p(s, a, s') - \widetilde{N}_k^p(s, a, s') \right| \le c_{k,4}(\varepsilon_0, \delta_0, \delta)$$

*The functions* $c_{k,1}(\varepsilon_0, \delta_0, \delta)$, $c_{k,2}(\varepsilon_0, \delta_0, \delta)$, $c_{k,3}(\varepsilon_0, \delta_0, \delta)$ *and* $c_{k,4}(\varepsilon_0, \delta_0, \delta)$ *must be increasing functions of* $k$ *and decreasing functions of* $\delta$. *We also write* $c_{k,1}(\varepsilon_0, \delta)$, $c_{k,2}(\varepsilon_0, \delta)$, $c_{k,3}(\varepsilon_0, \delta)$ *and* $c_{k,4}(\varepsilon_0, \delta)$ *when* $\delta_0 = 0$.

In Sec. 5, we will present schemas satisfying Asm. 3 and discuss their impacts on privacy and regret.

### 4.2 Our LDP Algorithm For Exploration

In this section, we introduce LDP-OBI (*Local Differentially Private Optimistic Backward Induction*), a flexible optimistic model-based algorithm for exploration that can be paired with any privacy mechanism satisfying Asm. 3. When developing optimistic algorithms it is necessary to define confidence intervals using an estimated model that are broad enough to capture the true model with high probability, but narrow enough to ensure low regret. This is made more complicated in the LDP setting, since the estimated model is defined using randomized counters. In particular, this means we cannot use standard concentration inequalities such as those used in [32, 33]. Moreover, when working with randomized counters, classical estimators like the empirical mean can even be ill-defined as the number of visits to a state-action pair, for example, can be negative.

Nevertheless, we show that by exploiting the properties of the mechanism $\mathcal{M}$ in Asm. 3, it is still possible to define an empirical model which can be shown to be close to the true model with high probability. To construct this empirical estimator, we rely on the fact that for each state-action pair $(s, a)$, $\widetilde{N}_k^r(s, a) + c_{k,2}(\varepsilon_0, \delta_0, \delta) \ge N_k^r(s, a) \ge 0$ with high probability where the precision $c_{k,2}(\varepsilon_0, \delta_0, \delta)$ ensures the positivity of the noisy number of visits to a state action-pair. A similar argument holds for the transitions. Formally, the estimated *private* rewards and transitions before episode $k$ are defined as follows:

$$\widetilde{r}_k(s, a) = \frac{\widetilde{R}_k(s, a)}{\widetilde{N}_k^r(s, a) + \alpha c_{k,2}(\varepsilon_0, \delta_0, \delta)}, \qquad \widetilde{p}_k(s' \mid s, a) = \frac{\widetilde{N}_k^p(s, a, s')}{\widetilde{N}_k^p(s, a) + \alpha c_{k,3}(\varepsilon_0, \delta_0, \delta)} \tag{4}$$

Note that unlike in classic optimistic algorithms, $\widetilde{p}_k$ is not a probability measure but a signed sub-probability measure. However, this does not preclude good performance. By leveraging properties of Asm. 3 we are able to build confidence intervals using these private quantities (see App. E).

**Proposition 4.** *For any* $\varepsilon_0 > 0$, $\delta_0 \ge 0$, $\delta > 0$, $\alpha > 1$ *and episode* $k$, *using mechanism* $\mathcal{M}$ *satisfying Asm. 3, then with probability at least* $1 - 2\delta$, *for any* $(s, a) \in \mathcal{S} \times \mathcal{A}$

$$|r(s, a) - \widetilde{r}_k(s, a)| \le \beta_k^r(s, a) = \sqrt{\frac{2 \ln\left(\frac{4\pi^2 SAHk^3}{3\delta}\right)}{\widetilde{N}_k^r(s, a) + \alpha c_{k,2}(\varepsilon_0, \delta_0, \delta)}} + \frac{(\alpha + 1)c_{k,2}(\varepsilon_0, \delta_0, \delta) + c_{k,1}(\varepsilon_0, \delta_0, \delta)}{\widetilde{N}_k^r(s, a) + \alpha c_{k,2}(\varepsilon_0, \delta_0, \delta)}$$

$$\|p(\cdot | s, a) - \widetilde{p}_k(\cdot | s, a)\|_1 \le \beta_k^p(s, a) = \sqrt{\frac{14S \ln\left(\frac{4\pi^2 SAHk^3}{3\delta}\right)}{\widetilde{N}_k^p(s, a) + \alpha c_{k,3}(\varepsilon_0, \delta_0, \delta)}} + \frac{Sc_{k,4}(\varepsilon_0, \delta_0, \delta)}{\widetilde{N}_k^p(s, a) + \alpha c_{k,3}(\varepsilon_0, \delta_0, \delta)} +$$

$$\frac{(\alpha + 1)c_{k,3}(\varepsilon_0, \delta_0, \delta)}{\widetilde{N}_k^p(s, a) + \alpha c_{k,3}(\varepsilon_0, \delta_0, \delta)}$$

The shape of the bonuses in Prop. 4 highlights two terms. The first term is reminiscent of Hoeffding bonuses as it scales with $\mathcal{O}\left(1/\sqrt{\widetilde{N}_k^p}\right)$. The other term is of order $\mathcal{O}\left(1/\widetilde{N}_k^p\right)$ and accounts for the variance (and potentially bias) of the noise added by the privacy-preserving mechanism.

As commonly done in the literature [e.g., 32, 34, 35], we use these concentration results to define a bonus function $b_{h,k}(s, a) := (H - h + 1) \cdot \beta_k^p(s, a) + \beta_k^r(s, a)$ which is used to define an optimistic value function and policy by running the following backward induction procedure:

$$Q_{h,k}(s, a)s = \widetilde{r}_k(s, a) + b_{h,k}(s, a) + \widetilde{p}_k(\cdot | s, a)^\mathsf{T} V_{h+1,k}, \quad \pi_{h,k}(s) = \arg\max_a Q_{h,k}(s, a) \tag{5}$$

where $V_{h,k}(s) = \min\{H - h + 1, \max_a Q_{h,k}(s, a)\}$ and $V_{H+1,k}(s) = 0$.

| $\mathcal{M}$ | Noise | $(\epsilon, \delta)$-LDP level | Regret $\Delta(T)$ |
|---|---|---|---|
| Laplace | $\mathrm{Lap}(6H/\varepsilon)$ | $(\varepsilon, 0)$ | $\widetilde{O}(H^3 S^2 A\sqrt{K}/\varepsilon)$ |
| Gaussian | $\mathcal{N}(0, (H/\varepsilon)^2)$ | $(\varepsilon, \delta_0)$ | $\widetilde{O}(H^3 S^2 A\sqrt{K \ln(1/\delta_0)}/\varepsilon)$ |
| Randomized Response | $\mathrm{Ber}((e^{\varepsilon/H}-1)^{-1})$ | $(\varepsilon, 0)$ | $\widetilde{O}(H^{7/2} S^2 A\sqrt{K}/\varepsilon)$ |
| Bounded Noise | See [37] and App. F.3 | $(\varepsilon, \delta_0)$ | $\widetilde{O}(H^2 S^3 A^{3/2}\sqrt{K \ln(1/\delta_0)}/\varepsilon)$ |

Table 1: Summary of the guarantees of LDP-OBI with different randomizers for $\varepsilon > 0$ and $\delta_0 > 0$. For the mechanism in this table, we have approximately $c_{k,i} = \widetilde{\mathcal{O}}(\sqrt{kH}/\varepsilon)$ for $i \in \{1, 2, 4\}$ (ignoring log terms) and $c_{k,3} = \widetilde{\mathcal{O}}(\sqrt{SkH}/\varepsilon)$

### 4.3 Regret Guarantees

We get the following general guarantees for any LDP mechanism satisfying Asm. 3 in LDP-OBI.

**Theorem 5.** *For any privacy mechanism $\mathcal{M}$ satisfying Asm. 3 with $\varepsilon > 0$, $\delta_0 \geq 0$, and for any $\delta > 0$ the regret of* LDP-OBI *is bounded with probability at least $1 - \delta$ by:*

$$\Delta(K) \leq \tilde{\mathcal{O}}\Bigg( \underbrace{HS\sqrt{AT}}_{\mathbf{0}} + SAH^2 c_{K,3}\left(\varepsilon, \delta_0, \frac{3\delta}{2\pi^2 K^2}\right) + H^2 S^2 A c_{K,4}\left(\varepsilon, \delta_0, \frac{3\delta}{2\pi^2 K^2}\right)$$
$$+ SAH c_{K,2}\left(\varepsilon, \delta_0, \frac{3\delta}{2\pi^2 K^2}\right) + SAH c_{K,1}\left(\varepsilon, \delta_0, \frac{3\delta}{2\pi^2 K^2}\right) \Bigg) \tag{6}$$

*The combination of $\mathcal{M}$ and* LDP-OBI *is also $(\varepsilon, \delta_0)$-LDP.*

Thm. 5 shows that the regret of LDP-OBI *1)* is lower bounded by the regret in non-private settings; and *2)* depends directly on the precision of the privacy mechanism used though $c_{K,1}, \ldots, c_{K,4}$. Thus improving the precision, that is to say reducing the amount of noise that needs to be added to the data to guarantee LDP of the privacy mechanism, directly improves the regret bounds of LDP-OBI. The first term in the regret bound ($\mathbf{0}$) is of the order expected in the non-private setting (see e.g., [36]). Classical results in DP suggest that the $\{c_{K,i}\}_{i \leq 4}$ terms should be *approximately* of order $\sqrt{K}/\varepsilon$ (this is indeed the case for many natural choices of randomizer). In such a case, the dominant term in (38), is no longer $\mathbf{0}$ but rather a term of order $H^2 S^2 A\sqrt{K}/\varepsilon$ (from e.g. $c_{K,4}$). The dependency on $S, A, H$ is larger than in the non-private setting. This is because the cost of LDP is multiplicative, so it also impacts the lower order terms in the concentration results (see e.g. the second term in 4), which are typically ignored in the non-private setting. In addition, this implies that variance reduction techniques for RL (e.g., based on Bernstein) classically used to decrease the dependence on $S, H$ will not lead to any improvement here. This is to be contrasted with the JDP setting where [9] shows that the cost of privacy is additive so using variance reduction techniques can reduce the dependency of the regret on $S, A, H$.

## 5   Choice of Randomizer

There are several randomizers that satisfy Asm. 3, for example Laplace [14], randomized response [13, 38], Gaussian [39] and bounded noise [40] mechanisms. Since one method can be preferred to another depending on the application, we believe it is important to understand the regret and privacy guarantees achieved by LDP-OBI with these randomizers. Tab. 1 provides a global overview of the properties of LDP-OBI with different randomized mechanism. The detailed derivations are deferred to App. F.

**Privacy.** All the mechanisms satisfy Asm. 3 but only the Laplace and Randomized Response mechanisms guarantees $(\varepsilon, 0)$-LDP. Note that in all cases, in order to guarantee a $\varepsilon$ level of privacy (or $(\varepsilon, \delta)$ for the Gaussian and bounded noise mechanisms), it is necessary to scale the parameter $\varepsilon$ proportional to $1/H$. This is because the statistics computed by the privacy-preserving mechanism are the sum of $H$ observations which are bounded in $[0, 1]$, the sensitivity[4] of those statistics is bounded

---

[4]For a function $f : \mathcal{X} \to \mathbb{R}$ the sensitivity is defined as $S(f) = \max_{x,y \in \mathcal{X}} |f(x) - f(y)|$

by $H$. Directly applying the composition theorem for DP [14, Thm 3.14] over the different counters, would lead to an upper-bound on the privacy of the mechanism of $S^2AH\varepsilon$ and corresponding regret bound of $\widetilde{O}\left((H^4S^4A^2\sqrt{K})/\varepsilon\right)$. For the randomizers that we use, the impact on $\varepsilon$ is lower thanks to fact that they are designed to exploit the structure of the input data (a trajectory).

**Regret Bound.** From looking at Table 1, we see that while all the mechanisms achieve a regret bound of order $\widetilde{O}(\sqrt{K})$ the dependence on the privacy level $\varepsilon$ varies as well as the privacy guarantees. The regret of Laplace, Gaussian and bounded noise mechanisms scale with $\varepsilon^{-1}$, whereas the randomized response has an exponential dependence in $\varepsilon$ similar to the lower bound. However, this improvement comes at the price of worse dependency in $H$ when $\varepsilon$ is small, and a worse multiplicative constant in the regret. This is due to the randomized response mechanism perturbing the counters for each stage $h \in [H]$, leading to up to $HS^2A$ obfuscated elements. This worse dependence is also observed in our numerical simulations.

For many of the randomizers, our regret bounds scale as $\widetilde{O}(H^3S^2A\sqrt{K}/\varepsilon)$. Aside from the $\sqrt{K}/\epsilon$ rate which is expected, our bounds exhibit worse dependence on the MDP characteristics when compared to the non-private setting. We believe that this is unavoidable due to the fact that we have to make $S^2A$ terms private, while the extra dependence on $H$ comes from dividing $\varepsilon$ by $H$ to ensure privacy over the whole trajectory. Moreover, the DP literature [e.g., 41, 42, 43] suggests that the extra dependency on $S, A, H$ may be inherent to model-based algorithms due to the explicit estimation of private rewards and transitions. Indeed, [43] shows that the minimax error rate in $\ell_1$ norm for estimating a distribution over $S$ states is $\Omega\left(\frac{S}{\sqrt{n}(\exp(\varepsilon)-1)}\right)$ with $n$ samples in the high privacy regime ($\varepsilon < 1$), while there is no change in the low privacy regime. This means that in the high privacy regime the concentration scales with a multiplicative $\sqrt{S}$ term which would translate directly into the regret bound. Furthermore, this results assumes that the number $n$ of samples is known to the learner. In our setting, $n$ maps to $N_k(s, a)$ which is unknown to the algorithm. Since we only observe a perturbed estimate of $n$, estimating $p(\cdot|s, a)$ here is strictly harder than the aforementioned setting.[5] This suggests that it is impossible for any model-based algorithm which directly estimates the transition probabilities to match the lower bound. However, this does not rule out the possibility of a model-free algorithm being able to match the lower bound. Designing such a model-free algorithm which is able to work with LDP trajectories is non-trivial and we leave it to future work.

Another direction for future work is to investigate whether the recently developed shuffling model [45] may be used to improve our regret bounds in the LDP setting. Preliminary investigations of the shuffling model (see App. I) show that it is not possible while preserving a fixed $\varepsilon$-LDP constraint, which is the focus of this paper. Nonetheless, if we were to relax the privacy constraint to only guarantee $\varepsilon$-JDP then the shuffling model could be used to retrieve the regret bound in [9] while guaranteeing some level of local differential privacy, although the level of LDP would be much weaker than the one considered in this paper. We believe the study of this model sitting in-between the joint and local DP settings for RL is a promising direction for future work and that the tools developed in this paper will be helpful for tackling this problem.

## 6 Numerical Evaluation

In this section, we evaluate the empirical performance of LDP-OBI on a toy MDP. We compare LDP-OBI with the *non-private* algorithm UCB-VI [32]. To the best of our knowledge there is no other LDP algorithm for regret minimization in MDPs in the literature. To increase the comparators, we introduce a novel LDP algorithm based on Thompson sampling [e.g., 12].

**LDP-PSRL.** Thompson sampling algorithms [e.g., PSRL, 12] have proved to be effective in several applications [46]. Due to their inherent randomization, one may imagine that they are also well

---

[5]We are not aware of any lower-bound in the literature that applies to this setting but we believe that the $S^2A\sqrt{KH}/\varepsilon$ dependence may be unavoidable for model-based algorithms. This is because $N_k(s, a)$ and $\widetilde{N}_k(s, a)$ differ by at most $\sqrt{kH \log(SA)}$ (which is a well-known lower bound for the counting elements problem see [44]). Intuitively this difference creates a bias when estimating each component $p(\cdot|s, a)$, a bias that would scale with the size of the support $p(\cdot|s, a)$ and the relative difference between $N_k(s, a)$ and $\widetilde{N}_k(s, a)$. Hence, the bias would scale with $S\sqrt{kH}/N_k(s, a)$. Summing over all episodes and $SA$ counters gives the conjectured result.

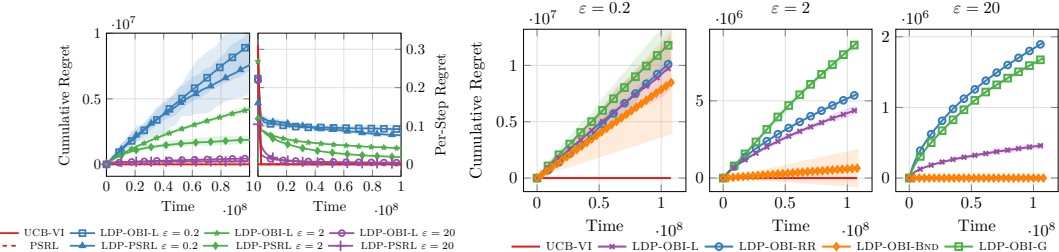

Figure 1: Evaluation of LDP-OBI with the Laplace mechanism and LDP-PSRL. *Left)* Cumulative regret. *Right)* per-step regret ($k \mapsto R_k/k$).

Figure 2: Regret for LDP-OBI coupled with different mechanisms. For all $\varepsilon$, $\delta = 0.1$ for the Gaussian and Bounded Noise mechanism.

suited to LDP regret minimization. Here, we introduce and evaluate LDP-PSRL, an LDP variant of PSRL and provide a first empirical evaluation. Informally, by defining by $\mathcal{W}_k = \{(S, A, p, r, H) : \|p - \widetilde{p}\|_1 \leq \beta_k^p, |r - \widetilde{r}| \leq \beta_k^r\}$ the *private* set of plausible MDPs constructed using the definition in Prop. 4, we can see posterior sampling as drawing an MDP from this set at each episode $k$ and running the associated optimal policy:

$$i) \, M_k \sim \mathbb{P}(\mathcal{W}_k), \quad ii) \, \pi_k = \max_{\pi}\{V_1^{\pi}(M_k)\}.$$

More formally, we consider Gaussian and Dirichlet prior for rewards and transition which lead to Normal-Gamma and Dirichlet distributions as posteriors. We use the private counters defined in Asm. 3 to update the parameters of the posterior distribution and thus the distribution over plausible models. We provide full details of this schema in App. G and show that it is LDP. However, we were not able to provide a regret bound for this algorithm.

**Simulations.** We consider the RandomMDP environment described in [25] where for each state-action pair transition probabilities are sampled from a Dirichlet($\alpha$) distribution (with $\alpha_{s,a,s'} = 0.1$ for all $(s, a, s')$) and rewards are deterministic in $\{0, 1\}$ with $r(s, a) = \mathbb{1}_{\{U_{s,a} \leq 0.5\}}$ for $(U_{s,a})_{(s,a) \in \mathcal{S} \times \mathcal{A}} \sim \mathcal{U}([0, 1])$ sampled once when generating the MDP. We set the number of states $S = 2$, number of actions $A = 2$ and horizon $H = 2$. We evaluate the regret of our algorithm for $\varepsilon \in \{0.2, 2, 20\}$ and $K = 1 \times 10^8$ episodes. For each $\varepsilon$, we run 20 simulations. Confidence intervals are the minimum and maximum runs. Fig. 1 shows that the learning speed of the optimistic algorithm LDP-OBI is severely impacted by the LDP constraint. This is consistent with our theoretical results. The reason for this is the very large confidence intervals that are needed to deal with the noise from the privacy preserving mechanism that is necessary to guarantee privacy. While the regret looks almost linear for $\varepsilon = 0.2$, the decreasing trend of the per-step regret shows that LDP-OBI-L is learning. Although these experimental results only consider a small MDP, we expect that many of the observations will carry across to larger, more practical settings. However, further experiments are needed to conclusively assess the impact of LDP in large MDPs. Fig. 1 also shows that LDP-PSRL performs slightly better than LDP-OBI. This is to be expected, since even in the non-private case PSRL usually outperforms optimistic algorithm empirically. Finally, Fig. 2 compares the mechanisms with different privacy levels and illustrates the empirical impact of the privacy-preserving mechanism on the performance of LDP-OBI. We observe empirically that the bounded noise mechanism is the most effective approach, followed by the Laplace mechanism. However, the former suffers from a higher variance in its performance.

## 7 Conclusion

We have introduced the definition of local differential privacy in RL and designed the first LDP algorithm, LDP-OBI, for regret minimization in finite-horizon MDPs. We provided an intuition why model-based approaches may suffer a higher dependence in the MDP characteristics. Designing a model-free algorithm able to reduce or close the gap with the lower-bound is an interesting technical question for future works. As mentioned in the paper, the shuffling privacy model does not provide any privacy/regret improvement in the strong LDP setting. An interesting direction is to investigate the trade-off between JDP and LDP that can be obtained in RL using shuffling. In particular, we believe that, sacrificing LDP guarantees, it is possible to achieve better regret leveraging variance reduction

techniques (that are not helpful in strong LDP settings). Finally, there are other privacy definition that can be interesting for RL. For example, profile-based privacy [47, 48] allows to privatize only specific information or geo-privacy [49] focuses on privacy between elements that are "similar".

## Acknowledgments and Disclosure of Funding

V. Perchet acknowledges support from the French National Research Agency (ANR) under grant number #ANR-19-CE23-0026 as well as the support grant, as well as from the grant "Investissements d'Avenir" (LabEx Ecodec/ANR-11-LABX-0047).

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
