., 13, 14, 15]. In stochastic multi-armed bandits, $\epsilon$-DP algorithms have been extensively studied [see e.g., 5, 6]. Recently, [22] proposed an $\epsilon$-DP algorithm for stochastic multi-armed bandits that achieves the private lower-bound presented in [7]. In contextual bandits, [7] derived an impossibility result for learning under DP by showing a regret lower-bound $\Omega(T)$ for any $(\epsilon, \delta)$-DP algorithm. Instead, they considered the relaxed JDP setting and proposed an optimistic algorithm with sublinear regret and $\epsilon$-JDP guarantees. Since the contextual bandit problem is an episodic RL problem with horizon $H = 1$, this suggests that DP is incompatible with regret minimization in RL as well.

Recently, *local differential privacy* [10] has attracted increasing interest in the bandit literature. [21] were the first to study LDP in stochastic MABs. [23] extended LDP to combinatorial bandits, and [11, 24] focused on LDP for MAB and contextual bandit. Private algorithms for regret minimization have also been investigated in multi-agent bandits (a.k.a. federated learning) in centralized and decentralized settings [e.g., 50, 51, 52], and empirical approaches have been considered in [53, 54].

In RL, [26] proposed the first private algorithm for policy evaluation with linear function approximation that ensures privacy with respect to the change of trajectories collected off-policy. [27] considered the RL problem in continuous space, where reward information is protected. They designed a private version of Q-learning with function approximation where privacy with respect to different reward functions is achieved by injecting noise in the value function. [28] recently studied LDP for actor-critic methods in the context of distributed RL. None of these works considered regret minimization under privacy constraints. Regret minimization with privacy guarantees has only been considered in RL recently. [9] designed a private optimistic algorithm for regret minimization with JDP. They proposed a variation of UBEV [25] using a randomized response mechanism with parameter $\epsilon/H$ to guarantee privacy. Their algorithm PUCB achieves a regret bound $\widetilde{O}(\sqrt{H^4 SAK} + SAH^3(S + H)/\varepsilon)$ while enjoying $\varepsilon$-JDP. Compared to the worst case regret of UBEV, the penalty for JDP privacy is only additive, as shown by their lower-bound of $\widetilde{\Omega}(H\sqrt{SAK} + SAH/\varepsilon)$.

# B    Regret Lower Bound (Proof of Thm. 2)

Let's consider the following MDP for a given number of states $S$ and actions $A$. The initial state $0$ has $A$ actions which deterministically lead the next state. The MDP is a tree with $A$ children for each node and exactly $S - 2$ states.

We denote by $x_1, \ldots, x_L$ the leaves of this tree. Each leaf can transition to one of the two terminal states denoted by $+$ and $-$, where the agent will receive reward of 1 or 0 respectively, and the agent will stay there until the end of the episode. There exists a unique action $a^\star$ and leaf $x_{i^\star}$ such that: $\mathbb{P}(+ \mid x_{i^\star}, a^\star) = 1/2 + \Delta$ for a chosen $\Delta$. Each other leaf transitions with equal probability to two states $+$ and $-$ where each has a reward of 1 and 0. All other states have a reward of 0 and every other transition is deterministic.

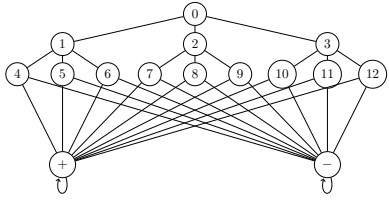

Figure 3: Example of an MDP described in this section with $S = 15$ and $A = 3$

Once the agent arrives at $+$ or $-$, it stay there until the end of the episode. In addition, we assume that $H \geq 2\ln(S - 2)/\ln(A) + 2$. Let $d > 0$ be the depth of the tree, i.e., the depth of the tree with $S - 2$ nodes is $d - 1$ and nodes $+, -$ are at depth $d$. Then leaves $x_1, \ldots, x_L$ are at depth either $d - 1$ or $d - 2$. Without loss of generality we assume that all $x_1, \ldots, x_L$ are at depth $d - 1$, i.e., the number of leaves is $L = A^{d-1} \geq (S - 2)/2$, stated otherwise, the tree without the nodes $+$ and $-$ is a perfect $A$-ary tree. In the general case we have that $L \geq (S - 2)/2$.

For a policy $\pi$, the value function can be written:

$$V^\pi(0) = (H - d)\mathbb{P}(s_d = +) = (H - d)(1/2 + \Delta\mathbb{P}(s_{d-1} = x_{i^\star}, a_{d-1} = a^\star)) \qquad (7)$$

Thus the regret can be written as:

$$R(K, I) = (H - d)\Delta\Big(K - \underbrace{\sum_{k=1}^{K}\mathbb{P}(s_{k,d-1} = x_{i^\star}, a_{k,d-1} = a^\star)}_{:=\mathbb{E}(T(K,I))}\Big) \qquad (8)$$

where $I = (x_{i^\star}, a^\star)$ is the optimal state action pair and we define $T(K, I)$ as:

$$T(K, I) = \sum_{k=1}^{K}\mathbb{1}_{\{s_{k,d-1}=x_{i^\star}, a_{k,d-1}=a^\star\}}. \qquad (9)$$

$T(K, I)$ is a function of the history observed by the algorithm. Since we consider the LDP setting, this history can be written as:

$$\mathcal{M}(\mathcal{H}_K) = \{\mathcal{M}(X_l) \mid l \leq K\} \qquad (10)$$

where $X_l = \{(s_{l,h}, a_{l,h}, r_{l,h}) \mid h \leq H\}$ is the trajectory observed by the user for episode $l$ and $\mathcal{M}$ is a privacy mechanism which maintains $\varepsilon$-LDP. Thus $T(K, I)$ is a function of $\mathcal{M}(\mathcal{H}_K)$. By Lem. A.1 in [30]:

$$\mathbb{E}(T(K, I)) \leq \mathbb{E}_0(T(K, I)) + K\sqrt{\mathrm{KL}\Big(\mathbb{P}_0(\mathcal{M}(\mathcal{H}_K)) \mid\mid \mathbb{P}(\mathcal{M}(\mathcal{H}_K))\Big)} \qquad (11)$$

where $\mathbb{E}_0$ is the expectation when $\Delta = 0$. However, because $T(K, I)$ can be seen as a function on the history only, we can use Exercise 14.4 in [31] which states that for any random variable $Y : \Omega \to [a, b]$ with $(\Omega, \mathcal{F})$ a measurable space, $a < b$ and two distributions $P$ and $Q$ on $\mathcal{F}$, then:

$$\left|\int_{w\in\Omega} Y(w)dP(w) - \int_{w\in\Omega} Y(w)dQ(w)\right| \leq (b - a)\sqrt{\frac{\mathrm{KL}(P||Q)}{2}} \qquad (12)$$

In our case the random variable $Y$ is the combination of $T(K, I)$ and the privacy mechanism $\mathcal{M}$ so we have:

$$\mathbb{E}(T(K, I)) \leq \mathbb{E}_0(T(K, I)) + K\sqrt{\mathrm{KL}\Big(\mathbb{P}_0(\mathcal{H}_K) \,||\, \mathbb{P}(\mathcal{H}_K)\Big)} \tag{13}$$

Putting together Eq. (11) and (13), we get:

$$\mathbb{E}(T(K, I)) \leq \mathbb{E}_0(T(K, I)) + K \min \left\{ \underbrace{\sqrt{\mathrm{KL}\Big(\mathbb{P}_0(\mathcal{M}(\mathcal{H}_K)) \,||\, \mathbb{P}(\mathcal{M}(\mathcal{H}_K))\Big)}}_{\text{①}}, \right.$$
$$\left. \underbrace{\sqrt{\mathrm{KL}\Big(\mathbb{P}_0(\mathcal{H}_K) \,||\, \mathbb{P}(\mathcal{H}_K)\Big)}}_{\text{②}} \right\} \tag{14}$$

**Bounding ①.** Now we bound the KL-divergence between the two measures for the history. Using the chain rule we have:

$$\mathrm{KL}\left(\mathbb{P}_0(\mathcal{M}(\mathcal{H}_K)) \,||\, \mathbb{P}(\mathcal{M}(\mathcal{H}_K))\right) = \sum_{k=1}^{K} \mathbb{E}_{\mathcal{H}_{k-1} \sim \mathbb{P}_0}\left(\mathrm{KL}\left(\mathbb{P}_0(\cdot|\mathcal{M}(\mathcal{H}_{k-1})) \,||\, \mathbb{P}(\cdot|\mathcal{M}(\mathcal{H}_{k-1}))\right)\right) \tag{15}$$

But because $\mathcal{M}$ is an $\varepsilon$-LDP mechanism, Thm. 1 in [10] ensures that:

$$\mathrm{KL}\left(\mathbb{P}_0(\cdot|\mathcal{M}(\mathcal{H}_{k-1})) \,||\, \mathbb{P}(\cdot|\mathcal{M}(\mathcal{H}_{k-1}))\right) \leq 4(\exp(\varepsilon) - 1)^2 \mathrm{KL}\left(\mathbb{P}_0(\cdot|\mathcal{H}_{k-1}) \,||\, \mathbb{P}(\cdot|\mathcal{H}_{k-1})\right) \tag{16}$$

Additionally, the KL-divergence can be written as:

$$\mathrm{KL}\left(\mathbb{P}_0(\cdot|\mathcal{H}_{k-1}) \,||\, \mathbb{P}(\cdot|\mathcal{H}_{k-1})\right) = \sum_{h=1}^{H} \mathbb{E}_{X_k \sim \mathbb{P}_0}\left(\ln\left(\frac{\mathbb{P}_0(s_{k,h}, a_{k,h}, r_{k,h} \mid \mathcal{H}_{k-1}, (s_{k,j}, a_{k,j}, r_{k,j})_{j \leq h-1})}{\mathbb{P}(s_{k,h}, a_{k,h}, r_{k,h} \mid \mathcal{H}_{k-1}, (s_{k,j}, a_{k,j}, r_{k,j})_{j \leq h-1})}\right)\right) \tag{17}$$

where $X_k = \{(s_{k,h}, a_{k,h}, r_{k,h}) \mid h \leq H\}$ is a trajectory sampled from the MDP with the transitions distributed according to $\mathbb{P}_0$ and for each step $h$, $s_{k,h}$ is a state, $a_{k,h}$ an action and $r_{k,h}$ the reward associated with $(s_{k,h}, a_{k,h})$.

Therefore for a step $h \geq 1$,

$$\ln\left(\mathbb{P}_0(s_{k,h}, a_{k,h}, r_{k,h} \mid \mathcal{H}_{k-1}, (s_{k,j}, a_{k,j}, r_{k,j})_{j \leq h-1})\right) = \ln\left(\mathbb{P}_0(s_{k,h} \mid \mathcal{H}_{k-1}, (s_{k,j}, a_{k,j}, r_{k,j})_{j \leq h-1})\right)$$
$$+ \ln\left(\mathbb{P}_0(a_{k,h} \mid \mathcal{H}_{k-1}, (s_{k,j}, a_{k,j}, r_{k,j})_{j \leq h-1}, s_{k,h})\right)$$
$$+ \ln\left(\mathbb{P}_0(r_{k,h} \mid \mathcal{H}_{k-1}, (s_{k,j}, a_{k,j}, r_{k,j})_{j \leq h-1}, s_{k,h}, a_{k,h})\right)$$

By the Markov property of the environment:

$$\ln\left(\mathbb{P}_0(s_{k,h} \mid \mathcal{H}_{k-1}, (s_{k,j}, a_{k,j}, r_{k,j})_{j \leq h-1})\right) = \ln\left(\mathbb{P}_0(s_{k,h} \mid s_{k,h-1}, a_{k,h-1})\right) \tag{18}$$

Also, since the reward only depends on the current state-action pair:

$$\ln\left(\mathbb{P}_0(r_{k,h} \mid \mathcal{H}_{k-1}, (s_{k,j}, a_{k,j}, r_{k,j})_{j \leq h-1}, s_{k,h}, a_{k,h})\right) = \ln\left(\mathbb{P}_0(r_{k,h} \mid s_{k,h}, a_{k,h})\right). \tag{19}$$

The same results holds for $\mathbb{P}$, thus:

$$\mathrm{KL}\left(\mathbb{P}_0(\cdot|\mathcal{H}_{k-1}) \,||\, \mathbb{P}(\cdot|\mathcal{H}_{k-1})\right) = \sum_{h=1}^{H} \mathbb{E}_{X_k \sim \mathbb{P}_0}\left(\ln\left(\frac{\mathbb{P}_0(s_{k,h} \mid s_{k,h-1}, a_{k,h-1})}{\mathbb{P}(s_{k,h} \mid s_{k,h-1}, a_{k,h-1})}\right)\right.$$
$$\left. + \ln\left(\frac{\mathbb{P}_0(a_{k,h} \mid \mathcal{H}_{k-1}, (s_{k,j}, a_{k,j}, r_{k,j})_{j \leq h-1}, s_{k,h})}{\mathbb{P}(a_{k,h} \mid \mathcal{H}_{k-1}, (s_{k,j}, a_{k,j}, r_{k,j})_{j \leq h-1}, s_{k,h})}\right) + \ln\left(\frac{\mathbb{P}_0(r_{k,h} \mid s_{k,h}, a_{k,h})}{\mathbb{P}(r_{k,h} \mid s_{k,h}, a_{k,h})}\right)\right) \tag{20}$$

But for $\mathbb{P}$ and $\mathbb{P}_0$ the rewards are distributed accordingly to the same distribution hence $\ln\left(\frac{\mathbb{P}_0(r_{k,h}|s_{k,h},a_{k,h})}{\mathbb{P}(r_{k,h}|s_{k,h},a_{k,h})}\right) = 0$ for each $h \leq H$. Also, the action taken at each step depends only the history of data and the current state, thus $\ln\left(\frac{\mathbb{P}_0(a_{k,h}|\mathcal{H}_{k-1}, (s_{k,j}, a_{k,j}, r_{k,j})_{j \leq h-1})}{\mathbb{P}(a_{k,h}|\mathcal{H}_{k-1}, (s_{k,j}, a_{k,j}, r_{k,j})_{j \leq h-1})}\right) = 0$. Lastly,

transition dynamics between $\mathbb{P}$ and $\mathbb{P}_0$ only differ when at step $d-1$ thus for all $h \neq d-1$, $\ln\left(\frac{\mathbb{P}_0(s_{k,h}|s_{k,h-1},a_{k,h-1})}{\mathbb{P}_0(s_{k,h}|s_{k,h-1},a_{k,h-1})}\right) = 0$. Overall, we get:

$$\mathrm{KL}\left(\mathbb{P}_0(\cdot|\mathcal{H}_{k-1}) \,||\, \mathbb{P}(\cdot|\mathcal{H}_{k-1})\right) = \sum_{l=1}^{L}\sum_{a=1}^{A}\sum_{j\in\{-,+\}} \mathbb{E}_{X_k\sim\mathbb{P}_0}\left(\ln\left(\frac{\mathbb{P}_0(j\mid x_l,a)}{\mathbb{P}(j\mid x_l,a)}\right)\mathbb{1}_{\left\{\substack{s_{k,d-1}=x_l,\\a_{k,d-1}=a,\\s_{k,d}=j}\right\}}\right)$$

Finally, for $j \in \{-,+\}$, $x_l \neq x_{i^\star}$ and $a \neq a^\star$, $\mathbb{P}(j\mid x_l,a) = \mathbb{P}_0(j\mid x_l,a)$. Hence,

$$\mathrm{KL}\left(\mathbb{P}_0(\cdot|\mathcal{H}_{k-1}) \,||\, \mathbb{P}(\cdot|\mathcal{H}_{k-1})\right) = \frac{1}{2}\ln\left(\frac{1}{1-4\Delta^2}\right)\mathbb{E}_{X_k\sim\mathbb{P}_0}\left(\mathbb{1}_{\{s_{k,d-1}=x_{i^\star},a_{k,d-1}=a^\star\}}\right) \quad (21)$$

where we have used $\mathbb{P}(+\mid x_{i^\star},a^\star) = \frac{1}{2}+\Delta$, $\mathbb{P}_0(+\mid x_{i^\star},a^\star) = \frac{1}{2}$, $\mathbb{P}(-\mid x_{i^\star},a^\star) = \frac{1}{2}-\Delta$ and $\mathbb{P}_0(-\mid x_{i^\star},a^\star) = \frac{1}{2}$.

Therefore combining (16) and (21) and summing over the episodes, we get:

$$\mathrm{KL}\left(\mathbb{P}_0(\mathcal{M}(\mathcal{H}_K)) \,||\, \mathbb{P}(\mathcal{M}(\mathcal{H}_K))\right) \leq 2(e^\varepsilon-1)^2\ln\left(\frac{1}{1-4\Delta^2}\right)\sum_{k=1}^{K}\mathbb{P}_0\left(s_{k,d-1}=x_{i^\star},a_{k,d-1}=a^\star\right)$$

$$= 2(e^\varepsilon-1)^2\ln\left(\frac{1}{1-4\Delta^2}\right)\mathbb{E}_0(T(K,I))$$

$$(22)$$

**Bounding ②.** Using again the chain rule of the KL-divergence, we have that:

$$\mathrm{KL}\left(\mathbb{P}_0(\mathcal{H}_K) \,||\, \mathbb{P}(\mathcal{H}_K)\right) = \sum_{k=1}^{K}\mathbb{E}_{\mathcal{H}_{k-1}\sim\mathbb{P}_0}\left(\mathrm{KL}\left(\mathbb{P}_0(\cdot|\mathcal{H}_{k-1}) \,||\, \mathbb{P}(\cdot|\mathcal{H}_{k-1})\right)\right) \quad (23)$$

Therefore, using Eq. (21), we have:

$$\mathrm{KL}\left(\mathbb{P}_0(\mathcal{H}_K) \,||\, \mathbb{P}(\mathcal{H}_K)\right) = \sum_{k=1}^{K}\mathbb{E}_{\mathcal{H}_{k-1}\sim\mathbb{P}_0}\left(\frac{1}{2}\ln\left(\frac{1}{1-4\Delta^2}\right)\mathbb{E}_{X_k\sim\mathbb{P}_0}\left(\mathbb{1}_{\left\{\substack{s_{k,d-1}=x_{i^\star},\\a_{k,d-1}=a^\star}\right\}}\right)\right)$$

$$= \frac{1}{2}\ln\left(\frac{1}{1-4\Delta^2}\right)\mathbb{E}_0(T(K,I))$$

$$(24)$$

**Finishing the proof.** Hence using Eq. (22) and Eq. (24) in Eq. (14):

$$\mathbb{E}(T(K,I)) \leq \mathbb{E}_0(T(K,I)) + K\min\left\{\sqrt{2}(e^\varepsilon-1),\frac{1}{\sqrt{2}}\right\}\sqrt{\mathbb{E}_0(T(K,I))\ln\left(\frac{1}{1-4\Delta^2}\right)} \quad (25)$$

Now, let's assume that $I = (x_{i^\star},a^\star)$ is distributed uniformly over $\{x_1,\ldots,x_L\}\times[A]$. That is to say, that the leaf $i^\star \sim \mathcal{U}([L])$ and given the realization of $i^\star$, $a^\star$ is drawn uniformly in the action set of node $x_{i^\star}$ i.e., $a^\star \sim \mathcal{U}([A])$. We denote the expectation over the random variable $(x_{i^\star},a^\star)$ by $\mathbb{E}_I$. It then holds that:

$$\mathbb{E}_I\mathbb{E}_0(T(K,I)) = \mathbb{E}_0\sum_{k=1}^{K}\sum_{l=1}^{L}\sum_{a=1}^{A}\frac{1}{LA}\mathbb{1}_{\{s_{k,d-1}=s,a_{k,d-1}=a\}} = \frac{K}{LA} \quad (26)$$

Therefore thanks to Jensen's inequality the regret is lower-bounded by:

$$\mathbb{E}_I R(K,I) \geq (H-d)\Delta K\left(1 - \frac{1}{LA} - \min\left\{\sqrt{2}(e^\varepsilon-1),\frac{1}{\sqrt{2}}\right\}\sqrt{\frac{K}{LA}\ln\left(1+\frac{4\Delta^2}{1-4\Delta^2}\right)}\right) \quad (27)$$

Therefore for $LA \geq 2$, $K \geq \frac{LA}{\min\{8(e^\varepsilon-1),4\}^2}$ and choosing $\Delta = \sqrt{\frac{LA}{K}}\times\frac{1}{16\sqrt{2}\min\left\{(e^\varepsilon-1),\frac{1}{2}\right\}}$ we get that:

$$\min\left\{\sqrt{2}(\exp(\varepsilon)-1),\frac{1}{\sqrt{2}}\right\}\sqrt{\frac{K}{LA}\ln\left(1+\frac{4\Delta^2}{1-4\Delta^2}\right)} \leq \frac{1}{4}$$

Hence:

$$\max_{I \in \{x_1,\ldots,x_L\} \times [A]} R(K,I) \geq \mathbb{E}_I R(K,I) \geq \frac{(H-d)\sqrt{KLA}}{64 \min\left\{(\exp(\varepsilon)-1), \frac{1}{2}\right\}} \tag{28}$$

And because $I$ is a finite random variable there exist $I^\star$ such that $\max_{I \in \{x_1,\ldots,x_L\} \times [A]} R(K,I) = R(K,I^\star)$.

$$R(K,I^\star) \geq \frac{(H-d)\sqrt{KLA}}{64 \min\left\{(\exp(\varepsilon)-1), \frac{1}{2}\right\}} \tag{29}$$

Thus we have that there exists an MDP such that its frequentist regret is $\Omega\left(\frac{H\sqrt{SAK}}{\min\{1,\exp(\varepsilon)-1\}}\right)$.

## C  Concentration under Local Differential Privacy (Proof of Prop. 4):

In this subsection, we proceed with the proof of Prop. 4 (recalled below).

**Proposition.** *For any $\varepsilon_0 > 0$, $\delta_0 \geq 0$, $\delta > 0$, $\alpha > 1$ and episode $k$, using mechanism $\mathcal{M}$ satisfying Asm. 3, then with probability at least $1 - 2\delta$, for any $(s,a) \in \mathcal{S} \times \mathcal{A}$*

$$|r(s,a) - \widetilde{r}_k(s,a)| \leq \beta_k^r(s,a) = \sqrt{\frac{2\ln\left(\frac{4\pi^2 SAHk^3}{3\delta}\right)}{\widetilde{N}_k^r(s,a) + \alpha c_{k,2}(\varepsilon_0,\delta_0,\delta)}} + \frac{(\alpha+1)c_{k,2}(\varepsilon_0,\delta_0,\delta) + c_{k,1}(\varepsilon_0,\delta_0,\delta)}{\widetilde{N}_k^r(s,a) + \alpha c_{k,2}(\varepsilon_0,\delta_0,\delta)}$$

$$\|p(\cdot|s,a) - \widetilde{p}_k(\cdot|s,a)\|_1 \leq \beta_k^p(s,a) = \sqrt{\frac{14S\ln\left(\frac{4\pi^2 SAHk^3}{3\delta}\right)}{\widetilde{N}_k^p(s,a) + \alpha c_{k,3}(\varepsilon_0,\delta_0,\delta)}} + \frac{Sc_{k,4}(\varepsilon_0,\delta_0,\delta)}{\widetilde{N}_k^p(s,a) + \alpha c_{k,3}(\varepsilon_0,\delta_0,\delta)} +$$
$$\frac{(\alpha+1)c_{k,3}(\varepsilon_0,\delta_0,\delta)}{\widetilde{N}_k^p(s,a) + \alpha c_{k,3}(\varepsilon_0,\delta_0,\delta)}$$

*Proof.* On the event that all inequalities of Def. 3 holds, we have:

$$\left|\frac{\widetilde{R}_k(s,a)}{\widetilde{N}_k^r(s,a) + \alpha c_{k,2}(\varepsilon_0,\delta_0,\delta)} - \frac{R_k(s,a)}{\widetilde{N}_k^r(s,a) + \alpha c_{k,2}(\varepsilon_0,\delta_0,\delta)}\right| \leq \frac{c_{k,1}(\varepsilon_0,\delta_0,\delta)}{\widetilde{N}_k^r(s,a) + \alpha c_{k,2}(\varepsilon_0,\delta_0,\delta)} \tag{30}$$

since $\widetilde{N}_k^r(s,a) + \alpha c_{k,2}(\varepsilon_0,\delta_0,\delta) > N_k^k(s,a) \geq 0$ with $\alpha > 1$. But, we also have that with probability $1 - \delta$:

$$\left|\frac{R_k(s,a)}{\widetilde{N}_k^r(s,a) + \alpha c_{k,2}(\varepsilon_0,\delta_0,\delta)} - r(s,a)\right| \leq \left|r(s,a)\left(\frac{N_k^r(s,a)}{\widetilde{N}_k^r(s,a) + \alpha c_{k,2}(\varepsilon_0,\delta_0,\delta)} - 1\right)\right| \tag{31}$$

$$+ \left|\frac{N_k^r(s,a)}{\widetilde{N}_k^r(s,a) + \alpha c_{k,2}(\varepsilon_0,\delta_0,\delta)} \times \underbrace{\left(\frac{R_k(s,a)}{N_k^r(s,a)} - r(s,a)\right)}_{:=\overline{r}_k(s,a) - r(s,a)}\right|$$

$$\leq \frac{N_k^r(s,a)}{\widetilde{N}_k^r(s,a) + \alpha c_{k,2}(\varepsilon_0,\delta_0,\delta)} \frac{L(\delta)}{\sqrt{N_k^r(s,a)}} + r(s,a)\left|1 - \frac{N_k^r(s,a)}{\widetilde{N}_k^r(s,a) + \alpha c_{k,2}(\varepsilon_0,\delta_0,\delta)}\right| \tag{32}$$

$$\leq \frac{L(\delta)\sqrt{N_k^r(s,a)}}{\widetilde{N}_k^r(s,a) + \alpha c_{k,2}(\varepsilon_0,\delta_0,\delta)} + \frac{(\alpha+1)c_{k,2}(\varepsilon_0,\delta_0,\delta)}{\widetilde{N}_k^r(s,a) + \alpha c_{k,2}(\varepsilon_0,\delta_0,\delta)} \tag{33}$$

where the second inequality follows from Chernoff-Hoeffding bound on the empirical non-private rewards with $L(\delta) = \sqrt{2\ln(4\pi^2 SAHk^3/3\delta)}$, and we use Def. 3 for the last. Furthermore:

$$\frac{L(\delta)\sqrt{N_k^r(s,a)}}{\widetilde{N}_k^r(s,a) + \alpha c_{k,2}(\varepsilon_0,\delta_0,\delta)} \leq \frac{L(\delta)\sqrt{\widetilde{N}_k^r(s,a) + c_{k,2}(\varepsilon_0,\delta_0,\delta)}}{\widetilde{N}_k^r(s,a) + \alpha c_{k,2}(\varepsilon_0,\delta_0,\delta)} \leq \frac{L(\delta)}{\sqrt{\widetilde{N}_k^r(s,a) + \alpha c_{k,2}(\varepsilon_0,\delta_0,\delta)}} \tag{34}$$

Therefore combining Eq. (30), (33) and (34), we have:

$$\left|\frac{\widetilde{R}_k(s,a)}{\widetilde{N}_k^r(s,a) + \alpha c_{k,2}(\varepsilon_0,\delta_0,\delta)} - r(s,a)\right| \leq \frac{c_{k,1}(\varepsilon_0,\delta_0,\delta) + (\alpha+1)c_{k,2}(\varepsilon_0,\delta_0,\delta)}{\widetilde{N}_k^r(s,a) + \alpha c_{k,2}(\varepsilon_0,\delta_0,\delta)}$$
$$+ \frac{L(\delta)}{\sqrt{\widetilde{N}_k^r(s,a) + \alpha c_{k,2}(\varepsilon_0,\delta_0,\delta)}}$$

thus proving the first statement of the proposition. Now, we bound the deviation between the private estimate $\widetilde{p}_k$ and the true transition dynamics $p$. First, because $\alpha > 1$, we have that $\sum_{s'} \widetilde{N}_k^p(s,a,s') + \alpha c_{k,3}(\varepsilon_0,\delta_0,\delta) \geq \sum_{s'} N_k^p(s,a,s') + (\alpha-1)c_{k,3}(\varepsilon_0,\delta_0,\delta) > 0$. We start by decomposing the error as

$$\sum_{s'\in\mathcal{S}} |\widetilde{p}(s'|s,a) - p(s'|s,a)| = \sum_{s'\in\mathcal{S}} \left| \frac{\widetilde{N}_k^p(s,a,s')}{\sum_{s'} \widetilde{N}_k^p(s,a,s') + \alpha c_{k,3}(\varepsilon_0,\delta_0,\delta)} - p(s'|s,a) \right|$$

$$\leq \underbrace{\sum_{s'\in\mathcal{S}} \left| \frac{N_k^p(s,a,s')}{\sum_{s'} \widetilde{N}_k^p(s,a,s') + \alpha c_{k,3}(\varepsilon_0,\delta_0,\delta)} - p(s'\mid s,a) \right|}_{①} + \underbrace{\sum_{s'\in\mathcal{S}} \left| \frac{\widetilde{N}_k^p(s,a,s') - N_k^p(s,a,s')}{\sum_{s'} \widetilde{N}_k^p(s,a,s') + \alpha c_{k,3}(\varepsilon_0,\delta_0,\delta)} \right|}_{②}$$

(35)

Recall that $\sum_{s'} \widetilde{N}_k^p(s,a,s') = \widetilde{N}_k^p(s,a)$ and $\sum_{s'} N_k^p(s,a,s') = N_k^p(s,a)$ and define $\overline{p}_k(\cdot|s,a) = \frac{N_k^p(s,a,\cdot)}{N_k^p(s,a)}$. Therefore:

$$① = \sum_{s'\in\mathcal{S}} \left| \frac{N_k^p(s,a,s')}{N_k^p(s,a)} \frac{N_k^p(s,a)}{\widetilde{N}_k^p(s,a) + \alpha c_{k,3}(\varepsilon_0,\delta_0,\delta)} - p(s'\mid s,a) \right|$$

$$= \sum_{s'} \left| \underbrace{\frac{\left(\frac{N_k^p(s,a,s')}{N_k^p(s,a)} - p(s'|s,a)\right) N_k^p(s,a)}{\widetilde{N}_k^p(s,a) + \alpha c_{k,3}(\varepsilon_0,\delta_0,\delta)}}_{>0} + p(s'|s,a) \left( \frac{N_k^p(s,a)}{\widetilde{N}_k^p(s,a) + \alpha c_{k,3}(\varepsilon_0,\delta_0,\delta)} - 1 \right) \right|$$

$$\leq \sum_{s'} \left( p(s'|s,a) \frac{(\alpha+1)c_{k,3}(\varepsilon_0,\delta_0,\delta)}{\widetilde{N}_k^p(s,a) + \alpha c_{k,3}(\varepsilon_0,\delta_0,\delta)} \right) + \frac{N_k^p(s,a)\|\overline{p}_k(\cdot|s,a) - p(\cdot|s,a)\|_1}{\widetilde{N}_k^p(s,a) + \alpha c_{k,3}(\varepsilon_0,\delta_0,\delta)}$$

$$\overset{(a)}{\leq} \frac{(\alpha+1)c_{k,3}(\varepsilon_0,\delta_0,\delta)}{\widetilde{N}_k^p(s,a) + \alpha c_{k,3}(\varepsilon_0,\delta_0,\delta)} + \frac{N_k^p(s,a)}{\widetilde{N}_k^p(s,a) + \alpha c_{k,3}(\varepsilon_0,\delta_0,\delta)} \frac{L(\delta)}{\sqrt{N_k^p(s,a)}}$$

$$\leq \frac{(\alpha+1)c_{k,3}(\varepsilon_0,\delta_0,\delta)}{\widetilde{N}_k^p(s,a) + \alpha c_{k,3}(\varepsilon_0,\delta_0,\delta)} + \frac{L(\delta)}{\sqrt{\widetilde{N}_k^p(s,a) + \alpha c_{k,3}(\varepsilon_0,\delta_0,\delta)}}$$

where $L(\delta) = \sqrt{14S\ln(4\pi^2 SAHk^3/3\delta)}$ and inequality $(a)$ follows from the Weissman inequality [55], and we have again used the fact that the inequalities in Def. 3 hold.

In addition, we have:

$$② \leq \sum_{s'\in\mathcal{S}} \frac{|c_{k,4}(\varepsilon_0,\delta_0,\delta)|}{\widetilde{N}_k^p(s,a) + \alpha c_{k,3}(\varepsilon_0,\delta_0,\delta)} = \frac{S c_{k,4}(\varepsilon_0,\delta_0,\delta)}{\widetilde{N}_k^p(s,a) + \alpha c_{k,3}(\varepsilon_0,\delta_0,\delta)}$$

(36)

Hence putting together Eq. (36) and Eq. (36), we have:

$$\sum_{s'\in\mathcal{S}} \left| \frac{\widetilde{N}_k^p(s,a,s')}{\widetilde{N}_k^p(s,a) + \alpha c_{k,3}(\varepsilon_0,\delta_0,\delta)} - p(s'\mid s,a) \right| \leq \frac{S c_{k,4}(\varepsilon_0,\delta_0,\delta) + (\alpha+1)c_{k,3}(\varepsilon_0,\delta_0,\delta)}{\widetilde{N}_k^p(s,a) + \alpha c_{k,3}(\varepsilon_0,\delta_0,\delta)}$$

$$+ \frac{L(\delta)}{\sqrt{\widetilde{N}_k^p(s,a) + \alpha c_{k,3}(\varepsilon_0,\delta_0,\delta)}}$$

(37)

$\square$

# D   Regret Upper Bound (Proof of Thm. 5)

In this section, we prove Thm 5, which we recall below.

**Theorem.** *For any privacy mechanism $\mathcal{M}$ satisfying Asm. 3 with $\varepsilon > 0$, $\delta_0 \geq 0$, and for any $\delta > 0$ the regret of* LDP-OBI *is bounded with probability at least $1 - \delta$ by:*

$$\Delta(K) \leq \tilde{\mathcal{O}}\Bigg( \underbrace{HS\sqrt{AT}}_{❶} + SAH^2 c_{K,3}\left(\varepsilon,\delta_0,\frac{3\delta}{2\pi^2 K^2}\right) + H^2 S^2 A c_{K,4}\left(\varepsilon,\delta_0,\frac{3\delta}{2\pi^2 K^2}\right)$$

$$+ SAH c_{K,2}\left(\varepsilon,\delta_0,\frac{3\delta}{2\pi^2 K^2}\right) + SAH c_{K,1}\left(\varepsilon,\delta_0,\frac{3\delta}{2\pi^2 K^2}\right) \Bigg)$$

(38)

*The combination of $\mathcal{M}$ and* LDP-OBI *is also $(\varepsilon,\delta_0)$-LDP.*

**Good Event:** Before proceeding the proof of the regret we define a good event under which all concentration inequalities holds with probability at least $1 - \delta$. First, we define the event that all inequalities from Def. 3 holds. Let:

$$L_{1,k} = \bigcap_{s,a} \left\{ \left| \widetilde{R}_k(s,a) - R_k(s,a) \right| \leq c_{k,1}(\varepsilon_0, \delta_0, 3\delta/2k^2\pi^2) \right\}$$

$$L_{2,k} = \bigcap_{s,a} \left\{ \left| \widetilde{N}_k^r(s,a) - N_k^r(s,a) \right| \leq c_{k,2}(\varepsilon_0, \delta_0, 3\delta/2k^2\pi^2) \right\}$$

$$L_{3,k} = \bigcap_{s,a} \left\{ \left| \sum_{s'} N_k^p(s,a,s') - \sum_{s`} \widetilde{N}_k^p(s,a,s') \right| \leq c_{k,3}(\varepsilon_0, \delta_0, 3\delta/2k^2\pi^2) \right\}$$

$$L_{4,k} = \bigcap_{s,a,s'} \left\{ \left| N_k^p(s,a,s') - \widetilde{N}_k^p(s,a,s') \right| \leq c_{k,4}(\varepsilon_0, \delta_0, 3\delta/2k^2\pi^2) \right\}$$

then thanks to Def. 3 we have :

$$\mathbb{P}\left( \bigcup_{k=1}^{+\infty} L_{1,k}^c \cup L_{2,k}^c \cup L_{3,k}^c \cup L_{4,k}^c \right) \leq \sum_{k=1}^{+\infty} \frac{3\delta}{\pi^2 k^2} = \frac{\delta}{4} \tag{39}$$

In addition, for all $k \in \mathbb{N}^\star$, we can define $\bar{r}_k(s,a) = R_k(s,a)/N_k^r(s,a)$ and $\bar{p}_k = N_k^p(s,a,s')/\sum_{s'} N_k^p(s,a,s')$ as the empirical reward and transition probability computed with the non-private counters. Note that in this case $N_k(s,a) := N_k^r(s,a) = \sum_{s'} N_k^p(s,a,s')$. We also define $\overline{\beta}_k^r(\delta,s,a) = \sqrt{\frac{2\ln(1/\delta)}{N_k(s,a)}}$ and $\overline{\beta}_k^p(\delta,s,a) = \sqrt{\frac{14S\log(1/\delta)}{N_k(s,a)}}$. as the size of the confidence intervals using Hoeffding and Weissman inequalities. Thus, we get:

$$\mathbb{P}\left( \bigcup_{k=1}^{+\infty} \bigcup_{s,a} |\bar{r}_k(s,a) - r(s,a)| \geq \overline{\beta}_k^r(3\delta/4\pi^2 SAHk^3, s, a) \right)$$

$$\leq \sum_{k=1}^{+\infty} \sum_{s,a} \mathbb{P}\left( |\bar{r}_k(s,a) - r(s,a)| \geq \sqrt{\frac{2\ln(4\pi^3 SAHk^3/3\delta)}{N_k(s,a)}} \right)$$

$$\leq \sum_{k=1}^{+\infty} \sum_{s,a} \sum_{n=0}^{kH} \mathbb{P}\left( |\bar{r}_k(s,a) - r(s,a)| \geq \sqrt{\frac{2\ln(4\pi^2 SAHk^3/3\delta)}{n}} \right) \leq \sum_{k=1}^{+\infty} \sum_{s,a} \sum_{n=0}^{kH} \frac{3\delta}{4\pi^2 SHAk^3} \leq \frac{\delta}{8}$$

A similar result holds for the transition dynamics, i.e.,:

$$\mathbb{P}\left( \bigcup_{k=1}^{+\infty} \bigcup_{s,a} ||\bar{p}_k(\cdot|s,a) - p(\cdot|s,a)||_1 \geq \overline{\beta}_k^p(3\delta/4\pi^2 SAHk^3, s, a) \right) \leq \frac{\delta}{8} \tag{40}$$

Thus we can define the good event $\mathcal{G}_k$ by:

$$\mathcal{G}_k = \bigcap_{l=1}^{k-1} \bigcap_{i=1}^{4} L_{i,l} \cap \bigcap_{s,a} \left\{ |\bar{r}_l(s,a) - r(s,a)| \leq \overline{\beta}_l^r(3\delta/(4\pi^2 SAHl^3), s, a) \right\}$$

$$\cap \left\{ ||\bar{p}_k(\cdot|s,a) - p(\cdot|s,a)||_1 \leq \overline{\beta}_k^p(3\delta/(4\pi^2 SAHl^3), s, a) \right\}$$

Then $\mathbb{P}\left( \bigcap_{k=1}^{+\infty} \mathcal{G}_k \right) \geq 1 - \delta/2$ and $\mathcal{G}_k \subset \sigma(\mathcal{H}_k)$ (i.e., the history before episode $k$).

**Optimism:** For each episode $k$, the value function $V_{k,1}$ computed by LDP-OBI is optimistic, that is to say: $V_{k,h}(s) \geq V_h^\star(s)$ for any $h$ and state $s$. We sum up this with the following lemma:

**Lemma 6.** *For any episode $k \in [k]$, the value function $V_{k,1}$ computed by running Alg. 2 is such that with probability $1 - \delta$:*

$$\forall s \in \mathcal{S}, h \in [1, H] \qquad V_{k,h}(s) \geq V_h^\star(s) \tag{41}$$

*Proof.* Fix an episode $k$ then we proceed by backward induction conditioned on the event $\mathcal{G}_k$:

- For $h = H$, we have for any state $s$ and action $a$:

$$V_{k,H}(s) \geq Q_{k,H}(s,a) \geq \widetilde{r}_k(s,a) + \beta_k^r(s,a) \geq r(s,a) \text{ thanks to Prop. 4} \qquad (42)$$

- For $h < H$ when the property is true for $h + 1$, we get for any state-action $(s,a)$:

$$V_{k,h}(s) \geq Q_{k,h}(s,a) = \widetilde{r}_k(s,a) + \beta_k^r(s,a) + \widetilde{p}_k(\cdot|s,a)^\mathsf{T} V_{k,h+1} + H\beta_k^p(s,a) \qquad (43)$$
$$\geq r(s,a) + p(\cdot|s,a)^\mathsf{T} V_{k,h+1} \geq Q_h^\star(s,a) \qquad (44)$$

where we used the fact that $\|(\widetilde{p}_k(\cdot|s,a) - p(\cdot|s,a))^\mathsf{T} V_{k,h+1}\| \leq \|\widehat{p}_k(\cdot|s,a) - p(\cdot|s,a)\|_1 \|V_{k,h+1}\|_\infty \leq H\beta_k^p(s,a)$ and the inductive hypothesis.

$\square$

**Regret Decomposition:**  We are now ready to analyze the regret of LDP-OBI. Consider an episode $k$, then, conditioned on $\mathcal{G}_k$:

$$V_1^\star(s_{k,1}) - V_1^{\pi_k}(s_{k,1}) \leq V_{k,1}(s_{k,1}) - V_1^{\pi_k}(s_{k,1}) \leq \widetilde{r}_k(s_{k,1}, a_{k,1}) + \beta_k^r(s_{k,1}, a_{k,1}) - r(s_{k,1}, a_{k,1})$$
$$+ \widetilde{p}_k(\cdot|s,a)^\mathsf{T} V_{k,2} - p(\cdot|s,a)^\mathsf{T} V_2^{\pi_k} + H\beta_k^p(s_{k,1}, a_{k,1})$$

where the last inequality follows from recursively applying the same technique. Then, observe that $(\eta_{k,h})_{k,h}$ is a Martingale Difference Sequence with respect to the history before episode $k$ and thanks to Azuma-Hoeffding inequality we have that with probability at least $1 - \delta/2$, $\sum_{k=1}^K \sum_{h=1}^{H-1} \eta_{k,h} \leq 2H\sqrt{KH \ln(2/\delta)}$. Therefore, we have with probability at least $1 - \delta$:

$$R(\text{LDP-OBI}, K) \leq 2\sum_{k=1}^K \sum_{h=1}^H \beta_k^r(s_{k,h}, a_{k,h}) + H\beta_k^p(s_{k,h}, a_{k,h}) + \underbrace{2H\sqrt{T \ln(2/\delta)}}_{\text{MDS error term}} \qquad (45)$$

Let $\nu_k(s,a) = \sum_{h=1}^H \mathbb{1}_{\{s_{k,h}=s, a_{k,h}=a\}}$. Then summing over the reward bonus and using the fact that $\alpha > 1$, we get:

$$\begin{aligned}
\sum_{k=1}^K \sum_{h=1}^H \beta_k^r(s_{k,h}, a_{k,h}) = &\sum_{s,a,k} \frac{\nu_k(s,a) L_{k,r}}{\sqrt{\widetilde{N}_k^r(s,a) + \alpha c_{k,2}\left(\varepsilon_0, \delta_0, \frac{3\delta}{2\pi^2 k^2}\right)}} \\
&+ \sum_{s,a,k} \frac{\nu_k(s,a)(\alpha+1)c_{k,2}\left(\varepsilon_0, \delta_0, \frac{3\delta}{2\pi^2 k^2}\right)}{\alpha c_{k,2}\left(\varepsilon_0, \delta_0, \frac{3\delta}{2\pi^2 k^2}\right) + \widetilde{N}_k^r(s,a)} \\
&+ \sum_{s,a,k} \frac{\nu_k(s,a) c_{k,1}\left(\varepsilon_0, \delta_0, \frac{3\delta}{2\pi^2 k^2}\right)}{\alpha c_{k,2}\left(\varepsilon_0, \delta_0, \frac{3\delta}{2\pi^2 k^2}\right) + \widetilde{N}_k^r(s,a)}
\end{aligned} \qquad (46)$$

where $L_{k,r} = \sqrt{2\ln\left(\frac{4\pi^2 SAH k^3}{3\delta}\right)}$. Then, using that $\widetilde{N}_k^r(s,a) + c_{k,2}\left(\varepsilon_0, \delta_0, \frac{3\delta}{2\pi^2 k^2}\right) \geq N_k(s,a)$ on the good event from $\mathcal{G}_k$:

$$\begin{aligned}
(46) \leq &\sum_{s,a,k} \frac{\nu_k(s,a) L_{k,r}}{\sqrt{N_k(s,a) + (\alpha-1)c_{k,2}\left(\varepsilon_0, \delta_0, \frac{3\delta}{2\pi^2 k^2}\right)}} + \frac{\nu_k(s,a)(\alpha+1)c_{k,2}\left(\varepsilon_0, \delta_0, \frac{3\delta}{2\pi^2 k^2}\right)}{(\alpha-1)c_{k,2}\left(\varepsilon_0, \delta_0, \frac{3\delta}{2\pi^2 k^2}\right) + N_k(s,a)} \\
&+ \sum_{s,a,k} \frac{\nu_k(s,a) c_{k,1}\left(\varepsilon_0, \delta_0, \frac{3\delta}{2\pi^2 k^2}\right)}{(\alpha-1)c_{k,2}\left(\varepsilon_0, \delta_0, \frac{3\delta}{2\pi^2 k^2}\right) + N_k(s,a)}
\end{aligned} \qquad (47)$$

But because $c_{k,2}$ is non-decreasing in $k$, we have that,

$$\begin{aligned}
(47) \leq &\left((\alpha+1)c_{K,2}\left(\varepsilon_0, \delta_0, \frac{3\delta}{2\pi^2 K^2}\right) + c_{K,1}\left(\varepsilon_0, \delta_0, \frac{3\delta}{2\pi^2 K^2}\right)\right) \sum_{k,s,a} \frac{\nu_k(s,a)}{N_k(s,a)} \\
&+ \sum_{s,a,k} \frac{\nu_k(s,a) L_{K,r}}{\sqrt{N_k(s,a)}}
\end{aligned} \qquad (48)$$

Which can be rewritten as:

$$(48) \leq 2\left((\alpha+1)c_{K,2}\left(\varepsilon_0,\delta_0,\frac{3\delta}{2\pi^2K^2}\right) + c_{K,1}\left(\varepsilon_0,\delta_0,\frac{3\delta}{2\pi^2K^2}\right)\right)SA(\ln(2TSA)+H)$$
$$+\sqrt{6\ln\left(14SAT/\delta\right)}\left(\sqrt{2SAT}+HSA\right) \tag{49}$$

where the last inequality comes from Lem. 19 in [36]. For the sum of the bonus on the transition dynamics we have that:

$$\sum_{k=1}^{K}\sum_{h=1}^{H}H\beta_k^p(s_{k,h},a_{k,h}) = \sum_{s,a,k}\frac{H\nu_k(s,a)L_{k,p}}{\sqrt{\widetilde{N}_k^p(s,a)+\alpha c_{k,3}\left(\varepsilon_0,\delta_0,\frac{3\delta}{2\pi^2k^2}\right)}}$$
$$+\sum_{s,a,k}\frac{HS\nu_k(s,a)c_{k,4}\left(\varepsilon_0,\delta_0,\frac{3\delta}{2\pi^2k^2}\right)}{\alpha c_{k,3}\left(\varepsilon_0,\delta_0,\frac{3\delta}{2\pi^2k^2}\right)+\widetilde{N}_k^p(s,a)} \tag{50}$$
$$+\sum_{s,a,k}\frac{H\nu_k(s,a)(\alpha+1)c_{k,3}\left(\varepsilon_0,\delta_0,\frac{3\delta}{2\pi^2k^2}\right)}{\alpha c_{k,3}\left(\varepsilon_0,\delta_0,\frac{3\delta}{2\pi^2k^2}\right)+\widetilde{N}_k^p(s,a)}$$

where $L_{k,p}=\sqrt{14S\ln\left(\frac{4\pi^2SAHk^3}{3\delta}\right)}$. Then similarly to the reasonning used to bound Eq. (46), we have:

$$(50) \leq \sum_{s,a,k}\frac{H\nu_k(s,a)L_{k,p}}{\sqrt{N_k(s,a)+(\alpha-1)c_{k,3}\left(\varepsilon_0,\delta_0,\frac{3\delta}{2\pi^2k^2}\right)}} + \sum_{s,a,k}\frac{H\nu_k(s,a)(\alpha+1)c_{k,3}\left(\varepsilon_0,\delta_0,\frac{3\delta}{2\pi^2k^2}\right)}{(\alpha-1)c_{k,3}\left(\varepsilon_0,\delta_0,\frac{3\delta}{2\pi^2k^2}\right)+N_k(s,a)}$$
$$+\sum_{k,s,a}\frac{HSc_{k,4}\left(\varepsilon_0,\delta_0,\frac{3\delta}{2\pi^2k^2}\right)}{(\alpha-1)c_{k,3}\left(\varepsilon_0,\delta_0,\frac{3\delta}{2\pi^2k^2}\right)+N_k(s,a)}$$
$$\leq +\left((\alpha+1)c_{K,3}\left(\varepsilon_0,\delta_0,\frac{3\delta}{2\pi^2K^2}\right)+Sc_{K,4}\left(\varepsilon_0,\delta_0,\frac{3\delta}{2\pi^2K^2}\right)\right)\sum_{k,s,a}\frac{H\nu_k(s,a)}{N_k(s,a)}$$
$$\sum_{s,a,k}\frac{H\nu_k(s,a)L_{K,p}}{\sqrt{N_k(s,a)}}$$
$$\leq 2SAH\left((\alpha+1)c_{K,3}\left(\varepsilon_0,\delta_0,\frac{3\delta}{2\pi^2K^2}\right)+Sc_{K,4}\left(\varepsilon_0,\delta_0,\frac{3\delta}{2\pi^2K^2}\right)\right)(\ln(2TSA)+H)$$
$$+H\sqrt{46S\ln\left(14SAT/\delta\right)}\left(\sqrt{2SAT}+HSA\right)$$

where the last inequality comes from [36, Lem. 19] and [56, Lem. 8]. Hence putting everything together, we get that with probability $1-\delta$:

$$R(\text{LDP-OBI},K) \leq H\sqrt{46S\ln(14SAT/\delta)}(\sqrt{2SAT}+HSA)+\sqrt{6\ln(14SAT/\delta)}(\sqrt{2SAT}+HSA)$$
$$+2SAH\left((\alpha+1)c_{K,3}\left(\varepsilon_0,\delta_0,\frac{3\delta}{2\pi^2K^2}\right)+Sc_{K,4}\left(\varepsilon_0,\delta_0,\frac{3\delta}{2\pi^2K^2}\right)\right)(\ln(2TSA)+H)$$
$$+2\left((\alpha+1)c_{K,2}\left(\varepsilon_0,\delta_0,\frac{3\delta}{2\pi^2K^2}\right)+c_{K,1}\left(\varepsilon_0,\delta_0,\frac{3\delta}{2\pi^2K^2}\right)\right)SA(\ln(2TSA)+H)+2H\sqrt{T\ln(2/\delta)}$$

In addition, because LDP-OBI has only access to the privatized data, that is to say it only uses the output of $\mathcal{M}(\{(s_{k,h},a_{k,h},r_{k,h})_{h\leq H}\})$ for each episode $k$, the LDP constraint is satsified as long as the privacy mechanism $\mathcal{M}$ satisfies Def. 1.

**Note:** the proof of this regret upper-bound relies on concentration inequalities more generally used in the average reward regret minimization setting. Stated otherwise, we directly study the error between the estimated model and the true model, i.e., $|\widetilde{r}_k - r|$ and $||\widetilde{p}_k(.\mid s,a)-p(.\mid s,a)||_1$ for each $s,a$. In the non-private setting, it is possible to get a more refined regret using more precise concentration inequalities, mainly Bernstein inequality and other tools introduced in [32]. However, in the private setting, using such results only leads to a gain in lower order terms and terms independent of $\varepsilon$ while the technical derivations are much more intricate.

# E   The Laplace Mechanism for Local Differential Privacy

In this appendix, we show how the well-known Laplace mechanism [4] can be used with LDP-OBI to ensure LDP and a sublinear regret.

---

**Algorithm 3** Laplace mechanism for LDP

---

**Input:** Trajectory: $X = \{(s_h, a_h, r_h) \mid h \leq H\}$, Privacy Parameter: $\varepsilon_0$
Draw $(Y_{i,X}(s,a))_{(s,a)\in\mathcal{S}\times\mathcal{A}, i\leq 2}$ i.i.d $\mathrm{Lap}(1/\varepsilon_0)$ and $(Z_X(s,a,s'))_{(s,a,s')\in\mathcal{S}\times\mathcal{A}\times\mathcal{S}}$ i.i.d $\mathrm{Lap}(1/\varepsilon_0)$ and independent from $Y_{i,X}$ for $i \in \{1, 2\}$
**for** $(s,a) \in \mathcal{S} \times \mathcal{A}$ **do**
$\quad \widetilde{R}_X(s,a) = \sum_{h=1}^{H} r_h \mathbb{1}_{\{s_h, a_h = s, a\}} + Y_{1,X}(s,a)$
$\quad \widetilde{N}_X^r(s,a) = \sum_{h=1}^{H} \mathbb{1}_{\{s_h, a_h = s, a\}} + Y_{2,X}(s,a)$
$\quad$ **for** $s' \in \mathcal{S}$ **do**
$\qquad \widetilde{N}_X^p(s,a,s') = \sum_{h=1}^{H-1} \mathbb{1}_{\{s_h, a_h, s_{h+1} = s, a, s'\}} + Z_X(s,a,s')$
$\quad$ **end for**
**end for**
**Return:** $(\widetilde{R}_X, \widetilde{N}_X^r, \widetilde{N}_X^p) \in \mathbb{R}^{S\times A} \times \mathbb{R}^{S\times A} \times \mathbb{R}^{S\times A\times S}$

---

## E.1   The Laplace mechanism (Alg. 3) satisfies local differential privacy (Asm. 3)

We first prove Thm. 7 which states that using Alg. 3 with parameter $\varepsilon_0 = \varepsilon/6H$ guarantees $(\varepsilon, \delta)$-LDP.

**Theorem 7.** *For any $\varepsilon > 0$, the Laplace mechanism described by Alg. 3 with parameter $\varepsilon_0 = \varepsilon/6H$ is $(\varepsilon, 0)$-LDP (and thus $(\varepsilon, \delta_0)$-LDP for every $\delta_0 \geq 0$).*

Formally, we need to show that, for any two trajectories $X$ and $X'$ and tuple $(r, n, n')$, the following inequality holds

$$\mathbb{P}\Big(\mathcal{M}(X) = (r, n, n')\Big) \leq e^\varepsilon \mathbb{P}\Big(\mathcal{M}(X') = (r, n, n')\Big) + \delta \tag{51}$$

where $r$, $n$, $n'$ are vectors of dimension $SA$, $SA$ and $S^2A$, respectively. See the LDP definition in Def. 1.

*Proof of Thm. 7.* Let's consider two trajectories $X = \{(s_h, a_h, r_h) \mid h \leq H\}$ and $X' = \{(s'_h, a'_h, r'_h) \mid h \leq H\}$. We denote the output of the private randomizer $\mathcal{M}$ by $\mathcal{M}(X) = (\widetilde{R}_X, \widetilde{N}_X^r, \widetilde{N}_X^p)$ and $\mathcal{M}(X') = (\widetilde{R}_{X'}, \widetilde{N}_{X'}^r, \widetilde{N}_{X'}^p)$. Recall that $\widetilde{R}_X(s,a) := \sum_{h=1}^{H} r_h \mathbb{1}_{\{s_h = s, a_h = a\}} + Y_{1,X}(s,a)$ where $(Y_{1,X}(s,a))_{(s,a)\in\mathcal{S}\times\mathcal{A}}$ are independent Laplace variables with parameter $\varepsilon/(6H)$. Consider a vector $r \in \mathbb{R}^{S\times A}$, then:

$$\frac{\mathbb{P}\left(\forall(s,a), \widetilde{R}_X(s,a) = r_{s,a} \mid X\right)}{\mathbb{P}\left(\forall(s,a), \widetilde{R}_{X'}(s,a) = r_{s,a} \mid X'\right)} = \prod_{s,a} \frac{\mathbb{P}\left(Y_{1,X}(s,a) = \sum_{h=1}^{H} r_h \mathbb{1}_{\{s_h = s, a_h = a\}} - r_{s,a} \mid X\right)}{\mathbb{P}\left(Y_{1,X'}(s,a) = \sum_{h=1}^{H} r'_h \mathbb{1}_{\{s'_h = s, a'_h = a\}} - r_{s,a} \mid X'\right)} \tag{52}$$

since the Laplace distribution is symmetric. But $Y_{1,X}(s,a)$ and $Y_{1,X'}(s,a)$ are independent random variables for any state-action pair. Thus:

$$\prod_{s,a} \frac{\mathbb{P}\left(Y_{1,X}(s,a) = \sum_{h=1}^{H} r_h \mathbb{1}_{\left\{\substack{s_h=s,\\a_h=a}\right\}} - r_{s,a} \mid X\right)}{\mathbb{P}\left(Y_{1,X'}(s,a) = \sum_{h=1}^{H} r'_h \mathbb{1}_{\left\{\substack{s'_h=s,\\a'_h=a}\right\}} - r_{s,a} \mid X'\right)} = \prod_{s,a} \frac{e^{\left(\varepsilon_0 \left|\sum_{h=1}^{H}(r_h \mathbb{1}_{\left\{\substack{s_h=s,\\a_h=a}\right\}}) - r_{s,a}\right|\right)}}{e^{\left(\varepsilon_0 \left|\sum_{h=1}^{H}(r'_h \mathbb{1}_{\left\{\substack{s'_h=s,\\a'_h=a}\right\}}) - r_{s,a}\right|\right)}}$$

$$\leq \exp\left(\varepsilon_0 \sum_{s,a} \left|\sum_{h=1}^{H}(r_h \mathbb{1}_{\{s_h=s,a_h=a\}} - r'_h \mathbb{1}_{\{s'_h=s,a'_h=a\}})\right|\right) \tag{53}$$

$$\leq \exp\left(\varepsilon_0 \sum_{s,a,h}(|r_h|\mathbb{1}_{\{s_h=s,a_h=a\}} + |r'_h|\mathbb{1}_{\{s'_h=s,a'_h=a\}})\right)$$

$$= \exp\left(\varepsilon_0 \sum_{h}(|r_h| + |r'_h|)\right) \leq \exp\left(2H\varepsilon_0\right) = \exp\left(\frac{\varepsilon}{3}\right)$$

where we used the definition of the Laplace distribution, $x \mapsto \frac{1}{2b}\exp(|x|/b)$. Let $n \in \mathbb{R}^{S \times A}$ and $n' \in \mathbb{R}^{S \times A \times S}$. Similarly, since $\widetilde{N}_X^r(s,a) = \sum_{h=1}^{H} \mathbb{1}_{\{s_h=s,a_h=a\}} + Y_{2,X}(s,a)$ and $\widetilde{N}_X^p(s,a,s') = \sum_{h=1}^{H-1} \mathbb{1}_{\{s_h=s,a_h=a,s_{h+1}=s'\}} + Z_X(s,a,s')$, we have:

$$\frac{\mathbb{P}\left(\forall(s,a), \widetilde{N}_X^r(s,a) = n_{s,a} \mid X\right)}{\mathbb{P}\left(\forall(s,a), \widetilde{N}_{X'}^r(s,a) = n_{s,a} \mid X'\right)} \leq \exp\left(\frac{\varepsilon}{3}\right) \tag{54}$$

and:

$$\frac{\mathbb{P}\left(\forall(s,a,s'), \widetilde{N}_X^p(s,a,s') = n'_{s,a,s'} \mid X\right)}{\mathbb{P}\left(\forall(s,a,s'), \widetilde{N}_{X'}^p(s,a,s') = n'_{s,a,s'} \mid X'\right)} \leq \exp\left(\frac{\varepsilon}{3}\right) \tag{55}$$

Then because $(Y_{i,X}(s,a))_{i \leq 2,(s,a) \in \mathcal{S} \times \mathcal{A}}$, $(Z_X(s,a,s'))_{(s,a,s') \in \mathcal{S} \times \mathcal{A} \times \mathcal{S}}$ are independent it holds that:

$$\mathbb{P}\left(\widetilde{R}_X = r, \widetilde{N}_X^r = n, \widetilde{N}_X^p = n' \mid X\right) = \mathbb{P}\left(\widetilde{R}_X = r \mid X\right)\mathbb{P}\left(\widetilde{N}_X^r = n \mid X\right)\mathbb{P}\left(\widetilde{N}_X^p = n' \mid X\right)$$

Thus for any $(r,n,n') \in \mathbb{R}^{S \times A} \times \mathbb{R}^{S \times A} \times \mathbb{R}^{S \times A \times S}$ and any two trajectories $X$ and $X'$:

$$\mathbb{P}\Big(\mathcal{M}(X) = (r,n,n') \mid X\Big) = \mathbb{P}\left(\widetilde{R}_X = r, \widetilde{N}_X^r = n, \widetilde{N}_X^p = n' \mid X\right)$$

$$= \mathbb{P}\left(\widetilde{R}_X = r \mid X\right)\mathbb{P}\left(\widetilde{N}_X^r = n \mid X\right)\mathbb{P}\left(\widetilde{N}_X^p = n' \mid X\right)$$

where we use the convention that $\widetilde{R}_X = r$ implies that $\widetilde{R}_X(s,a) = r_{x,a}$, and similarly for $\widetilde{N}_X^r = n, \widetilde{N}_X^p = n'$. Therefore using inequalities (53), (54) and (55) in (**??**), we have:

$$\mathbb{P}\Big(\mathcal{M}(X) = (r,n,n') \mid X\Big) = \mathbb{P}\left(\widetilde{R}_X = r \mid X\right)\mathbb{P}\left(\widetilde{N}_X^r = n \mid X\right)\mathbb{P}\left(\widetilde{N}_X^p = n' \mid X\right)$$

$$\leq \exp(\varepsilon)\mathbb{P}\left(\widetilde{R}_{X'} = r \mid X'\right)\mathbb{P}\left(\widetilde{N}_{X'}^r = n \mid X'\right)\mathbb{P}\left(\widetilde{N}_{X'}^p = n' \mid X'\right)$$

$$= \exp(\varepsilon)\mathbb{P}\left(\widetilde{R}_{X'} = r, \widetilde{N}_{X'}^r = n, \widetilde{N}_{X'}^p = n' \mid X'\right)$$

$$= \exp(\varepsilon)\mathbb{P}\left(\mathcal{M}(X') = (r,n,n') \mid X'\right)$$

This concludes the proof. □

Now that we shown the Laplace mechanism ensures LDP with the reight parameter, let's show that the latter satisfies Asm. 3 by showing the following proposition:

**Proposition 8.** *For any $\varepsilon > 0$, the Laplace mechnism, Alg. 3, with parameter $\varepsilon_0 = \varepsilon/(6H)$ satisfies Def. 3 for any $\delta > 0$ and $k \in \mathbb{N}$ with $c_{k,1}(\varepsilon, \delta) = c_{k,2}(\varepsilon, \delta)$, $c_{k,3}(\varepsilon, \delta) = \sqrt{S}c_{k,4}(\varepsilon, \delta)$ and:*

$$c_{k,1}(\varepsilon, \delta) = \max\left\{\sqrt{k}, \ln\left(\frac{6SA}{\delta}\right)\right\} \frac{\sqrt{8\ln\left(\frac{6SA}{\delta}\right)}}{\varepsilon/6H},$$

$$c_{k,3}(\varepsilon, \delta) = \max\left\{\sqrt{kS}, \ln\left(\frac{6S^2A}{\delta}\right)\right\} \frac{\sqrt{8\ln\left(\frac{6S^2A}{\delta}\right)}}{\varepsilon/6H}$$

Before proving Prop. 8 we state the following concentration inequality for the sum of Laplace variables.

**Proposition 9.** *[14, Cor. 12.3] Let $Y_1, \ldots, Y_k$ be independent Lap(b) random variables with $b > 0$ and $\delta \in (0, 1)$ then for any $\nu > b \max\left\{\sqrt{k}, \sqrt{\ln(2/\delta)}\right\}$,*

$$\mathbb{P}\left(\left|\sum_{l=1}^{k} Y_l\right| > \nu\sqrt{8\ln(2/\delta)}\right) \leq \delta$$

We can now prove Prop. 8 that shows that Alg. 3 satisfies Def. 3.

*Proof of Prop. 8.* Let $X_1, \ldots, X_{k-1}$ be the $k-1$ trajectories generated before episode $k \geq 1$. Consider the private statistic $\widetilde{R}_k(s, a)$ generated by the private randomizer before episode $k$. Then for any state-action pair $(s, a) \in \mathcal{S} \times \mathcal{A}$:

$$\left|\widetilde{R}_k(s, a) - R_k(s, a)\right| = \left|\sum_{l < k}(\widetilde{R}_{X_l}(s, a) - R_{X_l}(s, a))\right|$$

$$= \left|\sum_{l < k}\left(Y_{1, X_l}(s, a) + \sum_{h=1}^{H} r_h \mathbb{1}_{\left\{\substack{s_{l,h}=s, \\ a_{l,h}=a}\right\}}\right) - \sum_{l < k}\sum_{h=1}^{H} r_h \mathbb{1}_{\left\{\substack{s_{l,h}=s, \\ a_{l,h}=a}\right\}}\right|$$

$$= \left|\sum_{l=1}^{k-1} Y_{1, X_l}(s, a)\right|$$

which is the sum of independent Laplace variables. Let $\delta > 0$. By Prop. 9 we have that with probability at least $1 - \delta/(3SA)$

$$\left|\sum_{l=1}^{k-1} Y_{1, X_l}(s, a)\right| \leq \frac{1}{\varepsilon_0} \max\left\{\sqrt{k-1}, \ln\left(\frac{6SA}{\delta}\right)\right\} \sqrt{8\ln\left(\frac{6SA}{\delta}\right)} \tag{56}$$

The same property holds for $\widetilde{N}_k^r$ and $\widetilde{N}_k^p$ and we again apply Prop. 9. Properties in Def. 3 follow from union bounds. $\square$

# F  Other Privacy Preserving Mechanisms

We have shown in App. E.1 that the Laplace mechanism, Alg. 3, satisfies Def. 3. However it is not the only mechanism to do so. In this appendix we present the Gaussian, Randomized Response and bounded noise mechanisms and show that these also satisfy Def. 3.

## F.1  Gaussian Mechanism:

The Gaussian mechanism is a fundamental mechanism in the differential privacy literature [see e.g., 14]. However, contrary to the Laplace mechanism the Gaussian mechanism can only guarantees $(\varepsilon, \delta)$-LDP for $\delta > 0$. The mechanism is based on the same idea as the Laplace mechanism, that is to say it adds Gaussian noise to the result of a given computation on the input data. This noise is centered and the standard deviation $\sigma(\varepsilon, \delta)$ is $\frac{cH}{\epsilon_0}$.

---
**Algorithm 4** Gaussian mechanism for LDP
---

**Input:** Trajectory: $X = \{(s_h, a_h, r_h) \mid h \leq H\}$, Privacy Parameter: $\varepsilon_0, c$
Draw $(Y_{i,X}(s,a))_{(s,a)\in\mathcal{S}\times\mathcal{A}, i\leq 2}$ i.i.d $\mathcal{N}\left(0, \sigma^2\right)$ and $(Z_X(s,a,s'))_{(s,a,s')\in\mathcal{S}\times\mathcal{A}\times\mathcal{S}}$ i.i.d $\mathcal{N}\left(0, \sigma^2\right)$
and independent from $Y_{i,X}$ for $i \in \{1, 2\}$ with $\sigma = cH/\varepsilon_0$
**for** $(s, a) \in \mathcal{S} \times \mathcal{A}$ **do**
   $\widetilde{R}_X(s,a) = \sum_{h=1}^{H} r_h \mathbb{1}_{\{s_h=s, a_h=a\}} + Y_{1,X}(s,a)$
   $\widetilde{N}_X^r(s,a) = \sum_{h=1}^{H} \mathbb{1}_{\{s_h=s, a_h=a\}} + Y_{2,X}(s,a)$
   **for** $s' \in \mathcal{S}$ **do**
      $\widetilde{N}_X^p(s,a,s') = \sum_{h=1}^{H-1} \mathbb{1}_{\{s_h=s, a_h=a, s_{h+1}=s'\}} + Z_X(s,a,s')$
   **end for**
**end for**
**Return:** $(\widetilde{R}_X, \widetilde{N}_X^r, \widetilde{N}_X^p) \in \mathbb{R}^{S\times A} \times \mathbb{R}^{S\times A} \times \mathbb{R}^{S\times A\times S}$

---

In the following, we show that the Gaussian mechanism almost satisfies Def. 3. The Gaussian mechanism can not guarantee $(\varepsilon_0, 0)$-LDP for any $\varepsilon_0 > 0$, however we show that it satisfies the other necessary conditions, including $(\varepsilon_0, \delta)$-LDP for any $\delta > 0$. First, we show that the mechanism guarantees Local Differential Privacy for high enough noise.

**Proposition 10.** *For any* $1 \geq \varepsilon_0 > 0$ *and* $\delta_0 > 0$ *and parameter* $c > 4\ln\left(\frac{24}{\delta_0}\right)$, *the Gaussian mechanism, Alg. 4, is* $(\varepsilon_0, \delta_0)$-*LDP.*

*Proof of Prop. 10:* The proof is based on the proof presented in [14]. Similarly to the proof of Prop. 8 let's consider two trajectories $X = \{(s_h, a_h, r_h) \mid h \leq H\}$ and $X' = \{(s'_h, a'_h, r'_h) \mid h \leq H\}$ and also denote the output of the private randomizer $\mathcal{M}$ by $\mathcal{M}(X) = (\widetilde{R}_X, \widetilde{N}_X^r, \widetilde{N}_X^p)$ and $\mathcal{M}(X') = (\widetilde{R}_{X'}, \widetilde{N}_{X'}^r, \widetilde{N}_{X'}^p)$.

For a given vector $r \in \mathbb{R}^{S\times A}$,

$$\frac{\mathbb{P}\left(\forall(s,a), \widetilde{R}_X(s,a) = r_{s,a} \mid X\right)}{\mathbb{P}\left(\forall(s,a), \widetilde{R}_{X'}(s,a) = r_{s,a} \mid X'\right)} = \prod_{s,a} \frac{\mathbb{P}\left(Y_{1,X}(s,a) = \sum_{h=1}^{H} r_h \mathbb{1}_{\{s_h=s, a_h=a\}} - r_{s,a} \mid X\right)}{\mathbb{P}\left(Y_{1,X'}(s,a) = \sum_{h=1}^{H} r'_h \mathbb{1}_{\{s'_h=s, a'_h=a\}} - r_{s,a} \mid X'\right)} \quad (57)$$

since the Gaussian distribution is symmetric. Then,

$$\prod_{s,a} \frac{\mathbb{P}\left(Y_{1,X}(s,a) = \sum_{h=1}^{H} r_h \mathbb{1}_{\{s_h=s, a_h=a\}} - r_{s,a} \mid X\right)}{\mathbb{P}\left(Y_{1,X'}(s,a) = \sum_{h=1}^{H} r'_h \mathbb{1}_{\{s'_h=s, a'_h=a\}} - r_{s,a} \mid X'\right)}$$
$$= \prod_{s,a} \exp\left(\frac{\left(\sum_{h=1}^{H} r_h \mathbb{1}_{\{s_h=s, a_h=a\}} - r_{s,a}\right)^2 - \left(\sum_{h=1}^{H} r'_h \mathbb{1}_{\{s'_h=s, a'_h=a\}} - r_{s,a}\right)^2}{2\sigma^2}\right) \quad (58)$$

But, considering the squared term, we get

$$\left(\sum_{h=1}^{H} r_h \mathbb{1}_{\left\{\substack{s_h=s,\\a_h=a}\right\}} - r_{s,a}\right)^2 = \left(\sum_{h=1}^{H} r_h \mathbb{1}_{\left\{\substack{s_h=s,\\a_h=a}\right\}} - \sum_{h=1}^{H} r'_h \mathbb{1}_{\left\{\substack{s'_h=s,\\a'_h=a}\right\}} + \sum_{h=1}^{H} r'_h \mathbb{1}_{\left\{\substack{s'_h=s,\\a'_h=a}\right\}} - r_{s,a}\right)^2$$

$$= \left(\sum_{h=1}^{H} r_h \mathbb{1}_{\left\{\substack{s_h=s,\\a_h=a}\right\}} - \sum_{h=1}^{H} r'_h \mathbb{1}_{\left\{\substack{s'_h=s,\\a'_h=a}\right\}}\right)^2 + \left(\sum_{h=1}^{H} r'_h \mathbb{1}_{\left\{\substack{s'_h=s,\\a'_h=a}\right\}} - r_{s,a}\right)^2$$

$$+ 2\left(\sum_{h=1}^{H} r_h \mathbb{1}_{\left\{\substack{s_h=s,\\a_h=a}\right\}} - \sum_{h=1}^{H} r'_h \mathbb{1}_{\left\{\substack{s'_h=s,\\a'_h=a}\right\}}\right)\left(\sum_{h=1}^{H} r'_h \mathbb{1}_{\left\{\substack{s'_h=s,\\a'_h=a}\right\}} - r_{s,a}\right)$$

Hence we get that

$$
(58) = \prod_{s,a} \exp\left( \frac{1}{2\sigma^2}\left( \left( \sum_{h=1}^{H} r_h \mathbb{1}_{\left\{ \substack{s_h=s,\\ a_h=a} \right\}} - \sum_{h=1}^{H} r'_h \mathbb{1}_{\left\{ \substack{s'_h=s,\\ a'_h=a} \right\}} \right)^2 \right.\right.
$$
$$
\left.\left. - 2\left( \sum_{h=1}^{H} r_h \mathbb{1}_{\left\{ \substack{s_h=s,\\ a_h=a} \right\}} - r'_h \mathbb{1}_{\left\{ \substack{s'_h=s,\\ a'_h=a} \right\}} \right)\left( \sum_{h=1}^{H} r'_h \mathbb{1}_{\left\{ \substack{s'_h=s,\\ a'_h=a} \right\}} - r_{s,a} \right) \right) \right). \tag{59}
$$

But, $\sum_{s,a}\left( \sum_{h=1}^{H} r_h \mathbb{1}_{\{s_h=s,a_h=a\}} - \sum_{h=1}^{H} r'_h \mathbb{1}_{\{s'_h=s,a'_h=a\}} \right)^2 \leq 2H^2$ because for each step $h$, $r_h \in [0,1]$. By the same reasonning, we have $\sum_{s,a}\left| \left( \sum_{h=1}^{H} r_h \mathbb{1}_{\{s_h=s,a_h=a\}} - r'_h \mathbb{1}_{\{s'_h=s,a'_h=a\}} \right) \sum_{h=1}^{H} r'_h \mathbb{1}_{\{s'_h=s,a'_h=a\}} \right| \leq H^2$. Therefore, we have:

$$
(58) \leq \exp\left( \frac{1}{2\sigma^2}\left( 2\sum_{s,a}\left( \sum_{h=1}^{H} r_h \mathbb{1}_{\{s_h=s,a_h=a\}} - r'_h \mathbb{1}_{\{s'_h=s,a'_h=a\}} \right) r_{s,a} + 3H^2 \right) \right)
$$
$$
\leq \exp\left( \frac{1}{2\sigma^2}\left( 2\sqrt{2}H\sqrt{\sum_{s,a} r_{s,a}^2} + 3H^2 \right) \right) \tag{60}
$$

where the last inequality follows from Cauchy-Schwartz. Note that if $||r||_2 \leq \frac{\sigma^2 \varepsilon_0}{3\sqrt{2}H} - \frac{3H}{2\sqrt{2}}$, Eq. (60) is bounded by $\exp(\varepsilon_0/3)$. Therefore, to finish, we partition $\mathbb{R}^{S\times A}$ in two subspaces $R_1 = \left\{ x \in \mathbb{R}^{S\times A} \mid ||x||_2 \leq \frac{c^2 H}{3\sqrt{2}\varepsilon_0} - \frac{3H}{2\sqrt{2}} \right\}$ and $R_2 = \left\{ x \in \mathbb{R}^{S\times A} \mid ||x||_2 > \frac{c^2 H}{3\sqrt{2}\varepsilon_0} - \frac{3H}{2\sqrt{2}} \right\}$ where we used the fact that $\sigma = cH/\varepsilon_0$ with $c$ a constant to be chosen later. Then for $c^2 \geq 4\ln\left( \frac{3}{\delta_1} \right)$, for $\delta_1$ to be chosen later, $\mathbb{P}\left( Y_{1,X} \in R_2 \right) \leq \delta_1$ and $\mathbb{P}\left( Y_{1,X'} \in R_2 \right) \leq \delta_1$. Thus for Eq. (57):

$$
\mathbb{P}\left( \forall (s,a), \widetilde{R}_X(s,a) = r_{s,a} \mid X \right) = \mathbb{P}\left( \forall (s,a), \widetilde{R}_X(s,a) = r_{s,a} \mid X \right) \mathbb{1}_{\{r - (\sum_{h=1}^{H} r_h \mathbb{1}_{\{\substack{s_h=s,\\a_h=a}\}})_{s,a} \in R_1\}} \tag{61}
$$

$$
+ \mathbb{P}\left( \forall (s,a), \widetilde{R}_X(s,a) = r_{s,a} \mid X \right) \mathbb{1}_{\{r - (\sum_{h=1}^{H} r_h \mathbb{1}_{\{\substack{s_h=s,\\a_h=a}\}})_{s,a} \in R_2\}}
$$
$$
\leq e^{\frac{\varepsilon_0}{3}} \mathbb{P}\left( \forall (s,a), \widetilde{R}_{X'}(s,a) = r_{s,a} \mid X' \right) \mathbb{1}_{\{r - (\sum_{h=1}^{H} r_h \mathbb{1}_{\{\substack{s_h=s,\\a_h=a}\}})_{s,a} \in R_1\}} \tag{62}
$$
$$
+ \mathbb{P}\left( Y_{1,X} \in R_2 \right)
$$
$$
\leq \exp(\varepsilon_0/3)\mathbb{P}\left( \forall (s,a), \widetilde{R}_{X'}(s,a) = r_{s,a} \mid X' \right) + \delta_1 \tag{63}
$$

We get the same results for $\widetilde{N}^r$ and $\widetilde{N}^p$. Then, because $(Y_{i,X}(s,a))_{i\leq 2, (s,a)\in \mathcal{S}\times\mathcal{A}}$, $(Z_X(s,a,s'))_{(s,a,s')\in \mathcal{S}\times\mathcal{A}\times\mathcal{S}}$ are independent, see Alg. 4 it holds that:

$$
\mathbb{P}\left( \widetilde{R}_X = r, \widetilde{N}_X^r = n, \widetilde{N}_X^p = n' \mid X \right) = \mathbb{P}\left( \widetilde{R}_X = r \mid X \right)\mathbb{P}\left( \widetilde{N}_X^r = n \mid X \right)\mathbb{P}\left( \widetilde{N}_X^p = n' \mid X \right)
$$

and so,

$$
\mathbb{P}\left( \mathcal{M}(X) = (r,n,n') \mid X \right) = \mathbb{P}\left( \widetilde{R}_X = r, \widetilde{N}_X^r = n, \widetilde{N}_X^p = n' \mid X \right)
$$
$$
= \mathbb{P}\left( \widetilde{R}_X = r \mid X \right)\mathbb{P}\left( \widetilde{N}_X^r = n \mid X \right)\mathbb{P}\left( \widetilde{N}_X^p = n' \mid X \right)
$$

Then for any two trajectories $X$ and $X'$, we have:

$$
\mathbb{P}\left( \widetilde{R}_X = r \mid X \right)\mathbb{P}\left( \widetilde{N}_X^r = n \mid X \right)\mathbb{P}\left( \widetilde{N}_X^p = n' \mid X \right) \leq \left( e^{\frac{\varepsilon_0}{3}}\mathbb{P}\left( \widetilde{R}_{X'} = r \mid X' \right) + \delta_1 \right)
$$
$$
\times \left( e^{\frac{\varepsilon_0}{3}}\mathbb{P}\left( \widetilde{N}_{X'}^r = n \mid X' \right) + \delta_1 \right)
$$
$$
\times \left( e^{\frac{\varepsilon_0}{3}}\mathbb{P}\left( \widetilde{N}_{X'}^p = n' \mid X' \right) + \delta_1 \right)
$$
$$
\leq e^{\varepsilon_0}\mathbb{P}\left( \widetilde{R}_{X'} = r \mid X' \right)\mathbb{P}\left( \widetilde{N}_{X'}^r = n \mid X' \right)\mathbb{P}\left( \widetilde{N}_{X'}^p = n' \mid X' \right) + 2\delta_1\exp\left( 2\varepsilon_0/3 \right)
$$
$$
+ 2\delta_1^2\exp\left( \varepsilon_0/3 \right) + \delta_1^3
$$

Thus by choosing $\delta_1 = \delta_0/8$, it holds that $2\delta_1 \exp(2\varepsilon_0/3) + 2\delta_1^2 \exp(\varepsilon_0/3) + \delta_1^3 \leq \delta_0$ for $\varepsilon_0 \leq 1$, and so we can conclude that the Gaussian mechanism is $(\varepsilon_0, \delta_0)$-LDP. $\qquad\square$

In addition, the precision of the Gaussian mechanism is of the same order as the Laplace mechanism, that is to say:

**Proposition 11.** *The Gaussian mechanism, Alg. 4, with parameter $\varepsilon_0 > 0$ and $c^2 \geq 4\ln\left(\frac{24}{\delta_0}\right)$ for any $\delta_0 > 0$ satisfies Def. 3 for any $\delta > 0$ and $k \in \mathbb{N}^\star$ with:*

$$c_{k,1}(\varepsilon_0, \delta_0, \delta) = c_{k,2}(\varepsilon_0, \delta_0, \delta) = c_{k,4}(\varepsilon_0, \delta_0, \delta) = \max\left\{\frac{cH}{\varepsilon_0}\sqrt{(k-1)\ln\left(\frac{6SA}{\delta}\right)}, 1\right\}$$

$$c_{k,3}(\varepsilon_0, \delta_0, \delta) = \max\left\{\frac{cH}{\varepsilon_0}\sqrt{(k-1)S\ln\left(\frac{6SA}{\delta}\right)}, 1\right\}$$

This result shows that using the Gaussian mechanism rather than the Laplace mechanism would not lead to improved regret rate as the utilities $c_{k,1}, c_{k,2}, c_{k,3}, c_{k,4}$ have the same depency of $S, A, H, \varepsilon_0$ and $k$. Moreover, the Gaussian mechanism only guarantees LDP for $\delta > 0$ whereas using the Laplace mechanism ensures that we can guarantee LDP for $\delta = 0$ as well.

*Proof of Prop. 11:* Following the same steps as in the proof of Prop 8, we have that at the beginning of episode $k$ with probability at least $1 - \frac{\delta}{3SA}$:

$$\left|\widetilde{R}_k(s,a) - R_k(s,a)\right| = \left|\sum_{l<k}(\widetilde{R}_{X_l}(s,a) - R_{X_l}(s,a))\right| \qquad (64)$$

$$= \left|\sum_{l<k}\left(Y_{1,X_l}(s,a) + \sum_{h=1}^H r_h \mathbb{1}_{\left\{\substack{s_{l,h}=s,\\a_{l,h}=a}\right\}}\right) - \sum_{l<k}\sum_{h=1}^H r_h \mathbb{1}_{\left\{\substack{s_{l,h}=s,\\a_{l,h}=a}\right\}}\right| \qquad (65)$$

$$= \left|\sum_{l=1}^{k-1} Y_{1,X_l}(s,a)\right| \leq \sigma\sqrt{2(k-1)\ln\left(\frac{6SA}{\delta}\right)} \qquad (66)$$

for $\sigma = cH/\varepsilon_0$ thanks to Chernoff bounds. The same result follows for $\widetilde{N}^r$ and $\widetilde{N}^p$. Therefore, the Gaussian mechanism satisfies Def. 3 with $c_{k,1}(\varepsilon_0, \delta_0, \delta) = c_{k,2}(\varepsilon_0, \delta_0, \delta) = c_{k,4}(\varepsilon_0, \delta_0, \delta)$ with:

$$c_{k,1}(\varepsilon_0, \delta_0, \delta) = \max\left\{\frac{cH}{\varepsilon_0}\sqrt{(k-1)\ln\left(\frac{6SA}{\delta}\right)}, 1\right\} \qquad (67)$$

with $c > 0$ and:

$$c_{k,3}(\varepsilon_0, \delta_0, \delta) = \max\left\{\frac{cH}{\varepsilon_0}\sqrt{(k-1)S\ln\left(\frac{6SA}{\delta}\right)}, 1\right\} \qquad (68)$$

where $c_{k,3}(\varepsilon_0, \delta_0, \delta)$ is defined such that $\left|\sum_{s'} N_k^p(s,a,s') - \sum_{s'} \widetilde{N}_k^p(s,a,s')\right| \leq c_{k,3}(\varepsilon_0, \delta_0, \delta)$. $\qquad\square$

### F.2 Randomized Response Mechanism:

The second alternative mechanism we consider is the Randomized Response mechanism. In general, it is used for discrete data like indicator functions $(\mathbb{1}_{\{s_h=s,a_h=a\}})_{h,s,a}$. We therefore use it to privatize the number of visits of a state-action pair and state-action-next-state tuple for each trajectory. With the assumption that reward are supported in $[0,1]$, we can also use this mechanism for privatizing the cumulative reward of a given trajectory. Contrary to previous ones, the output of the Randomized Response mechanism is three vectors, two of size $H \times S \times A$, and the last one of size $(H-1) \times S \times A \times S$. We slightly modify the requirements of Def. 3 by changing the size of the output of the privacy preserving mechanism. We summarize the mechanism in Alg. 5.

Just as for the Gaussian mechanism, we show that Alg. 5 satisfies Def. 3. We begin by showing that this mechanism satisfies $(\varepsilon_0, 0)$-LDP for any $\varepsilon_0 > 0$.

---

**Algorithm 5** Randomized Response mechanism for LDP

---

**Input:** Trajectory: $X = \{(s_h, a_h, r_h) \mid h \leq H\}$, Privacy Parameter: $\varepsilon_0$
Draw $(Y_{i,X}(s,a))_{(s,a) \in \mathcal{S} \times \mathcal{A}, i \leq 2}$ i.i.d $\mathcal{N}\left(0, \sigma^2\right)$ and $(Z_X(s, a, s'))_{(s,a,s') \in \mathcal{S} \times \mathcal{A} \times \mathcal{S}}$ i.i.d $\mathcal{N}\left(0, \sigma^2\right)$
and independent from $Y_{i,X}$ for $i \in \{1, 2\}$ with $\sigma = cH/\varepsilon_0$
**for** $(s,a) \in \mathcal{S} \times \mathcal{A}$ **do**
   **for** $h = 1, \ldots, H$ **do**
     Sample $Y_{1,X}(h, s, a) \sim \text{Ber}\left(\frac{e^{\varepsilon_0}-1}{e^{\varepsilon_0}+1} r_h \mathbb{1}_{\{s_h=s, a_h=a\}} + \frac{1}{e^{\varepsilon_0}+1}\right)$
     $\widetilde{R}_X(h, s, a) = \frac{e^{\varepsilon_0}+1}{e^{\varepsilon_0}-1}\left(Y_{1,X}(h, s, a) - \frac{1}{e^{\varepsilon_0}+1}\right)$
     Sample $\widetilde{n}^r_X(h, s, a) \sim \text{Ber}\left(\frac{e^{\varepsilon_0}-1}{e^{\varepsilon_0}+1} \mathbb{1}_{\{s_h=s, a_h=a\}} + \frac{1}{e^{\varepsilon_0}+1}\right)$
     **if** $h < H$ **then**
       **for** $s' \in \mathcal{S}$ **do**
         Sample $\widetilde{n}^p_X(h, s, a, s') \sim \text{Ber}\left(\frac{e^{\varepsilon_0}-1}{e^{\varepsilon_0}+1} \mathbb{1}_{\{s_h=s, a_h=a, s_{h+1}=s'\}} + \frac{1}{e^{\varepsilon_0}+1}\right)$
         $\widetilde{N}^p_X(h, s, a, s') = \frac{e^{\varepsilon_0}+1}{e^{\varepsilon_0}-1}\left(\widetilde{n}^p_X(h, s, a, s') - \frac{1}{e^{\varepsilon_0}+1}\right)$
       **end for**
     **end if**
   **end for**
**end for**
**Return:** $\qquad (\widetilde{R}_X, \widetilde{N}^r_X, \widetilde{N}^p_X) \qquad \in \qquad \left\{\frac{-1}{e^{\varepsilon_0}-1}, \frac{e^{\varepsilon_0}}{e^{\varepsilon_0}-1}\right\}^{HSA} \quad \times \quad \left\{\frac{-1}{e^{\varepsilon_0}-1}, \frac{e^{\varepsilon_0}}{e^{\varepsilon_0}-1}\right\}^{HSA} \quad \times$
$\left\{\frac{-1}{e^{\varepsilon_0}-1}, \frac{e^{\varepsilon_0}}{e^{\varepsilon_0}-1}\right\}^{(H-1)SAS}$

---

**Proposition 12.** *For any $\varepsilon > 0$, the Randomized Response mechanism, Alg. 5, with parameter $\varepsilon_0 = \varepsilon/6H$ is $(\varepsilon, 0)$-LDP.*

*Proof of Prop. 12:* Just as in the proof of Prop. 10 and Prop. 8, let's consider two trajectories $X = \{(s_h, a_h, r_h) \mid h \leq H\}$ and $X' = \{(s'_h, a'_h, r'_h) \mid h \leq H\}$ and also denote the output of the private randomizer $\mathcal{M}$ by $\mathcal{M}(X) = (\widetilde{R}_X, \widetilde{N}^r_X, \widetilde{N}^p_X)$ and $\mathcal{M}(X') = (\widetilde{R}_{X'}, \widetilde{N}^r_{X'}, \widetilde{N}^p_{X'})$.

For a given $r \in \left\{\frac{-1}{e^{\varepsilon_0}-1}, \frac{e^{\varepsilon_0}}{e^{\varepsilon_0}-1}\right\}^{HSA}$ (note that by definition of $r$ in Alg. 5, these are the only values it can take), we have that:

$$\frac{\mathbb{P}\left(\forall(h,s,a), \widetilde{R}_X(h,s,a) = r_{h,s,a} \mid X\right)}{\mathbb{P}\left(\forall(h,s,a), \widetilde{R}_{X'}(h,s,a) = r_{h,s,a} \mid X'\right)} = \prod_{h,s,a} \left(\frac{\frac{e^{\varepsilon_0}-1}{e^{\varepsilon_0}+1} r_h \mathbb{1}_{\{s_h=s, a_h=a\}} + \frac{1}{e^{\varepsilon_0}+1}}{\frac{e^{\varepsilon_0}-1}{e^{\varepsilon_0}+1} r'_h \mathbb{1}_{\{s'_h=s, a'_h=a\}} + \frac{1}{e^{\varepsilon_0}+1}}\right)^{y^r_{h,s,a}} \times$$
$$\times \left(\frac{1 - \left(\frac{e^{\varepsilon_0}-1}{e^{\varepsilon_0}+1} r_h \mathbb{1}_{\{s_h=s, a_h=a\}} + \frac{1}{e^{\varepsilon_0}+1}\right)}{1 - \left(\frac{e^{\varepsilon_0}-1}{e^{\varepsilon_0}+1} r'_h \mathbb{1}_{\{s'_h=s, a'_h=a\}} + \frac{1}{e^{\varepsilon_0}+1}\right)}\right)^{1-y^r_{h,s,a}} \tag{69}$$

where for every $(h, s, a) \in H \times \mathcal{S} \times \mathcal{A}$, we define $y^r_{h,s,a} = \frac{e^{\varepsilon_0}-1}{e^{\varepsilon_0}+1} r + \frac{1}{e^{\varepsilon_0}+1}$ belongs to $\{0, 1\}$ because $r \in \left\{\frac{-1}{e^{\varepsilon_0}-1}, \frac{e^{\varepsilon_0}}{e^{\varepsilon_0}-1}\right\}^{HSA}$. Eq. (69) can be rewritten as:

$$(69) = \prod_{h,s,a} \left(\frac{(e^{\varepsilon_0}-1) r_h \mathbb{1}_{\{s_h=s, a_h=a\}} + 1}{(e^{\varepsilon_0}-1) r'_h \mathbb{1}_{\{s'_h=s, a'_h=a\}} + 1}\right)^{y^r_{h,s,a}} \left(\frac{e^{\varepsilon_0} - (e^{\varepsilon_0}-1) r_h \mathbb{1}_{\{s_h=s, a_h=a\}}}{e^{\varepsilon_0} - (e^{\varepsilon_0}-1) r'_h \mathbb{1}_{\{s'_h=s, a'_h=a\}}}\right)^{1-y^r_{h,s,a}} \tag{70}$$

Then for a given $(h, s, a)$, because $r_h \in [0, 1]$ we have:

$$\frac{(e^{\varepsilon_0} - 1)r_h \mathbb{1}_{\{s_h=s,a_h=a\}} + 1}{(e^{\varepsilon_0} - 1)r'_h \mathbb{1}_{\{s'_h=s,a'_h=a\}} + 1} \leq \begin{cases} e^{\varepsilon_0} & \text{if } \mathbb{1}_{\{s_h=s,a_h=a\}} = \mathbb{1}_{\{s'_h=s,a'_h=a\}} = 1 \\ 1 & \text{if } \mathbb{1}_{\{s_h=s,a_h=a\}} = \mathbb{1}_{\{s'_h=s,a'_h=a\}} = 0 \\ e^{\varepsilon_0} & \text{if } \mathbb{1}_{\{s_h=s,a_h=a\}} = 1 \text{ and } \mathbb{1}_{\{s'_h=s,a'_h=a\}} = 0 \\ 1 & \text{if } \mathbb{1}_{\{s_h=s,a_h=a\}} = 0 \text{ and } \mathbb{1}_{\{s'_h=s,a'_h=a\}} = 1 \end{cases} \tag{71}$$

$$\frac{e^{\varepsilon_0} - (e^{\varepsilon_0} - 1)r_h \mathbb{1}_{\{s_h=s,a_h=a\}}}{e^{\varepsilon_0} - (e^{\varepsilon_0} - 1)r'_h \mathbb{1}_{\{s'_h=s,a'_h=a\}}} \leq \begin{cases} e^{\varepsilon_0} & \text{if } \mathbb{1}_{\{s_h=s,a_h=a\}} = \mathbb{1}_{\{s'_h=s,a'_h=a\}} = 1 \\ 1 & \text{if } \mathbb{1}_{\{s_h=s,a_h=a\}} = \mathbb{1}_{\{s'_h=s,a'_h=a\}} = 0 \\ 1 & \text{if } \mathbb{1}_{\{s_h=s,a_h=a\}} = 1 \text{ and } \mathbb{1}_{\{s'_h=s,a'_h=a\}} = 0 \\ e^{\varepsilon_0} & \text{if } \mathbb{1}_{\{s_h=s,a_h=a\}} = 0 \text{ and } \mathbb{1}_{\{s'_h=s,a'_h=a\}} = 1 \end{cases} \tag{72}$$

Therefore, we can simplify each term in (70) by:

$$\frac{(e^{\varepsilon_0} - 1)r_h \mathbb{1}_{\{s_h=s,a_h=a\}} + 1}{(e^{\varepsilon_0} - 1)r'_h \mathbb{1}_{\{s'_h=s,a'_h=a\}} + 1} \leq \exp\left(\varepsilon_0\left(\mathbb{1}_{\{s_h=s,a_h=a\}} + \mathbb{1}_{\{s'_h=s,a'_h=a\}}\right)\right)$$

$$\frac{e^{\varepsilon_0} - (e^{\varepsilon_0} - 1)r_h \mathbb{1}_{\{s_h=s,a_h=a\}}}{e^{\varepsilon_0} - (e^{\varepsilon_0} - 1)r'_h \mathbb{1}_{\{s'_h=s,a'_h=a\}}} \leq \exp\left(\varepsilon_0\left(\mathbb{1}_{\{s_h=s,a_h=a\}} + \mathbb{1}_{\{s'_h=s,a'_h=a\}}\right)\right)$$

Hence, using the two inequalities above:

$$(70) \leq \prod_{h,s,a} \exp\left(y^r_{h,s,a}\varepsilon_0\left(\mathbb{1}_{\left\{\substack{s_h=s, \\ a_h=a}\right\}} + \mathbb{1}_{\left\{\substack{s'_h=s, \\ a'_h=a}\right\}}\right) + (1 - y^r_{h,s,a})\varepsilon_0\left(\mathbb{1}_{\left\{\substack{s'_h=s, \\ a'_h=a}\right\}} + \mathbb{1}_{\left\{\substack{s_h=s, \\ a_h=a}\right\}}\right)\right)$$

$$= \prod_{h,s,a} \exp\left(\varepsilon_0\left(\mathbb{1}_{\left\{\substack{s_h=s, \\ a_h=a}\right\}} + \mathbb{1}_{\left\{\substack{s'_h=s, \\ a'_h=a}\right\}}\right)\right)$$

$$= \exp\left(2\varepsilon_0 H\right)$$

In addition, let's consider $m \in \left\{\frac{-1}{e^{\varepsilon_0}-1}, \frac{e^{\varepsilon_0}}{e^{\varepsilon_0}-1}\right\}^{H \times S \times A}$ and $y = \frac{e^{\varepsilon_0}-1}{e^{\varepsilon_0}+1}m + \frac{1}{e^{\varepsilon_0}+1} \in \{0, 1\}$, we then have that:

$$\frac{\mathbb{P}\left(\forall (h,s,a), \widetilde{N}^r_X(h,s,a) = m_{h,s,a} \mid X\right)}{\mathbb{P}\left(\forall (h,s,a), \widetilde{N}^r_{X'}(h,s,a) = m_{h,s,a} \mid X'\right)} = \prod_{h,s,a} \left(\frac{\frac{e^{\varepsilon_0}-1}{e^{\varepsilon_0}+1}\mathbb{1}_{\{s_h=s,a_h=a\}} + \frac{1}{e^{\varepsilon_0}+1}}{\frac{e^{\varepsilon_0}-1}{e^{\varepsilon_0}+1}\mathbb{1}_{\{s'_h=s,a'_h=a\}} + \frac{1}{e^{\varepsilon_0}+1}}\right)^{y_{h,s,a}} \times$$

$$\times \left(\frac{1 - \left(\frac{e^{\varepsilon_0}-1}{e^{\varepsilon_0}+1}\mathbb{1}_{\{s_h=s,a_h=a\}} + \frac{1}{e^{\varepsilon_0}+1}\right)}{1 - \left(\frac{e^{\varepsilon_0}-1}{e^{\varepsilon_0}+1}\mathbb{1}_{\{s'_h=s,a'_h=a\}} + \frac{1}{e^{\varepsilon_0}+1}\right)}\right)^{1-y_{h,s,a}} \tag{73}$$

Which can be rewritten as:

$$\frac{\mathbb{P}\left(\forall (h,s,a), \widetilde{N}^r_X(h,s,a) = m_{h,s,a} \mid X\right)}{\mathbb{P}\left(\forall (h,s,a), \widetilde{N}^r_{X'}(h,s,a) = m_{h,s,a} \mid X'\right)} = \prod_{h,s,a} \left(\frac{(e^{\varepsilon_0}-1)\mathbb{1}_{\{s_h=s,a_h=a\}} + 1}{(e^{\varepsilon_0}-1)\mathbb{1}_{\{s'_h=s,a'_h=a\}} + 1}\right)^{y_{h,s,a}} \times$$

$$\times \left(\frac{e^{\varepsilon_0} - (e^{\varepsilon_0}-1)\mathbb{1}_{\{s_h=s,a_h=a\}}}{e^{\varepsilon_0} - (e^{\varepsilon_0}-1)\mathbb{1}_{\{s'_h=s,a'_h=a\}}}\right)^{1-y_{h,s,a}} \tag{74}$$

Thus for a given $(h, s, a)$:

$$\frac{(e^{\varepsilon_0}-1)\mathbb{1}_{\{s_h=s,a_h=a\}} + 1}{(e^{\varepsilon_0}-1)\mathbb{1}_{\{s'_h=s,a'_h=a\}} + 1} = \begin{cases} 1 & \text{if } \mathbb{1}_{\{s_h=s,a_h=a\}} = \mathbb{1}_{\{s'_h=s,a'_h=a\}} \\ e^{\varepsilon_0} & \text{if } \mathbb{1}_{\{s_h=s,a_h=a\}} = 1 \text{ and } \mathbb{1}_{\{s'_h=s,a'_h=a\}} = 0 \\ e^{-\varepsilon_0} & \text{if } \mathbb{1}_{\{s_h=s,a_h=a\}} = 0 \text{ and } \mathbb{1}_{\{s'_h=s,a'_h=a\}} = 1 \end{cases} \tag{75}$$

$$\frac{e^{\varepsilon_0} - (e^{\varepsilon_0}-1)\mathbb{1}_{\{s_h=s,a_h=a\}}}{e^{\varepsilon_0} - (e^{\varepsilon_0}-1)\mathbb{1}_{\{s'_h=s,a'_h=a\}}} = \begin{cases} 1 & \text{if } \mathbb{1}_{\{s_h=s,a_h=a\}} = \mathbb{1}_{\{s'_h=s,a'_h=a\}} \\ e^{-\varepsilon_0} & \text{if } \mathbb{1}_{\{s_h=s,a_h=a\}} = 1 \text{ and } \mathbb{1}_{\{s'_h=s,a'_h=a\}} = 0 \\ e^{\varepsilon_0} & \text{if } \mathbb{1}_{\{s_h=s,a_h=a\}} = 0 \text{ and } \mathbb{1}_{\{s'_h=s,a'_h=a\}} = 1 \end{cases} \tag{76}$$

Therefore, here again we can simplify each term in (74) by:

$$\frac{(e^{\varepsilon_0}-1)\mathbb{1}_{\{s_h=s,a_h=a\}}+1}{(e^{\varepsilon_0}-1)\mathbb{1}_{\{s'_h=s,a'_h=a\}}+1} \leq \exp\left(\varepsilon_0\left(\mathbb{1}_{\{s_h=s,a_h=a\}} - \mathbb{1}_{\{s'_h=s,a'_h=a\}}\right)\right)$$

$$\frac{e^{\varepsilon_0}-(e^{\varepsilon_0}-1)\mathbb{1}_{\{s_h=s,a_h=a\}}}{e^{\varepsilon_0}-(e^{\varepsilon_0}-1)\mathbb{1}_{\{s'_h=s,a'_h=a\}}} \leq \exp\left(\varepsilon_0\left(\mathbb{1}_{\{s_h=s,a_h=a\}} - \mathbb{1}_{\{s'_h=s,a'_h=a\}}\right)\right)$$

Therefore:

$$(74) = \prod_{h,s,a} \exp\left(y_{h,s,a}\varepsilon_0\left(\mathbb{1}_{\left\{\substack{s_h=s,\\a_h=a}\right\}} - \mathbb{1}_{\left\{\substack{s'_h=s,\\a'_h=a}\right\}}\right) + (1-y_{h,s,a})\varepsilon_0\left(\mathbb{1}_{\left\{\substack{s'_h=s,\\a'_h=a}\right\}} - \mathbb{1}_{\left\{\substack{s_h=s,\\a_h=a}\right\}}\right)\right)$$

$$= \prod_{h,s,a} \exp\left((2y_{h,s,a}-1)\varepsilon_0\left(\mathbb{1}_{\{s_h=s,a_h=a\}} - \mathbb{1}_{\{s'_h=s,a'_h=a\}}\right)\right)$$

$$\leq \exp\left(2\varepsilon_0 H\right)$$

Using the same reasonning we have that for any $m' \in \left\{-\frac{1}{e^{\varepsilon_0}-1}, \frac{e^{\varepsilon_0}}{e^{\varepsilon_0}-1}\right\}^{(H-1)\times S\times A\times S}$:

$$\frac{\mathbb{P}\left(\forall(h,s,a,s'), \widetilde{N}^p_X(h,s,a,s') = m'_{h,s,a,s'} \mid X\right)}{\mathbb{P}\left(\forall(h,s,a,s'), \widetilde{N}^p_{X'}(h,s,a,s') = m'_{h,s,a,s'} \mid X'\right)} \leq \exp(2\varepsilon_0 H) \tag{77}$$

We conclude the proof the same way as the proof of Prop. 7. $\qquad\square$

In addition, the precision $c_{k,1}$, $c_{k,2}$, $c_{k,3}$ and $c_{k,4}$ of the Randomized Response mechanism are still of order $\sqrt{k}$ just as the Gaussian and Laplace mechanisms. Contrary to any of those two, the dependence is exponential on $\varepsilon_0$ which is closer to the lower bound of Sec. 3. Indeed, we have an additional factor $S$ for $c_{k,3}$ compared to the other mechanisms but those terms scale with $1/(e^{\varepsilon_0}-1)$ instead of the worse dependency $1/\varepsilon$.

**Proposition 13.** *The Randomized Response mechanism, Alg. 5, with parameter $\varepsilon_0 > 0$ satisfies Def. 3 for any $\delta > 0$ and $k \in \mathbb{N}^\star$ with:*

$$c_{k,1}(\varepsilon_0,\delta) = c_{k,2}(\varepsilon_0,\delta) = \max\left\{1, \frac{2e^{\varepsilon_0}-1}{e^{\varepsilon_0}-1}\sqrt{\frac{(k-1)H}{2}\ln\left(\frac{4SA}{\delta}\right)}\right\}$$

$$c_{k,3}(\varepsilon_0,\delta) = \max\left\{1, \frac{S(2e^{\varepsilon_0}-1)}{e^{\varepsilon_0}-1}\sqrt{\frac{(k-1)H}{2}\ln\left(\frac{4SA}{\delta}\right)}\right\}$$

$$c_{k,4}(\varepsilon_0,\delta) = \max\left\{1, \frac{2e^{\varepsilon_0}-1}{e^{\varepsilon_0}-1}\sqrt{\frac{(k-1)H}{2}\ln\left(\frac{4S^2A}{\delta}\right)}\right\}$$

*Proof of Prop. 13:* Let's consider a given state-action-next state tuple, $(s,a,s')$, then when summing over $h$:

$$\left|\sum_{h=1}^{H} \widetilde{N}^r_k(h,s,a) - \sum_{l<k}\sum_{h=1}^{H}\mathbb{1}_{\{s_{l,h}=s,a_{l,h}=a\}}\right| = \left|\sum_{h=1}^{H}\sum_{l<k}\widetilde{N}^r_{X_l}(h,s,a) - \mathbb{1}_{\{s_{l,h}=s,a_{l,h}=a\}}\right| \tag{78}$$

We now construct a filtration $(\mathcal{F}_{k,h})_{k,h}$ such that $(\widetilde{N}^r_{X_l}(h,s,a) - \mathbb{1}_{\{s_{l,h}=s,a_{l,h}=a\}})_{l,h}$ is a Martingale Difference Sequence. For an episode $k$ and step $h$, define $\mathcal{F}_{k,h} = \sigma(\{(s_{l,j},a_{l,j},r_{l,j})_{j\leq H}, \mathcal{M}((s_{l,j},a_{l,j},r_{l,j})_{j\leq H})\} \mid l < k\} \cup \{(s_{k,j},a_{k,j},r_{k,j})_{j\leq h}\})$ to be the filtration that contains the history before episode $k$. Then $\mathbb{1}_{\{s_{k,h}=s,a_{k,h}=a\}}$ is $\mathcal{F}_{k,h}$-measurable and thus we have:

$$\mathbb{E}\left(\widetilde{N}^r_{X_k}(h,s,a) - \mathbb{1}_{\{s_{k,h}=s,a_{k,h}=a\}} \mid \mathcal{F}_{k,h}\right) = \frac{e^{\varepsilon_0}+1}{e^{\varepsilon_0}-1}\left(\mathbb{E}\left(\widetilde{n}_{X_k}(h,s,a) \mid \mathcal{F}_{k,h}\right) - \frac{1}{e^{\varepsilon_0}+1}\right)$$
$$-\mathbb{1}_{\{s_{k,h}=s,a_{k,h}=a\}} = 0$$

where $\tilde{n}_{X_k}(h, s, a)$ is a Randomized Response random variable generated by Alg. 5 for each step $h$, state $s$, action $a$ and trajectory $X_k$. And $\left|\widetilde{N}_{X_k}^r(h, s, a) - \mathbb{1}_{\{s_{k,h}=s, a_{k,h}=a\}}\right| \leq \frac{2e^{\varepsilon_0}-1}{e^{\varepsilon_0}-1}$. Then thanks to Azuma-Hoeffding inequality we have that with probability at least $1 - \delta/(4SA)$:

$$\left|\sum_{h=1}^{H}\widetilde{N}_k^r(h, s, a) - \sum_{l<k}\sum_{h=1}^{H}\mathbb{1}_{\{s_{l,h}=s, a_{l,h}=a\}}\right| \leq \frac{2e^{\varepsilon_0}-1}{e^{\varepsilon_0}-1}\sqrt{\frac{(k-1)H}{2}\ln\left(\frac{4SA}{\delta}\right)} \quad (79)$$

With the same reasonning, we have with probability at least $1 - \delta/4S^2A$:

$$\left|\sum_{h=1}^{H}\widetilde{N}_k^p(h, s, a, s') - \sum_{l<k}\sum_{h=1}^{H-1}\mathbb{1}_{\{s_{l,h}=s, a_{l,h}=a, s_{l,h+1}=s'\}}\right| \leq \frac{2e^{\varepsilon_0}-1}{e^{\varepsilon_0}-1}\sqrt{\frac{(k-1)H}{2}\ln\left(\frac{4S^2A}{\delta}\right)} \quad (80)$$

Also, we have:

$$\left|\sum_{h=1}^{H}\widetilde{R}_k^r(h, s, a) - \sum_{l<k}\sum_{h=1}^{H}r_h\mathbb{1}_{\{s_{l,h}=s, a_{l,h}=a\}}\right| \leq \frac{2e^{\varepsilon_0}-1}{e^{\varepsilon_0}-1}\sqrt{\frac{(k-1)H}{2}\ln\left(\frac{4SA}{\delta}\right)} \quad (81)$$

with $\widetilde{R}_k^r(h, s, a) = \sum_{l<k}\widetilde{R}_{X_l}$. Finally, with probability at least $1 - \delta/4SA$:

$$\left|\sum_{h=1}^{H}\sum_{s'}\widetilde{N}_k^p(h, s, a, s') - \sum_{s'}\sum_{l<k}\sum_{h=1}^{H-1}\mathbb{1}_{\left\{\begin{smallmatrix}s_{l,h}=s,\\a_{l,h}=a,\\s_{l,h+1}=s'\end{smallmatrix}\right\}}\right| \leq \frac{S(2e^{\varepsilon_0}-1)}{e^{\varepsilon_0}-1}\sqrt{\frac{(k-1)H}{2}\ln\left(\frac{4SA}{\delta}\right)} \quad (82)$$

Compared to the bounds we derived for previous mechanisms there is an additional factor $\sqrt{S}$. This comes from using a triangular inequality instead of using concentration inequalities like in previous mechanisms. Then thanks to a union bound over the state-action pair and the state-action-next state tuple we have that the Randomized Response mechanism satisfies Def. 3 with:

$$c_{k,1}(\varepsilon_0, \delta) = c_{k,2}(\varepsilon_0, \delta) = \max\left\{1, \frac{2e^{\varepsilon_0}-1}{e^{\varepsilon_0}-1}\sqrt{\frac{(k-1)H}{2}\ln\left(\frac{4SA}{\delta}\right)}\right\} \quad (83)$$

$$c_{k,3}(\varepsilon_0, \delta) = \max\left\{1, \frac{S(2e^{\varepsilon_0}-1)}{e^{\varepsilon_0}-1}\sqrt{\frac{(k-1)H}{2}\ln\left(\frac{4SA}{\delta}\right)}\right\}, \quad (84)$$

$$c_{k,4}(\varepsilon_0, \delta) = \max\left\{1, \frac{2e^{\varepsilon_0}-1}{e^{\varepsilon_0}-1}\sqrt{\frac{(k-1)H}{2}\ln\left(\frac{4S^2A}{\delta}\right)}\right\} \quad (85)$$

$\square$

### F.3 Bounded Noise Mechanism for DP:

Recently, [40] showed how to construct a differential privacy with an almost surely bounded noise mechanism. This mechanism, $\mathcal{M}$, computes an $(\varepsilon, \delta)$-DP approximation of the average of a dataset $\mathcal{D} = \{x_1, \dots, x_n\} \subset \mathbb{R}^{n \times k}$, for any $\varepsilon > 0$ and $\delta \in [\exp(-k/\log(k)^8), 1/2]$ (see Theorem 1.1 in [40]). In the local differentially private setting in RL, we apply this bounded noise mechanism to each user $k$ in order to compute the cumulative reward for each state-action $(s, a)$, the number of visits to $(s, a)$ and the number of visits to state-action-next state tuple $(s, a, s')$.

This noise mechanism is similar to the Laplace or Gaussian mechanism and add a noise drawn from a well-chosen distribution, $\mu_{DE,R}$ supported on $(-R, R)$ for any $R$, whose density at $\eta \in (-R, R)$ is:

$$\frac{\exp(-f_{DE,R}(\eta))}{Z_{DE,R}} \text{ with } f_{DE,R}(\eta) = \exp\left(\frac{R^2}{R^2 - \eta^2}\right) \text{ and } Z_{DE,R} = \int_{-R}^{R} e^{-f_{DE,R}(\eta)}d\eta \quad (86)$$

[40] shows that when taking $\delta \geq \exp(-k/\log(k)^8)$ and $\varepsilon \in (0, 1)$ there exists a universal constant $C > 0$ such that when taking $R = \frac{C}{\varepsilon n}\sqrt{k\log\left(\frac{1}{\delta}\right)}$ adding noise from $\mu_{DE,R}$ ensures $(\varepsilon, \delta)$-DP to the average of $n$ data of dimension $k$.

Similarly to the previous mechanisms we studied we can show the following proposition, which states the parameter we need to use to ensure $(\varepsilon, \delta)$-DP.

---

**Algorithm 6** Bounded Noise Mechanism for LDP

---

**Input:** Trajectory: $X = \{(s_h, a_h, r_h) \mid h \leq H\}$, Privacy Parameter: $\varepsilon, \delta$, Constant: $C$
Set $R_1 = \frac{C}{\varepsilon}\sqrt{SA\ln(1/\delta)}$ and $R_2 = \frac{CS}{\varepsilon}\sqrt{A\ln(1/\delta)}$
**for** $(s,a) \in \mathcal{S} \times \mathcal{A}$ **do**
    Sample $Y_{1,X}(s,a) \sim \mu_{\text{DE},R_1}$
    $\widetilde{R}_X(s,a) = Y_{1,X}(s,a) + \sum_{h=1}^{H} r_h \mathbb{1}_{\{s_h = s, a_h = a\}}$
    Sample $\widetilde{n}_X^r(s,a) \sim \mu_{\text{DE},R_1}$
    $\widetilde{N}_X^r(s,a) = \widetilde{n}_X^r(s,a) + \sum_{h=1}^{H} \mathbb{1}_{\{s_h = s, a_h = a\}}$
    **for** $s' \in \mathcal{S}$ **do**
        Sample $\widetilde{n}_X^p(s,a,s') \sim \mu_{\text{DE},R_2}$
        $\widetilde{N}_X^p(s,a,s') = \widetilde{n}_X^r(s,a,s') + \sum_{h=1}^{H-1} \mathbb{1}_{\{s_h = s, a_h = a, s_{h+1} = s'\}}$
    **end for**
**end for**
**Return:** $(\widetilde{R}_X, \widetilde{N}_X^r, \widetilde{N}_X^p) \in \mathbb{R}^{S \times A} \times \mathbb{R}^{S \times A} \times \mathbb{R}^{S \times A \times S}$

---

**Proposition 14.** *For any $\varepsilon \in (0,1)$, $\delta_0 \geq \exp(-SA/\log(SA)^8)$ and $\delta_1 \geq \exp(-S^2A/\log(S^2A)^8)$ then the bounded noise mechanism, Alg. 6, is $(3H\varepsilon, \delta')$-LDP with $\delta_0' = \delta_0 \frac{e^{H\varepsilon}-1}{e^\varepsilon - 1}$, $\delta_1' = \delta_1 \frac{e^{H\varepsilon}-1}{e^\varepsilon-1}$ and $\delta' = \delta_1' e^{2H\varepsilon} + 2\delta_0' e^{2H\varepsilon} + 2\delta_0'\delta_1' e^{H\varepsilon} + (\delta_0')^2 e^{H\varepsilon} + (\delta_0')^2\delta_1'$.*

*Proof.* of Prop. 14

For any $\varepsilon \in (0,1)$ and $\delta_0 \geq \exp(-SA/\log(SA)^8)$, for any $r \in \mathbb{R}^{S\times A}$ and two trajectories $X = \{(s_h, a_h, r_h)_{h\leq H}\}$ and $X' = \{(s_h', a_h', r_h')_{h\leq H}\}$ let's define $R_X(s,a) = \sum_{h=1}^{H} r_h \mathbb{1}_{\{s_h=s, a_h=a\}}$ the cumulative reward in state-action $(s,a)$ associated to trajectory $X$. Finally, let's define for a set of indexes $I \subset [\![H]\!]$ the new trajectory $X_I$ where for $h \in I$, $(X_I)_h = (s_h, a_h, r_h)$ and for $h \notin I$, $(X_I)_h = (s_h', a_h', r_h')$. Therefore, using Theorem 3.2 from [40], we have that for $I = [\![H-1]\!]$ and $\widetilde{R}_X$ defined as in Alg. 6,

$$\mathbb{P}\left(\widetilde{R}_X = r\right) \leq \exp(\varepsilon)\mathbb{P}\left(\widetilde{R}_{X_I} = r\right) + \delta_0 \tag{87}$$

$$\leq \exp(\varepsilon)\left(\exp(\varepsilon)\mathbb{P}\left(\widetilde{R}_{X_{[H-2]}} = r\right) + \delta_0\right) + \delta_0 \tag{88}$$

Therefore repeating the same argument $H$ times, we have that:

$$\mathbb{P}\left(\widetilde{R}_X = r\right) \leq \exp(H\varepsilon)\mathbb{P}\left(\widetilde{R}_{X'} = r\right) + \delta_0 \sum_{h=0}^{H-1} \exp(h\varepsilon) \tag{89}$$

$$= \exp(H\varepsilon)\mathbb{P}\left(\widetilde{R}_{X'} = r\right) + \delta_0 \frac{\exp(H\varepsilon) - 1}{\exp(\varepsilon) - 1} \tag{90}$$

In addition, we have with the same reasoning that for any $n \in \mathbb{R}^{S\times A}$ and $n^p \in \mathbb{R}^{S\times A\times S}$ that:

$$\mathbb{P}\left(\widetilde{N}_X^r = n\right) \leq \exp(H\varepsilon)\mathbb{P}\left(\widetilde{N}_{X'} = n\right) + \delta_0 \frac{\exp(H\varepsilon) - 1}{\exp(\varepsilon) - 1} \tag{91}$$

and for any $\delta_1 \geq \exp(-S^2A/\log(S^2A)^8)$:

$$\mathbb{P}\left(\widetilde{N}_X^p = n^p\right) \leq \exp(H\varepsilon)\mathbb{P}\left(\widetilde{N}_{X'}^p = n^p\right) + \delta_1 \frac{\exp(H\varepsilon) - 1}{\exp(\varepsilon) - 1} \tag{92}$$

Therefore we have that:

$$\mathbb{P}\left(\widetilde{R}_X = r, \widetilde{N}_X^r = n, \widetilde{N}_X^p = n^p \mid X\right) = \mathbb{P}\left(\widetilde{R}_X = r \mid X\right)\mathbb{P}\left(\widetilde{N}_X^r = n \mid X\right)\mathbb{P}\left(\widetilde{N}_X^p = n^p \mid X\right)$$

$$\leq \left(e^{H\varepsilon}\mathbb{P}\left(\widetilde{R}_{X'} = r\right) + \delta_0 \frac{e^{H\varepsilon} - 1}{e^{\varepsilon} - 1}\right)\left(e^{H\varepsilon}\mathbb{P}\left(\widetilde{N}_{X'} = n\right) + \delta_0 \frac{e^{H\varepsilon} - 1}{e^{\varepsilon} - 1}\right) \times$$

$$\times \left(e^{H\varepsilon}\mathbb{P}\left(\widetilde{N}_{X'}^p = n^p\right) + \delta_1 \frac{e^{H\varepsilon} - 1}{e^{\varepsilon} - 1}\right)$$

$$\leq e^{3H\varepsilon}\mathbb{P}\left(\widetilde{R}_{X'} = r, \widetilde{N}_{X'}^r = n, \widetilde{N}_{X'}^p = n^p\right) + \delta_1' e^{2H\varepsilon}\mathbb{P}\left(\widetilde{R}_{X'} = r\right)\mathbb{P}\left(\widetilde{N}_{X'}^r = n\right)$$

$$+ \delta_0' e^{2H\varepsilon}\mathbb{P}\left(\widetilde{N}_{X'}^p = n^p\right)\left(\mathbb{P}\left(\widetilde{N}_{X'}^r = n\right) + \mathbb{P}\left(\widetilde{R}_{X'} = r\right)\right)$$

$$+ \delta_0'\delta_1' e^{H\varepsilon}\left(\mathbb{P}\left(\widetilde{N}_{X'}^r = n\right) + \mathbb{P}\left(\widetilde{R}_{X'} = r\right)\right) + (\delta_0')^2 e^{H\varepsilon}\mathbb{P}\left(\widetilde{N}_{X'}^p = n^p\right) + (\delta_0')^2 \delta_1'$$

with $\delta_0' = \delta_0 \frac{e^{H\varepsilon}-1}{e^{\varepsilon}-1}$ and $\delta_1' = \delta_1 \frac{e^{H\varepsilon}-1}{e^{\varepsilon}-1}$. Therefore, we have that the mechanism is $(3H\varepsilon, \delta')$-LDP that is to say:

$$\mathbb{P}\left(\widetilde{R}_X = r, \widetilde{N}_X^r = n, \widetilde{N}_X^p = n^p \mid X\right) \leq e^{3H\varepsilon}\mathbb{P}\left(\widetilde{R}_{X'} = r, \widetilde{N}_{X'}^r = n, \widetilde{N}_{X'}^p = n^p\right) + \delta_1' e^{2H\varepsilon}$$

$$+ 2\delta_0' e^{2H\varepsilon} + 2\delta_0'\delta_1' e^{H\varepsilon} + (\delta_0')^2 e^{H\varepsilon} + (\delta_0')^2 \delta_1'$$

with $\delta_0' = \delta_0 \frac{e^{H\varepsilon}-1}{e^{\varepsilon}-1}$, $\delta_1' = \delta_1 \frac{e^{H\varepsilon}-1}{e^{\varepsilon}-1}$ and $\delta' = \delta_1' e^{2H\varepsilon} + 2\delta_0' e^{2H\varepsilon} + 2\delta_0'\delta_1' e^{H\varepsilon} + (\delta_0')^2 e^{H\varepsilon} + (\delta_0')^2 \delta_1'$. $\qquad\square$

In addition, because the noise is bounded we can apply standard sub-gaussian concentration inequalities to show that Alg. 6 satisfies Def. 1.

**Proposition 15.** *The bounded noise mechanism, Alg. 6, with parameter $\varepsilon_0 > 0$ satisfies Def. 3 for any $\delta > 0$ and $k \in \mathbb{N}^\star$ with:*

$$c_{k,1}(\varepsilon_0, \delta) = c_{k,2}(\varepsilon_0, \delta) = R\sqrt{2(k-1)\ln\left(\frac{6SA}{\delta}\right)}$$

$$c_{k,3}(\varepsilon_0, \delta) = R_2\sqrt{2S(k-1)\ln\left(\frac{6S^2A}{\delta}\right)}$$

$$c_{k,4}(\varepsilon_0, \delta) = R_2\sqrt{2(k-1)\ln\left(\frac{6S^2A}{\delta}\right)}$$

*with $R = \frac{1}{\varepsilon}\sqrt{SA\ln(1/\delta_0)}$ and $R_2 = \frac{S}{\varepsilon}\sqrt{A\ln(1/\delta_0)}$*

*Proof.* of Prop. 15 For any $\delta > 0$ and at the beginning of episode $k$, we have thanks to Hoeffding inequality that with probability at least $1 - \frac{\delta}{3SA}$ for any state-action $(s,a) \in \mathcal{S} \times \mathcal{A}$:

$$\left|\widetilde{R}_k(s,a) - R_k(s,a)\right| = \left|\sum_{l=1}^{k-1} Y_{1,X_l}(s,a)\right| \leq R\sqrt{2(k-1)\ln\left(\frac{6SA}{\delta}\right)} \tag{93}$$

with $(Y_{1,X_l}(s,a))_{l \leq k-1}$ are i.i.d distributed according to $\mu_{\text{DE},R_1}$. With the same reasonning, we have that with probability at least $1 - \frac{\delta}{3SA}$:

$$\left|\widetilde{N}_k^r(s,a) - N_k^r(s,a)\right| = \left|\sum_{l=1}^{k-1} \widetilde{n}_{X_l}^r(s,a)\right| \leq R\sqrt{2(k-1)\ln\left(\frac{6SA}{\delta}\right)} \tag{94}$$

Finally, still using Hoeffding inequality, and definning $R_2 = \frac{CS}{\varepsilon}\sqrt{A\ln(1/\delta)}$, we have that with probability at least $1 - \frac{\delta}{3S^2A}$:

$$\left|\widetilde{N}_k^p(s,a,s') - \sum_{l<k}\sum_{h=1}^{H-1}\mathbb{1}_{\{s_{l,h}=s,a_{l,h}=a,s_{l,h+1}=s'\}}\right| \leq R_2\sqrt{2(k-1)\ln\left(\frac{6S^2A}{\delta}\right)} \tag{95}$$

And finally with probability at least $1 - \frac{\delta}{3SA}$:

$$\left| \sum_{s' \in \mathcal{S}} \widetilde{N}_k^p(s, a, s') - \sum_{s' \in \mathcal{S}} \sum_{l < k} \sum_{h=1}^{H-1} \mathbb{1}_{\{s_{l,h} = s, a_{l,h} = a, s_{l,h+1} = s'\}} \right| \leq R_2 \sqrt{2S(k-1) \ln\left(\frac{6S^2A}{\delta}\right)} \tag{96}$$

$\square$

### F.4 Experimental Results:

We show empirical results for three mechanisms discussed in the RandomMDP environment in Figures 4, 5 and 6.

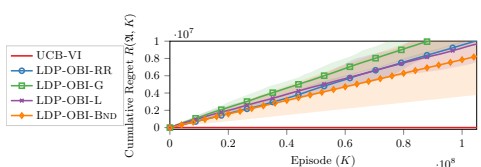

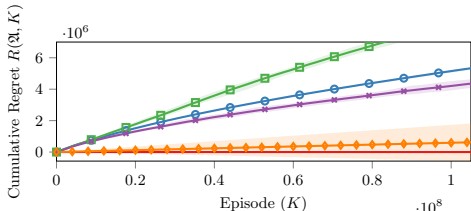

Figure 4: $\varepsilon = 0.2$ and $\delta = 0.1$ (*only* for the Gaussian and bounded noise mechanism)

Figure 5: $\varepsilon = 2$ and $\delta = 0.1$ (*only* for the Gaussian and bounded noise mechanism)

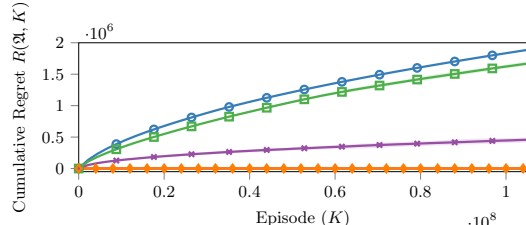

Figure 6: $\varepsilon = 20$ and $\delta = 0.1$ (*only* for the Gaussian and bounded noise mechanism)

As we have seen in Fig. 1, the LDP constraint has a significant impact on the regret especially as $\varepsilon$ decreases. In particular for $\varepsilon = 0.2$, LDP-OBI-L, LDP-OBI-G, LDP-OBI-RR, LDP-OBI-BND have not reached the usual square root growth phase of the regret usually seen in UCB-VI or other regret minimizing algorithm.

From figures 4, 5 and 6, we can observe that the bounded noise mechanism has a lower impact on the regret compared to the Laplace, Gaussian and Randomized Response mechanisms. However, this benefit does not appear in the regret bound of Table 1. This suggests that the regret analysis of Sec. 4.3 may be improved to show this empirically observed advantage.

## G  Posterior Sampling for Local Differential Privacy

The Posterior Sampling for Reinforcement Learning algorithm [PSRL, 12] is a Thompson Sampling based algorithm for Reinforcement Learning. It works by maintaining a Bayesian posterior distribution over MDPs. We focus on a particular instantiation of PSRL where for each state-action pair $(s, a)$ we have an independent Gaussian prior for the reward distribution and a Dirichlet prior for the transition dynamics. With those priors, the posterior distributions are Normal-Gamma and Dirichlet distributions.

Let $\alpha_0(s, a)$ denote the parameters of the prior distribution over the transition dynamics, so the prior is given by $\text{Dir}(\alpha_0(s, a))$. In addition, let $\mu_0(s, a) \in \mathbb{R}$, $\lambda_0(s, a) \in \mathbb{R}_+^\star$, $\nu_0(s, a) \in \mathbb{R}_+^\star$ and $\beta_0(s, a) \in \mathbb{R}_+^\star$ be the parameters of the Normal-Gamma prior distribution that we place on the rewards. Then, at the beginning of episode $k$ and for a given pair $(s, a) \in \mathcal{S} \times \mathcal{A}$, let $\alpha_k(s, a) \in (\mathbb{R}_+^\star)^S$ be such that the posterior distribution over the transition dynamics is $\text{Dir}(\alpha_k(s, a))$. We then define $\mu_k(s, a) \in \mathbb{R}$, $\lambda_k(s, a) \in \mathbb{R}_+^\star$, $\nu_k(s, a) \in \mathbb{R}_+^\star$ and $\beta_k(s, a) \in \mathbb{R}_+^\star$ to the parameters of the Normal-Gamma posterior distributions. Using standard results from Bayesian Learning we have that, for all

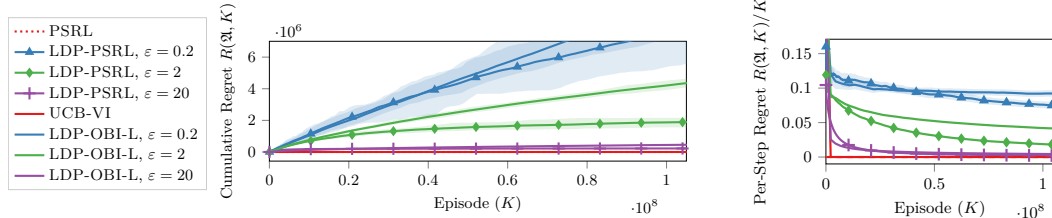

Figure 7: Evaluation of LDP-PSRL in the RandomMDP environment. *Left)* Cumulative regret. *Right)* per-step regret ($k \mapsto R_k/k$). Results are averaged over 20 runs and the the confidence intervals are the minimum and maximum runs. While the regret looks almost linear for $\varepsilon = 0.2$, the decreasing trend of the per-step regret shows that the algorithms are learning.

state $s' \in \mathcal{S}$:

$$\alpha_k(s,a) = \alpha_0(s,a) + N_k(s,a,s') \tag{97}$$

$$\lambda_k(s,a) = \lambda_0(s,a) + N_k(s,a) \tag{98}$$

$$\nu_k(s,a) = \nu_0(s,a) + \frac{N_k(s,a)}{2} \tag{99}$$

$$\mu_k(s,a) = \frac{\lambda_0(s,a)\mu_0(s,a) + N_k(s,a)\hat{R}_k(s,a)}{\lambda_0(s,a) + N_k(s,a)} \tag{100}$$

$$\beta_k(s,a) = \beta_0(s,a) + \frac{1}{2}\widehat{\text{Var}}(R(s,a)) + \frac{N_k(s,a)\lambda_0(s,a)}{2(\lambda_0(s,a) + N_k(s,a))}\left(\hat{R}_k(s,a) - \mu_0(s,a)\right)^2 \tag{101}$$

where $\alpha_0, \mu_0, \lambda_0, \nu_0, \beta_0$ are prior parameters provided at the beginning of the algorithm. We denote by $N_k(s,a)$, the number of visits to the state-action pair $(s,a)$, $N_k(s,a,s')$ the number visits to $(s,a,s')$, $\hat{R}_k(s,a)$ the average reward observed for $(s,a)$ and $\widehat{\text{Var}}(R(s,a))$ the empirical variance for $(s,a)$.

At each episode $k$, PSRL samples an MDP from the posterior distributions, then computes the optimal policy and executes it in the true MDP. [12] showed that the *Bayesian* regret of this algorithm is bounded by $\tilde{O}\left(HS\sqrt{AT}\right)$.

**Locally Differentially Private Posterior Sampling for Reinforcement Learning:** We now discuss how to adapt PSRL to ensure it is locally differentially private. Def. 1 states that LDP is ensured at the collection time of trajectories therefore it is enough for us to design a LDP posterior sampling algorithm which takes as input the trajectories outputted by a mechanism similar to Alg. 3. Here, we use the LDP mechanism to pertub the statistics used to define the parameters of the posterior distribution in PSRL. More precisely, we replace the aggregate counts in Eqs. 97-101 by noisy counts provided by an LDP mechanism. In order to do this, we need to modify the initial values of those parameters to guarantee they are non-negative.

In this appendix, we assume that the privacy-preserving mechanism $\mathcal{M}$ is such that for a given trajectory $X$, $\mathcal{M}(X) = (\widetilde{R}_X, \widetilde{R}_{2,X}, \widetilde{N}_X^r, \widetilde{N}_X^p)$ where $\widetilde{R}_X, \widetilde{R}_{2,X}, \widetilde{N}_X^r$ and $\widetilde{N}_X^p$ are noisy version of the following aggregate statistics:

$$R_X(s,a) = \sum_{h=1}^{H} r_h \mathbb{1}_{\{s_h=s,a_h=a\}}, \qquad R_{2,X}(s,a) = \sum_{h=1}^{H} r_h^2 \mathbb{1}_{\{s_h=s,a_h=a\}}$$

$$N_X^r(s,a) = \sum_{h=1}^{H} \mathbb{1}_{\{s_h=s,a_h=a\}}, \qquad N_X^p(s,a,s') = \sum_{h=1}^{H-1} \mathbb{1}_{\{s_h=s,a_h=a,s_{h+1}=s'\}}$$

In particular, $\widetilde{R}_X, \widetilde{N}_X^r$ and $\widetilde{N}_X^p$ are defined as for the optimistic algorithm in Section 4.1 and $\widetilde{R}_{2,X}$ is a privatized version of $R_{2,X}(s,a) = \sum_{h=1}^{H} r_h^2 \mathbb{1}_{\{s_h=s,a_h=a\}}$ for a trajectory $X$.s

**Algorithm 7** LDP-PSRL

---

**Input:** Initial values: $\alpha_0, \mu_0, \lambda_0, \nu_0$ and $\beta_0$
**for** episodes $k = 1, \ldots, K$ **do**
 Draw empirical MDP, $\theta_k$ from the posterior and compute $\pi_k$ as the optimal policy for MDP $\theta_k$
 User $u_k$ executes policy $\pi_k$, collect trajectory $X_k = \{(s_{k,h}, a_{k,h}, r_{k,h}) \mid h \leq H\}$
 Update noisy counts with $(\widetilde{R}_{X_k}(s,a), \widetilde{R}_{X_k,2}(s,a), \widetilde{N}^r_{X_k}(s,a), \widetilde{N}^p_{X_k}(s,a))$ and posterior distribution
**end for**

---

The posterior updates we use in LDP-PSRL are then for all $s' \in \mathcal{S}$:

$$\widetilde{\alpha}_k(s,a) = \alpha_0(s,a) + \widetilde{N}^p_k(s,a,s')$$

$$\widetilde{\mu}_k(s,a) = \frac{\lambda_0(s,a)\mu_0(s,a) + \widetilde{R}_k(s,a)}{\lambda_0(s,a) + \widetilde{N}^r_k(s,a)}$$

$$\widetilde{\lambda}_k(s,a) = \lambda_0(s,a) + \widetilde{N}^r_k(s,a)$$

$$\widetilde{\nu}_k(s,a) = \tilde{\nu}_0(s,a) + \frac{\widetilde{N}^r_k(s,a)}{2} \tag{102}$$

$$\widetilde{\beta}_k(s,a) = \beta_0(s,a) + \frac{\lambda_0(s,a)\widetilde{N}^r_k(s,a)\mu_0^2(s,a) - \widetilde{R}_k^2(s,a)}{2(\lambda_0(s,a) + \widetilde{N}^r_k(s,a))}$$
$$+ \frac{1}{2}\sum_{l \leq k-1} \widetilde{R}_{2,l} - \frac{\mu_0(s,a)\widetilde{R}_k(s,a)}{\lambda_0(s,a) + \widetilde{N}^r_k(s,a)}$$

In the following, we choose the Laplace mechanism as our privacy-preserving mechanism for LDP-PSRL, although we believe that it should be possible to use one of the other mechanisms we discussed. For each trajectory $X$, we add independent Laplace variables to $(R_X(s,a), R_{X,2}(s,a), N^r_X(s,a), N^p_X(s,a))$ with parameter $8H/\varepsilon$. Following the same argument outlined in the proof of Thm. 7, we can show that this privacy-preserving mechanism is $(\varepsilon, 0)$-LDP.

To ensure positivity, by concentration of Laplace variables we set the initial values of the parameters of the posterior distributions to:

$$\alpha_0(s,a,s') = \max\{\sqrt{KS}, \ln(6S^2A/\delta)\}\frac{\sqrt{8\ln(6S^2A/\delta)}}{\varepsilon_0} \tag{103}$$

$$\mu_0(s,a) = 0 \tag{104}$$

$$\lambda_0(s,a) = \max\{\sqrt{K}, \ln(6SA/\delta)\}\frac{\sqrt{8\ln(6SA/\delta)}}{\varepsilon_0} \tag{105}$$

$$\nu_0(s,a) = \max\{\sqrt{K}, \ln(6SA/\delta)\}\frac{\sqrt{8\ln(6SA/\delta)}}{\varepsilon_0} \tag{106}$$

$$\beta_0(s,a) = 5\max\{\sqrt{K}, \ln(6SA/\delta)\}\frac{\sqrt{8\ln(6SA/\delta)}}{\varepsilon_0} \tag{107}$$

where $K$ is the total number of episodes. The pseudocode of LDP-PSRL is reported in Alg. 7.

**Empirical results** We show empirical results for the LDP-PSRL algorithm in the RandomMDP environment in Figure 7. While we have shown that this algorithm is $\varepsilon$-LDP and empirically outperforms optimistic approaches, we leave the regret analysis to future work.

## H Additional Experiment

In this section, we explore a second experiment, in which we use the same the RandomMDP environment with the same parameters as in Sec. 6 in order to investigate the effect of differential privacy on the learning process. For this, we run the UCB-VI algorithm for $K = 10^3$ episodes and collect the aggregate noisy statistics, $(\widetilde{R}_K(s,a))_{(s,a)\in\mathcal{S}\times\mathcal{A}}$, $(\widetilde{N}^r_K(s,a))_{(s,a)\in\mathcal{S}\times\mathcal{A}}$ and

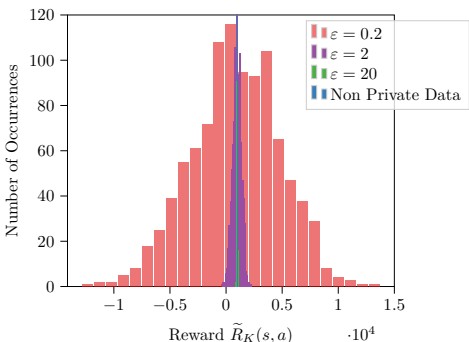

Figure 8: Aggregate reward for privatized data with $\varepsilon \in \{0.2, 2, 20\}$ and non-privatized data for state 0 and action 1

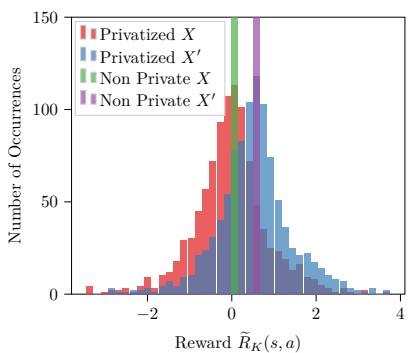

Figure 9: Privatized cumulative reward over an episode for a given state-action pair and two different trajectories $X$ and $X'$ with $\varepsilon = 20$ for state 0 and action 1

$(\widetilde{N}_K^p(s, a, s'))_{(s,a,s') \in \mathcal{S} \times \mathcal{A} \times \mathcal{S}}$ that have been generated by using the Laplace mechanism for each episode. We collect those statistics, $10^3$ times. We compare the histogram of those noisy statistics to that of the noiseless statistics used by UCB-VI in Fig. 8. This demonstrates that, as expected, there is much more variation in the statistics provided by the private mechanism. In Fig. 9, we applied the Laplace mechanism to two different random trajectories, $X$ and $X'$. We can see that, after applying the Laplace mechanism, the two distinct trajectories become almost indistinguishable. These two figures combined demonstrate the difficulty of learning from locally differentially private data.

## I  Privacy Amplification by Shuffling in RL

In recent years, the shuffle model for privacy [16, 17, 18, 19, 20, 45] has attracted a lot of attention thanks its amplification property tof the differential privacy guarantees of locally differential data.

In this model of privacy, we consider $n$ users equipped with a local differential privacy mechanism, each user submits a locally private report to a random shuffler which computes a random permutation of the users' reports. Those randomly shuffled reports are then sent to a analyzer which computes functions of interests based on them. This setting was first introduced in [57] and was named the *ESA* model (Encode-Shuffle-Analyze) and motivated by need for anonymous data collection. [45] later provided an analysis of the amplification of privacy thanks to the combined use of shuffling and local differential privacy showing that the shuffling model of privacy is able to strike a middle ground between the totally decentralized but somewhat sample inefficient *local* model and the centralized but more sample efficient central model of privacy.

The shuffling model has then been refined to study the impact on the size of the reports sent by users, i.e., how the accuracy of a shuffling protocol can be improved when user are allowed to have higher communication threshold [16, 58]. It has also been studied for different analyzer function, for instance histograms [59] or summation [16, 19], obtaining optimal protocol with better accuracy and lesser communication costs (i.e., the number of messages or the size of those messages sent by a user). Finally, the shuffle model has inspired a privacy amplification algorithm for learning in distributed setting without server-initiated communication [19].

Overall, the most attractive feature of this privacy model is that it offers a smooth transition in terms of privacy/utility tradeoff between stringent LDP requirements and differential privacy requirements (see [17] for an example of this transition in the problem of estimating a distribution).

Formally, in our RL setting each episode $k$ represents a user $u_k$ which completes a trajectory $X_{u_k}$ in the MDP. The user computes a locally private version of its trajectory thanks to a privacy-preserving mechanism $\mathcal{M}$. The result $\mathcal{M}(X_{u_k})$ is passed to a shuffler $\mathcal{R}$. This shuffler stores all the previous privatized trajectories before the current episode $k$, $(\mathcal{M}(X_{u_l}))_{l<k}$, computes a random permutation $\sigma : [k-1] \to [k-1]$ and sends the permuted set of privatized trajectories, $(\mathcal{M}(X_{u_{\sigma(l)}}))_{l \leq k-1}$ to an RL algorithm like LDP-OBI. This interaction protocol is detailed in Alg. 8.

---

**Algorithm 8** Shuffling Protocol

---

**Input:** number of episodes $K$, horizon $H$, failure probability $\delta \in (0,1)$, bias $\alpha > 1$, private randomizer $\mathcal{M}_{sh}$ with LDP parameters $(\epsilon_0, \delta_0)$
**for** $k = 1$ **to** $K$ **do**
    Shuffler $\mathcal{R}$ sends $(\mathcal{M}_{sh}(X_{u_{\sigma_k(l)}}))_{l \leq k-1}$ with $\sigma_k$ a random permutatioon at each episode
    LDP-OBI computes policy $\pi_k$ based on $(\mathcal{M}_{sh}(X_{u_{\sigma_k(l)}}))_{l \leq k-1}$
    User $u_k$ executes policy $\pi_k$ in the environment, collects trajectory $X_k = \{(s_{k,h}, a_{k,h}, r_{k,h})_{h \leq H}\}$ and sends the privatized trajectory $\mathcal{M}_{sh}(X_k)$ to $\mathcal{R}$
**end for**

---

---

**Algorithm 9** Local randomizer $R_p^{0/1}$

---

**Input:** randomization probability: $p \in [0,1]$, $x \in \{0,1\}$
Let $b \sim \text{Ber}(p)$
**if** $b = 0$ **then**
    Return $x$
**else**
    Return $\text{Ber}(1/2)$
**end if**

---

In the specific case of RL, thanks to [9] we know that any regret minimizing algorithm using $(\varepsilon, \delta)$-DP counters, like $(N_k^p)_{k \leq K}$ is $(\varepsilon, \delta)$-joint differentially private.

### I.1 Privacy-preserving mechanism $\mathcal{M}_{sh}$

A trajectory $X_u := \{(s_h, a_h, r_h) \mid h \leq H\}$ is a sequence of $H$ states, actions and rewards. In order to build a model of the MDP, LDP-OBI uses counters of the numbers of occurrences of each tuple of state-action $(s, a)$ and state, actions and next-state $(s, a, s')$. We adapt to the RL setting, the algorithm for bit-sum protocol presented in [16]. The first step of the process $\mathcal{M}_{sh}$ is to apply a one-hot encoding the trajectory for each state-action. Let $x \in \{0,1\}^{H \times S \times A}$ and $y \in \{0,1\}^{(H-1) \times S \times A \times S}$ such that for each $(s, a, s') \in \mathcal{S} \times \mathcal{A} \times \mathcal{S}$

$$\forall h \in [\![1, H]\!], \qquad x_{h,s,a} = \mathbb{1}_{\{s_h=s, a_h=a\}}, \text{ and } y_{h,s,a,s'} = \mathbb{1}_{\{s_h=s, a_h=a, s_{h+1}=s'\}} \tag{108}$$

To encode the reward, we first compute the reward for each state-action pair, $\left(r_h \mathbb{1}_{\{s_h=s, a_h=a\}}\right)_{(h,s,a) \in [\![1,H]\!] \times \mathcal{S} \times \mathcal{A}}$ then given a parameter $m \in \mathbb{N}^\star$ for each state-action pair $(s, a)$, we compute $b_{h,s,a} \in \{0,1\}^m$ such that for $j \in [\![1, m]\!]$:

$$(b_{h,s,a})_j = \begin{cases} 1 & \text{if } j < \mu_{h,s,a} \\ \text{Ber}\left(p_{h,s,a}\right) & \text{if } j = \mu_{h,s,a} \\ 0 & \text{if } j > \mu_{h,s,a} \end{cases} \tag{109}$$

with $\mu_{h,s,a} = \lceil m r_h \mathbb{1}_{\{s_h=s, a_h=a\}} \rceil$ and $p_{h,s,a} = m r_h \mathbb{1}_{\{s_h=s, a_h=a\}} - \mu_{h,s,a} + 1$.

It is a well known result, [16] that Alg. 9 with parameter $p$ guarantees $\ln(2/p - 1)$ differential privacy. Finally, the privacy-preserving mechanism $\mathcal{M}_{sh}$ is described by Alg. 10.

Using standard analysis, we can show that this local mechanism $R_p^{0/1}$ is roughly $H\varepsilon$-LDP for any $\varepsilon > 0$. Upon receiving the shuffled privatized, the algorithm LDP-OBI computes the different counts $(\tilde{N}_k^p(s, a, s'))_{(s,a,s')}$, $(\tilde{N}_k^r(s, a))_{(s,a)}$ and $(\tilde{R}_k(s, a))_{(s,a)}$. For any $(s, a, s') \in \mathcal{S} \times \mathcal{A} \times \mathcal{S}$, we define

---

**Algorithm 10** Privacy-preserving mechanism $\mathcal{M}_{sh}$

---

**Input:** trajectory $\tau = \{(s_h, a_h, r_h)_{h \leq H}\}$, privacy parameter $\varepsilon > 0$, parameter $m \in \mathbb{N}^\star$
Compute $x$ and $y$ as in Eq. (108) and $(b_{h,s,a})_{(s,a) \in \mathcal{S} \times \mathcal{A}}$ as in Eq. (109)
Set $p = \frac{2}{\exp(\varepsilon)+1}$
Return $(R_p^{0/1}(x_{h,s,a}))_{(h,s,a)}$, $(R_p^{0/1}(y_{h,s,a,s'}))_{(h,s,a,s')}$ and $((R_p^{0/1}((b_{h,s,a})_j)_{j \leq m})_{(h,s,a)}$

---

the counters as:

$$\tilde{N}_k^r(s,a) = \frac{1}{1-p}\left(\sum_{l=1}^{k-1}\sum_{h=1}^{H}\left[R_p^{0/1}(x_{h,s,a}) - \frac{p}{2}\right]\right) \tag{110}$$

$$\tilde{N}_k^p(s,a,s') = \frac{1}{1-p}\left(\sum_{l=1}^{k-1}\sum_{h=1}^{H}\left[R_p^{0/1}(y_{h,s,a,s'}) - \frac{p}{2}\right]\right) \tag{111}$$

$$\tilde{R}_k^r(s,a) = \frac{1}{m(1-p)}\left(\sum_{j=1}^{m}\sum_{l=1}^{k-1}\sum_{h=1}^{H}\left[R_p^{0/1}((b_{h,s,a})_j) - \frac{p}{2}\right]\right) \tag{112}$$

Therefore, thanks to Claim 4.6 of [16], we have at the beginning of episode $k$, $(\tilde{N}_k^r(s,a))_{(s,a)}$ and $(\tilde{N}_k^p(s,a,s'))_{(s,a,s')}$ are $(\varepsilon_{k,c}, \delta_0)$-DP with any $\delta_0 > 0$ and:

$$\varepsilon_{k,c} = \frac{32\log(4/\delta_0)/\sqrt{(k-1)H}}{\sqrt{p - \sqrt{\frac{2p\log(2/\delta_0)}{(k-1)H}}}}\left(1 - \left(p - \sqrt{\frac{2p\log(2/\delta_0)}{(k-1)H}}\right)\right) \tag{113}$$

with $p \in \left[\frac{14}{(k-1)H}\log(4/\delta_0), 1\right]$. But we have that with probability at least $1 - \delta$, for any $\delta > 0$, that:

$$\left|\sum_{l=1}^{k-1}\sum_{h=1}^{H}\mathbb{1}_{\{s_{l,h}=s, a_{l,h}=a\}} - \tilde{N}_k^r(s,a)\right| \le \frac{1}{1-p}\left(\sqrt{(k-1)Hp(1-p/2)\ln(1/\delta)} + \frac{2\ln(1/\delta)}{3}\right)$$

$$\left|\sum_{l=1}^{k-1}\sum_{h=1}^{H-1}\mathbb{1}_{\left\{\substack{s_{k,h}=s,\\a_{k,h}=a\\,s_{k,h+1}=s'}\right\}} - \tilde{N}_k^p(s,a,s')\right| \le \frac{1}{1-p}\left(\sqrt{(k-1)Hp(1-p/2)\ln(1/\delta)} + \frac{2\ln(1/\delta)}{3}\right)$$

The same type of result of result holds for the cumulative reward in each state-action pair $(s,a)$, albeit some small technical difficulties due the estimated sum being in $\mathbb{R}$ and not an integer contrary to the counters for the number of visits.

## I.2  Impact on the Regret

We have mentioned that thanks to the shuffling mechanism the counters $(\tilde{R}_k(s,a))_{(s,a)}$, $(\tilde{N}_k^r(s,a))_{(s,a)}$, $(\tilde{N}_k^p(s,a,s'))_{(s,a,s')}$ enjoy a $(\varepsilon_c, \delta)$-DP guarantee, in addition to the $\epsilon_0$-LDP guarantee. But the utility bound in the last subsection highlights that for a strict constraint on the level of local differential privacy the utility of each counters is of order $\frac{\sqrt{kH}}{\exp(\varepsilon_0)-1}$ therefore using Thm. 5, the regret of LDP-OBI coupled with $\mathcal{M}_{\text{sh}}$ is bounded with high probability by $\frac{H^2 S^2 A\sqrt{KH}}{\exp(\varepsilon_0/H)-1}$. This result is similar to the result of [17] of Sec. 5.1 about density estimation where the shuffle model recovers the known rate of convergence of $\mathcal{O}(1/\varepsilon\sqrt{n})$ under an $\varepsilon$-LDP constraint with $n$ samples.

However, in the reinforcement learning setting the shuffle model might allow to interpolate between LDP setting presented in this paper and the joint differential privacy setting of [7, 9]. One difficulty here being that because each user interacts only once with the RL algorithm the probability used by the local randomizer $R_p^{0/1}$ ha to be dependent on the number of previous episode to ensure a good $(\varepsilon, \delta)$-JDP guarantee. In other words, for the very first episodes the privacy amplification of the shuffle model is negligible therefore the privacy parameter for those early users has to be stronger than for the latter ones which are somewhat hidden by the crowd. Albeit this minor issue, a good choice of the probabilities $(p_i)_{k \le K}$ may be able to guarantee $(\varepsilon, \delta)$-JDP (for any $\varepsilon > 0$ and $\delta > 0$) and a regret of order $\mathcal{O}(\sqrt{K} + \frac{\log(K)}{\varepsilon})$.