# OpenReview forum: "Local Differential Privacy for Regret Minimization in Reinforcement Learning"
_NeurIPS.cc/2021/Conference — NeurIPS 2021 Poster_

### Official Review · Reviewer_fiXi · 2021-06-28

**Rating:** 6
**Confidence:** 5

**Summary:**

An interesting study on discrete MDP under LDP.

**Main Review:**

This manuscript investigates discrete MDPs under LDP constraints. The contribution is two-fold. First, it shows that the problem incurs a regret at least \Omega(K^1/2). The hardness is through the information theory-based technique previously used in bandits and a construction of an A-fold tree MDP. The contribution is significant. Second, armed with an LDP mechanism, it proposes a model-based exploration method for discrete MDP. The regret is upper bounded by O(H^3 S^2 A K^1/2) up to some logarithm factors. The upper bound is reasonably tight though there leaves a gap between the upper and the lower bounds. The techniques used in the proof of the upper bound are standard in discrete MDPs.

I suggest the authors have a thorough revision of their manuscript. The reading might not be very smooth. Here are a few examples:
line 17 - Privacy should be protected. It is not just due to that "users are becoming increasingly wary of that"
line 29 - circle 2 can be confusing (say something more explicitly when you define your scenarios)
line 55 - I don't see this footnote 2 very useful
line 57 - I don't think you can conclude that one problem is harder than another one simply by comparing their best known lower bounds. You'll need to compare a lower and an upper, or use a direct argument.
Also, the way the mathematical formulation are presented are not very professional (like the use of terms and punctuation).

**Time Spent Reviewing:**

4

---

> ### Author Response · Authors · 2021-08-10
> **Response to Reviewer fiXi**
>
> Thank you for your comments. We will improve the readability of the paper for the camera-ready version based on your comments, especially regarding the formulations of mathematical equations.
> 	We will use the final version extra-page to add more detail to the introduction (scenarios, privacy, comparison of model hardness…).
> 	Lastly, footnote $2$ was just included to prevent a mis-interpretation of the model; if you think that it is actually more confusing, we will remove it.

---

> > ### Comment · Reviewer_fiXi · 2021-08-14
> > **Thanks for responding**
> >
> > I have read your response and other reviewers comments. A major, common concern for the paper is that the technical contribution might be limited. I would also challenge the presentation (math formulation and language) of the manuscript. In case this manuscript is accepted, please spend some significant effort to improve the presentation for its accessibility.
> >
> > I would maintain my rating and my review.

---

### Official Review · Reviewer_QJNv · 2021-07-15

**Rating:** 6
**Confidence:** 1

**Summary:**

This paper introduces the concept of Local Differential Privacy for RL, a stronger notion of privacy than Joint Differential Privacy for RL. The authors both prove a regret bound for LDP in RL and provide a novel algorithm that achieves good regret in this setting. Furthermore, empirical experiments are provided that validate the method.


**Main Review:**

Unfortunately I am not familiar with this literature at all, but I will try to provide some good feedback.

My main concerns are that it seems that LDP is simply too strict of a privacy guarantee for RL, and that this will make it impractical in practice. With that being said, it seems that the main results in this paper show that decent results are possible in this setting, so this worry might be unfounded.

The authors submitted this work to two conferences before. At the first conference the reviewers said that the work was not well written and that the main contributions and importance of the problem were missing. I think these points have been solved, as the paper is very well written and the main contributions are clear. In the second conference, reviewers thought that the regret bound could be improved with shuffling. In lines 69 - 73 the authors address this issue, but not enough in my opinion. They refer to Theorem 5.2 in Feldman et. al., but it is not immediately clear to me why that theorem backs their claim. This could be because I am not familiar with the field, but regardless I think the paper could be improved with a more extensive discussion of this issue, perhaps in the appendix.

Other than that issue, I was not able to find any faults in the paper. The problem seems well motivated and the theory looks good. The experimental results back up the theory and the paper is well written.

--------------------------------
I have read the rebuttal and do not change my score.


**Time Spent Reviewing:**

3

---

> ### Author Response · Authors · 2021-08-10
> **Response to Reviewer QJNv**
>
> Thank you for your insightful comments. In the differential privacy literature the LDP constraint is indeed more stringent than the standard DP constraint. However, in practice, LDP is often preferred as it offers much more control over privacy guarantees for end users. We believe that this is also the case in RL and that it is important to study this privacy notion to understand the limitations of privacy in RL. In addition, we show that it is still possible to achieve a $\sqrt{K}$ regret (by paying a multiplicative cost in terms of privacy level).
>
> Concerning the shuffling mechanism, we provide an extensive discussion in the last 3 pages of the supplementary material, please refer to Appendix I. These developments are unfortunately not mentioned in the discussion in lines 69-73, so we will update this to include a pointer.
> Shuffling is indeed a very powerful tool in the DP literature. However the promise of this mechanism is to allow users to ensure a certain level of $(\varepsilon, \delta)$-DP with some additional $\varepsilon_{0}$-LDP guarantees, although in general we have $\varepsilon_{0} \geq \varepsilon$ but often $\varepsilon_{0} >> \varepsilon$. However, the scope of our paper is to study the RL problem with a given $\varepsilon_{0}$ LDP constraint. In this case, Thm. $5.1$ in [17] shows that in the density problem estimation shuffling does not provide any improvement in the final error (see Appendix I for details).

---

### Official Review · Reviewer_u5vy · 2021-07-16

**Rating:** 7
**Confidence:** 3

**Summary:**

The paper examines reinforcement learning under local differential privacy constraints. The paper proposes an algorithm, LDP-OBI, derives its regret bounds as well as privacy guarantees, and provided some simulation results.

**Limitations And Societal Impact:**

Limitations of the paper are clearly stated.

There are no potential negative societal impact.

**Main Review:**

- Originality: the paper focuses on the new intersection between local differential privacy and reinforcement learning. The author proposes local differential privacy adapted to RL and also applied existing privacy mechanisms in a RL setting, differentiating it from existing private RL works. Related works are cited and discussed thoroughly.

- Quality: the technical proofs are sound and simulations support the authors' claims in simple tasks. While implementation is not included in the supplement, the paper is largely theoretical and the experiment results should be sound.

- Clarity: the paper is clear and well-written.

- Significance: while local DP + RL is an interesting problem with great significance, LDP-OBI appears to be a direct combination of existing privacy mechanisms and existing RL algorithms and its regret bound is much greater than the derived lower bound. However, the results in the paper are still worth examining and the could pave way for future studies at the intersection of privacy and RL.

**Time Spent Reviewing:**

3

---

> ### Author Response · Authors · 2021-08-10
> **Response to Reviewer u5vy**
>
> Thank you for your thoughtful comments. This is indeed the first result about LDP in RL and highlights several interesting aspects about privacy. For example, we show that the application of privacy is not straightforward and requires a careful design of optimism and a potential limitation of model-based algorithms. The objective of the paper is indeed to pave the way for further study in the space of RL and privacy.

---

### Official Review · Reviewer_kXoY · 2021-07-16

**Rating:** 6
**Confidence:** 4

**Summary:**

This paper studied the regret analysis of reinforcement learning under the tabular setting and the LDP (local differential privacy) constraint. The authors first showed a minimax lower bound that, unlike the additive dependence on $\varepsilon$ under the JDP (joint differential privacy) constraint, the stronger LDP constraint must incur a minimax regret multiplicative in $1/\varepsilon$. Then the authors provided a general paradigm of the upper bound analysis: find private versions of aggregated statistics (rewards, number of visits), and then carry out an optimistic policy (called the LDP-OBI algorithm) based on (UCBs of) these noisy statistics. Depending on the usage of different private mechanisms, a regret multiplicative in $\sqrt{K}/\varepsilon$ could be obtained, but with worse dependence on other parameters $(H,S,A)$. Discussions and numerical experiments on the result are also provided.

**Main Review:**

Overall I like this paper. The LDP constraint in reinforcement learning is useful to preserve privacy, and the separation between LDP and JDP constraints shown in the paper is interesting. The authors proposed a general paradigm for the upper bound analysis which could be useful in other papers, and also gave a lower bound analysis with a tight dependence on $(K,\varepsilon)$ to support the optimality of the proposed mechanism. The writing is mostly clear to me.

However, I can also list some drawbacks of this work:

1. The upper bound analysis, mainly Section 4.2, is technical while quite standard. I do like the current presentation which summarizes all necessary conditions as Assumption 3 and proves a general result in Proposition 4, but the essentially the same argument was applied in the non-constrained RL setting, and it seems that this part does not add much novelty to the RL literature. If this is the case, then the contribution of just applying various private mechanisms to obtain aggregated statistics is limited. I suggest the authors to highlight the main novel points in Section 4.2 to differentiate from the non-constrained RL regret analysis.

2. The current upper bound has a potentially loose dependence on parameters (H,S,A). This may suggest that the privacy constraint is currently not handled in a most appropriate way, and there might be room for significant improvement. In fact, Table I already shows that using one private mechanism could have a much worse performance than using another one.

3. In the discussion the authors provided some evidence that the current dependence on (H,S,A) might not be tight. If this is the case, ideally I hope the authors could provide a rigorous argument (a partial result is okay) instead of only conjecturing it. I have to say that the current lower bound analysis is pretty standard - it is essentially the estimation of the bias of a coin under the LDP constraint (or in other words, a simple combination of the non-constrained case and the strong data-processing inequality applied to the LDP constraint). So a stronger lower bound would also increase the significance of the lower bound.

One minor comment: I do not understand why a minimum is present in Eqn. (14). Isn't it the case that by the data-processing inequality for general f-divergences, the first term is always no greater than the second term? Consequently, I also do not understand why these two terms need to be obtained from two different equations (11) and (13).

**Time Spent Reviewing:**

3

---

> ### Author Response · Authors · 2021-08-10
> **Response to Reviewer kXoY**
>
> Thank you for your insightful comments. Concerning the proof of the upper-bound it is true that the result depends on the optimism principle, which is now standard, and thus can “feel” familiar. Our contribution in this direction is the derivation of the new confidence bonuses adapted to the privacy preserving mechanism used to ensure LDP, see Proposition $4$ for instance. We will update the submission to highlight this.
>
> Your second point highlights that the suboptimal dependency of the regret upper bound  on  $(H, S, A)$ may be due to a suboptimal handling of the privacy constraint. We acknowledge the suboptimal dependency but we suspect that improving the upper bound is non trivial. There seem to be two potential ways of doing this. The first is designing a model-free algorithm instead of a model-based one (as done here). This may improve the dependency on the size of the state space and action space, as we point out in the discussion in Section $5$. Another possible improvement would indeed be to design a more adapted privacy-preserving scheme, as you suggest. The impact on the regret of changing the privacy-preserving mechanism can be simply derived by computing the different quantities appearing in  Assumption $3$. Indeed, one of the advantages of our result is its generality. Our result is more general than just “one regret for one mechanism” which means that if an improved mechanism is proposed, updating our result is quite easy using Theorem $5$.
>
> We would also like to point out that it is not clear by how much the lower-bound derived in the paper is suboptimal. Indeed the justifications after Section $5$ only apply to model-based algorithms (as ours). The recent work in lower bounds for LDP estimation ([Duchi and Rogers, 2019]) can not be applied in RL, the main objective is not to estimate the MDP but to minimize the regret which does not necessarily require estimating the model. This is why we suspect a model-free private LDP algorithm could move closer to the current lower-bound; but this is far beyond the scope of the current paper and left for future (or current actually) research.
>
> Finally, regarding your minor comment on the minimum in the lower-bound: it is necessary otherwise, for $\varepsilon \rightarrow +\infty$, the first term would dominate the second one and yield a lower bound of $0$.

---

> > ### Author Response · Authors · 2021-08-24
> > **Follow-up on the Response**
> >
> > Hello reviewer kXoY, we would be grateful if you can confirm whether our response has addressed your concerns, and let us know if any issues remain.
> > To recap our response:
> >  - Concerning your first point while the proof of regret can "feel" familiar we derive new confidence intervals that take into account the additional noise necessary to satisfy the privacy guarantee.
> >  - About the second point you raised, we indeed acknowledge the suboptimal dependency of the bound on the parameters $(H,S,A)$. We suspect the first way to improve the result, is to design a model-free algorithm as stated in Section $5$. The second way is to design a better privacy-preserving algorithm but note that our regret analysis is valid for any privacy-preserving algorithm satisfying Assumption $3$.
> >  - Regarding your last point, there is no indication that the lower bound we present in this paper is suboptimal as the current literature in privacy focuses on estimation (like density estimation or mean estimation)
> >  but the problem we are tackling is to minimize the regret. Something much different from estimating a model and that's why we feel a model-free algorithm may be more adapted.

---

> > > ### Comment · Reviewer_kXoY · 2021-08-26
> > > **Thanks for responding**
> > >
> > > Thanks for your detailed response. I agree with the authors that the current model-based private upper bound of the regret is of sufficient interest and significance, and novel techniques are necessary to improve the current upper and/or lower bounds. However, since this gap remains and is still an important factor in RL, I'd like to keep my score here and lean towards acceptance.
> > >
> > > A minor comment regarding your comparison with [Duchi and Rogers 2019]: in their paper they also considered the excess risk, which is also not the estimation error, and is also similar in spirit to the regret. I also do not want to distinguish too much between regret and estimation error: in most bandit and RL lower bounds, establishing the regret lower bound is reduced to lower bounding some estimation error, and the proof techniques are essentially similar.
> > >
> > > Regarding the minor point on data-processing inequality: you're talking about your upper bounds in (22) and (24), which could be potentially loose. If we directly look at the terms in (14), it is always the case that the first term is no larger than the second term. This is just the data-processing inequality for the KL divergence (or general f-divergences): $KL(P_{f(X)} \| P_{f(Y)} ) \leq KL(P_X \| P_Y)$

---

### Decision · Program_Chairs · 2021-09-27

**Decision:**

Accept (Poster)

**Comment:**

This paper studied the regret analysis of reinforcement learning under the tabular setting and the LDP (local differential privacy) constraint. All the reviewers believe that the paper is above the acceptance line. The reviewers also raised various minor concerns, e.g., the paper might read very smooth, there exists a gap in the upper and lower bound. I encourage the authors to address these comments in the next revision.